# Centriolar cap proteins CP110 and CPAP control slow elongation of microtubule plus ends

Saishree S. Iyer[1]*, Fangrui Chen[1]*, Funso E. Ogunmolu[1]*, Shoeib Moradi[2]*, Vladimir A. Volkov[1,3]*, Emma J. van Grinsven[1]*, Chris van Hoorn[1], Jingchao Wu[1], Nemo Andrea[3], Shasha Hua[1], Kai Jiang[1], Ioannis Vakonakis[4], Mia Potočnjak[5], Franz Herzog[5], Benoît Gigant[6], Nikita Gudimchuk[7], Kelly E. Stecker[8,9], Marileen Dogterom[3], Michel O. Steinmetz[2,10], and Anna Akhmanova[1]

**Centrioles are microtubule-based organelles required for the formation of centrosomes and cilia. Centriolar microtubules, unlike their cytosolic counterparts, are stable and grow very slowly, but the underlying mechanisms are poorly understood. Here, we reconstituted in vitro the interplay between the proteins that cap distal centriole ends and control their elongation: CP110, CEP97, and CPAP/SAS-4. We found that whereas CEP97 does not bind to microtubules directly, CP110 autonomously binds microtubule plus ends, blocks their growth, and inhibits depolymerization. Cryo-electron tomography revealed that CP110 associates with the luminal side of microtubule plus ends and suppresses protofilament flaring. CP110 directly interacts with CPAP, which acts as a microtubule polymerase that overcomes CP110-induced growth inhibition. Together, the two proteins impose extremely slow processive microtubule growth. Disruption of CP110–CPAP interaction in cells inhibits centriole elongation and increases incidence of centriole defects. Our findings reveal how two centriolar cap proteins with opposing activities regulate microtubule plus-end elongation and explain their antagonistic relationship during centriole formation.**

## Introduction

Centrioles are conserved organelles important for diverse processes such as cell division, motility, polarity, and signaling. They are required for the assembly of centrosomes, the major microtubule (MT)-organizing centers in animal cells, and form the basal bodies of cilia and flagella (reviewed in Azimzadeh and Marshall [2010]; Banterle and Gonczy [2017]; Bornens [2012]; Loncarek and Bettencourt-Dias [2018]; Nigg and Stearns [2011]). Centriole defects have been linked to different human diseases, such as cancer, and to developmental disorders, including microcephaly and ciliopathies (Azimzadeh and Marshall, 2010; Banterle and Gonczy, 2017; Bornens, 2012; Loncarek and Bettencourt-Dias, 2018; Nigg and Stearns, 2011).

Centrioles are barrel-like structures, which typically contain nine MT triplets. Unlike cytoplasmic MTs, which grow at a rate of 10–20 μm/min, centriolar MTs elongate with a rate of a few tens of nanometers per hour (Aydogan et al., 2018; Chretien et al., 1997; Kuriyama and Borisy, 1981). This can be explained by the presence of specific centriolar factors that stabilize MTs and control their growth. Previous work has shown that centrosomal-P4.1–associated protein (CPAP or SAS-4 in worms and flies) is essential for centriole formation, elongation, and maintenance ([Vasquez-Limeta et al., 2022]; reviewed in Banterle and Gonczy [2017]; Sharma et al. [2021]). CPAP causes overelongation of centrioles when overexpressed (Kohlmaier et al., 2009; Schmidt et al., 2009; Tang et al., 2009), but it can

[1]Cell Biology, Neurobiology and Biophysics, Department of Biology, Faculty of Science, Utrecht University, Utrecht, The Netherlands;   [2]Division of Biology and Chemistry, Laboratory of Biomolecular Research, Paul Scherrer Institut, Villigen, Switzerland;   [3]Department of Bionanoscience, Kavli Institute of Nanoscience, Delft University of Technology, Delft, The Netherlands;   [4]Department of Biochemistry, University of Oxford, Oxford, UK;   [5]Ludwig-Maximilians-Universität München, Munich, Germany; [6]CEA, CNRS, Institute for Integrative Biology of the Cell (I2BC), Université Paris-Saclay, Gif-sur-Yvette, France;   [7]Department of Physics, and Center for Theoretical Problems of Physico-Chemical Pharmacology, Russian Academy of Sciences, Lomonosov Moscow State University, Moscow, Russia;   [8]Biomolecular Mass Spectrometry and Proteomics, Bijvoet Center for Biomolecular Research and Utrecht Institute for Pharmaceutical Sciences, Utrecht University, Utrecht, The Netherlands;   [9]Netherlands Proteomics Center, Utrecht, The Netherlands;   [10]University of Basel, Biozentrum, Basel, Switzerland.

*S.S. Iyer, F. Chen, F.E. Ogunmolu, S. Moradi, V.A. Volkov, and E.J. van Grinsven contributed equally to this paper.   Correspondence to Anna Akhmanova: a.akhmanova@uu.nl;   Marileen Dogterom: m.dogterom@tudelft.nl;   Michel O. Steinmetz: michel.steinmetz@psi.ch

V.A. Volkov's current affiliation is School of Biological and Behavioural Sciences, Queen Mary University of London, London, UK.   S. Hua's and K. Jiang's current affiliation is State Key Laboratory of Oral & Maxillofacial Reconstruction and Regeneration, Key Laboratory of Oral Biomedicine Ministry of Education, Hubei Key Laboratory of Stomatology, School & Hospital of Stomatology, Medical Research Institute, Wuhan University, Wuhan, China.   I. Vakonakis's current affiliation is Lonza Biologics, Lonza Ltd., Visp, Switzerland.   F. Herzog's current affiliation is Institute Krems Bioanalytics, IMC University of Applied Sciences Krems, Krems, Austria.   A. Akhmanova is a lead contact.

also prevent outgrowth of long MT extensions from the distal centriole end (Sharma et al., 2016). CPAP is present at different sites at the centriole, including its distal end (Laporte et al., 2024; Vasquez-Limeta et al., 2022). In vitro reconstitutions showed that CPAP tracks growing MT ends and stabilizes MTs by preventing catastrophes and causing a fourfold reduction of the MT growth rate (Sharma et al., 2016). CPAP performs these functions through a combination of its MT-binding domain and its tubulin-binding domain that can cap MT plus ends and occlude the surface of the tip-exposed β-tubulin (Campanacci et al., 2022; Sharma et al., 2016; Zheng et al., 2016). However, these effects of CPAP on MT polymerization are not sufficient to explain how the elongation of centriolar MTs is controlled.

Other strong candidates for regulating centriolar MT plus-end growth are CP110 and CEP97. Together, they control centriole elongation and prevent overextension of the plus ends of the triplet MTs (Kleylein-Sohn et al., 2007; Kohlmaier et al., 2009; Schmidt et al., 2009; Spektor et al., 2007). The effects of CP110 and CEP97 on centriole length are species- and cell-specific. In mammalian cells, CP110 and CEP97 counteract the ability of CPAP to promote centriole elongation (Kohlmaier et al., 2009; Schmidt et al., 2009). In different types of *Drosophila* cells and tissues, both elongation and shrinkage of centrioles were reported upon the loss of CP110 and CEP97 (Aydogan et al., 2018; Delgehyr et al., 2012; Dobbelaere et al., 2020; Franz et al., 2013; Shoda et al., 2021). The emerging picture from these studies is that CP110 and CEP97 can counteract changes in centriole length imposed by positive or negative regulators of MT growth (Delgehyr et al., 2012; Sharma et al., 2021; Shoda et al., 2021). CP110 and CEP97 are also required for early stages of cilia formation (Dobbelaere et al., 2020; Walentek et al., 2016; Yadav et al., 2016), but the cap that these proteins form needs to be removed from the basal body to allow the formation of axonemal MTs (Goetz et al., 2012; Huang et al., 2018; Prosser and Morrison, 2015; Spektor et al., 2007).

While genetic and cell biological studies strongly support the role of CP110, CEP97, and CPAP in forming a regulatory cap at the distal centriolar end, biochemical understanding of their interplay is limited. It is well established that CP110 and CEP97 interact with each other and with other factors involved in the biogenesis of centrioles and cilia, such as CPAP (Galletta et al., 2016; Gupta et al., 2015; Jiang et al., 2012; Kobayashi et al., 2011; Sharma et al., 2021; Spektor et al., 2007; Tsang et al., 2006, 2008, 2009). However, it is currently unknown whether CP110 and CEP97 directly interact with MTs and whether they can affect MT growth, either alone or in combination with CPAP.

Here, we reconstituted in vitro the activities of purified CP110, CEP97, and CPAP on dynamic MTs. We found that CP110 can specifically bind to MT plus ends and block their growth, whereas CEP97 does not interact with MTs directly. CP110 can also directly bind to CPAP, and this interaction allows CPAP to overcome CP110-induced block of MT plus-end growth. Together, these proteins impose extremely slow but processive MT growth. Disruption of CP110–CPAP interaction in cells inhibited procentriole elongation and caused structural abnormalities in centrioles. Cryo-electron tomography (cryo-ET) indicated that CP110 interacts with the luminal side of MT plus ends, whereas

CPAP is known to bind to the outer MT surface (Campanacci et al., 2022; Sharma et al., 2016; Zheng et al., 2016). Thus, our data suggest that together, a luminal MT-pausing factor CP110 and a MT polymerase CPAP can span the MT tip and stabilize it in a slowly growing state.

## Results

### CP110 binds to MT plus ends and blocks their growth

To investigate the direct effects of CP110 and CEP97 on MT growth, we used in vitro reconstitution assays in which MTs polymerizing from GMPCPP-stabilized seeds that are attached to a glass slide are observed by total internal reflection fluorescence (TIRF) microscopy (Bieling et al., 2007; Sharma et al., 2016). Full-length CP110 with an N-terminal twin-Strep-tag (SII) and GFP tag and full-length CEP97 with a C-terminal GFP and SII tag were purified from HEK293T cells (Fig. 1 A and Fig. S1 A). The main contaminants detected by mass spectrometry (Fig. S1 B) were tubulins and the heat shock protein Hsp70, which we often observe in our protein preparations and which, to our knowledge, have no effect on MT dynamics (van den Berg et al., 2023). We observed that GFP-CP110 could bind to the plus ends of seeds and block their elongation at concentrations above 30 nM, whereas MT minus ends, which could be distinguished by their slower polymerization rate, were not affected (Fig. 1, B and C; and Fig. S1 C). At concentrations <30 nM, GFP-CP110 could occasionally bind to MT plus ends and induce pausing followed by catastrophes (Fig. 1 C). In contrast, CEP97-GFP displayed no binding to MTs and no effect on their dynamics (Fig. S1 D). The addition of up to 240 nM CEP97-GFP to the assays with 30 nM GFP-CP110 had no effect on MT seed blocking by GFP-CP110 (Fig. S1 E). Unfortunately, in all these assays, we observed significant aggregation of CP110, which complicated quantitative analyses, and the addition of CEP97 did not solve this problem.

We next generated different GFP-tagged CP110 deletion mutants. Since the N terminus of CP110 is known to bind to the middle part of CEP97 (Spektor et al., 2007), we hypothesized that the C terminus of CP110 is responsible for blocking MT growth. When expressed in cells, full-length CP110 and CEP97, as well as the N-terminal CP110 fragment 1–700 localized to centrosomes. However, GFP-tagged C-terminal CP110 fragments 581–991 and 791–991 did not associate with either centrosomes or MTs (Fig. S2 A), and we were not successful in purifying them from HEK293T cells. Since CP110 normally functions in association with CEP97, we reasoned that a direct fusion with CEP97 might improve the folding or stability of the CP110. We screened different fusion constructs by their localization in U2OS cells and found that a protein containing residues 1–650 of CEP97 and residues 581–991 of CP110 (termed here CEP97^CP110, Fig. 1 A) localized to centrioles (Fig. S2 A). Ultrastructure expansion microscopy (U-ExM), which entails 4.5× expansion of cells, followed by confocal imaging (Gambarotto et al., 2019) demonstrated that CEP97^CP110-GFP bound to the distal centriole ends, similar to CP110 (Fig. S2 B). In vitro, SII and GFP-tagged CEP97^CP110 blocked MT seed elongation and was much less aggregation prone than full-length CP110 (Fig. 1, D–F; and Video 1). While CEP97^CP110 did not bind along MT shafts, it

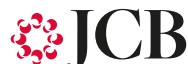

Figure 1. **CP110 binds to MT plus ends and blocks their growth. (A)** Scheme illustrating the domain organization of GFP-tagged human CEP97, CP110, and CEP97^CP110 chimeric constructs. Domain nomenclature: LRR, leucine-rich region; CC, coiled coil; IQ is the calmodulin-binding domain; GFP, green fluorescent protein; and SII, twin-Strep-tag. **(B and D)** Representative fields of view from time-lapse movies of in vitro reconstitution of MT growth from GMPCPP-stabilized seeds (blue) in presence of 15 μM tubulin (gray) and 30 nM GFP-CP110 (green) (B) or 80 nM CEP97^CP110-GFP (green) (D); blocked plus ends are indicated with arrowheads. **(C and E)** Kymographs illustrating a dynamic MT without CP110 or CEP97^CP110 binding, transient pausing, or plus end blocking by

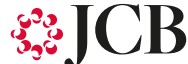

CP110 (C) or CEP97^CP110 (E); growth, pauses, or blocking events are indicated by white arrows. The plus and minus ends of the MTs are indicated by "+" and "−," respectively, and lines below kymographs indicate the position of the GMPCPP-stabilized seed. **(F and G)** Percentage of MTs displaying no pauses, occasional pauses, or fully blocked seeds (F) and pause duration (G), observed over 10 min with increasing concentrations of CEP97^CP110. Plots show percentage mean-SEM (F) and median ± interquartile range (IQR) of pause duration (G) at different CEP97^CP110 concentrations, with data points showing individual MT plus ends. Nonsignificant (ns), P > 0.05; **P = 0.0036 with Kruskal–Wallis ANOVA with Dunn's test for multiple comparisons. n = 7, 14, 13, 20, 30, and 34 MT plus ends for 2, 4, 7.5, 10, 20, and 40 nM of CEP97^CP110, respectively. Number of independent assays was 3, 3, 4, 3, 3, 4 for 2, 4, 7.5, 10, 20, 40, and 80 nM CEP97^CP110. **(H)** Histograms of fluorescence intensities of single molecules of GFP (n = 6,865), GFP-EB3 (n = 14,082), and CEP97^CP110-GFP (n = 6,942) immobilized on coverslips (symbols) and the corresponding fits with lognormal distributions (lines). The inset shows the number of CEP97^CP110-GFP molecules present at a paused or blocked MT plus end. The values were obtained by comparing the fitted mean intensity of CEP97^CP110-GFP at MT tips with the fitted mean intensity of single GFP molecules in parallel chambers. Floating bars represent maximum to minimum intensities of CEP97^CP110-GFP molecules relative to GFP per condition, with the line showing the mean value (n = 23 MTs for paused MTs at 7.5 nM, for blocked MTs n = 15 at 7.5 nM, n = 22 at 40 nM, and n = 28 at 80 nM). **(I)** Kymographs showing unbleached control and bleached CEP97^CP110-GFP at a blocked MT plus end. White arrow shows the moment of bleaching. **(J)** Mean + SD of the normalized intensity of CEP97^CP110-GFP at the MT plus end with (n = 28 MTs) and without bleaching (n = 12 MTs) from three independent assays. Frames were acquired at 2 s time interval. **(K)** Kymographs showing MT plus ends blocked with 2 µM unlabeled DARPin-(TM-3)$_2$ alone (right) or in combination with 3 nM (middle) or 40 nM (right) CEP97^CP110-GFP (green). CEP97^CP110-GFP was bound for a part of the observation time (partial) or for the whole 10 min duration of the movie (full). **(L)** Percentage of MT plus ends blocked by DARPin-(TM-3)$_2$ that also have CEP97^CP110 bound to them at 3 nM (n = 91 MTs) and 40 nM (n = 110 MTs) in two and four independent assays, respectively. **(M)** Mean-SEM of fluorescence intensity of CEP97^CP110-GFP on MT plus ends in presence (n = 83 MTs) and absence (n = 76 MTs) of DARPin-(TM-3)$_2$ in two and four independent assays, respectively. Nonsignificant (ns), P = 0.626 with a two tailed Mann–Whitney U test.

specifically bound to MT plus ends and completely blocked their growth at concentrations exceeding 80 nM, whereas MT minus ends underwent normal dynamics (Video 1). At lower concentrations of CEP97^CP110-GFP, MTs could still grow from both ends, but the binding of CEP97^CP110 caused transient plus-end pauses with an average duration ranging between 0.28 and 0.72 min, depending on the concentration (Fig. 1 G). Transient pauses were always followed by MT depolymerization (Fig. 1 E). These results demonstrate that the CP110-CEP97 fusion inhibits MT plus-end growth similarly to CP110 alone, and hence, we can use it to study the effect of CP110 on MT dynamics.

Next, we used measurements of fluorescence intensity to determine how many molecules of CEP97^CP110 are sufficient to block MT growth. By comparing the intensity of individual GFP-tagged CEP97^CP110 molecules immobilized on glass to the intensity of single molecules of purified GFP (monomers) or GFP-EB3 (dimers), we found that CEP97^CP110-GFP is a dimer (Fig. 1 H). We then compared the intensity of CEP97^CP110-GFP blocking or pausing a MT tip to the intensity of individual molecules of the same protein immobilized on glass in a separate, parallel chamber. We found that, on average, four CEP97^CP110-GFP molecules (two dimers) were observed at MT ends undergoing transient pausing at 7.5 nM, and six CEP97^CP110-GFP molecules (three dimers) were typically seen at the fully blocked tips of the seeds at concentrations between 7.5 and 80 nM (Fig. 1 H, inset). The total number of CEP97^CP110-GFP molecules bound to the MT plus end rarely exceeded 10 monomers, which is lower than the number of protofilaments (PFs) present in GMPCPP-stabilized MTs that predominantly contain 14 PFs (Hyman et al., 1995). Next, to examine the turnover of CEP97^CP110-GFP on blocked MT plus ends, we used FRAP and found that the protein displayed no turnover over a duration of 10 min (Fig. 1, I and J). Taken together, these data suggest that a relatively small number of CEP97^CP110-GFP molecules (fewer than the number of MT PFs) is sufficient to arrest MT plus-end growth and that they do so by stably binding to MT tips. Since CEP97 does not associate with MTs on its own, this binding depends on the C-terminal half (residues 581–991) of CP110.

The most obvious way for a protein to block MT plus-end growth is by occluding the longitudinal interface of β-tubulin and prevent α-tubulin from binding to it. An agent known to have such an activity is the tubulin-specific designed ankyrin repeat protein (DARPin), which binds to β-tubulin and inhibits subunit addition to the plus end (Ahmad et al., 2016; Pecqueur et al., 2012). We tested the potential competition between CEP97^CP110-GFP and DARPin by using (TM-3)$_2$, a dimeric version of the high affinity DARPin TM-3 (Ahmad et al., 2016; Campanacci et al., 2019). 2 µM of the DARPin-(TM-3)$_2$ completely blocked the elongation of MT plus ends, but not minus ends, in the presence of 15 µM soluble tubulin (Fig. 1 K). However, even when present at a 3 nM concentration, CEP97^CP110-GFP could still bind to and decorate ~50% of such blocked MT plus ends (Fig. 1, K and L). We also found no difference in the intensity of 40 nM CEP97^CP110-GFP at the MT plus ends in the presence or absence of 2 µM DARPin-(TM-3)$_2$ (Fig. 1 M). These data indicate that a very large molar excess of tubulin dimers and DARPin, both of which have a strong affinity for the plus-end–exposed part of β-tubulin, cannot compete CEP97^CP110 off, suggesting that the binding of CEP97^CP110 to MT tips does not rely on the interaction with the longitudinal interface of β-tubulin.

## CP110 binds to MT plus ends from the luminal side and suppresses PF flaring

To get insight into the binding mode of the CEP97^CP110 to MT plus ends and its effect on MT tip structure, we turned to cryo-ET. We reconstructed 3D volumes containing MTs grown in the presence or absence of 80 nM CEP97^CP110-GFP. We assumed that the majority of MTs must be elongating, as in our in vitro assays, the time MTs spend growing is much longer than the time they spend shortening. The use of a recently developed denoising algorithm (Buchholz et al., 2019) allowed to significantly improve the segmentation and tracing of individual PFs at MT ends. As reported previously (McIntosh et al., 2018), most MT ends in our samples terminated with curved PFs (Fig. 2 A).

As expected, MT growth from GMPCPP-stabilized seeds produced primarily 14-PF MTs (170 out of 202; 84%), which

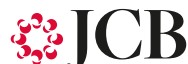

Figure 2. **CEP97^CP110 forms caps at the plus ends of dynamic MT and straightens their PFs. (A)** Slices through denoised tomograms containing MT plus ends in the absence or presence of 80 nM CEP97^CP110-GFP and 15 µM tubulin. **(B)** Segmented and 3D rendered volumes containing MT plus ends (blue), capping density (green), and manually segmented 3D models of traced PF shapes (orange). Arrows point to soluble tubulin oligomers. **(C)** Fraction of MT ends associated with a capping density. Data points show individual grids, line shows mean ± SD. **(D)** Scheme showing the parameters extracted from manual segmentations of terminal PFs. **(E)** All PF traces obtained from plus ends, aligned at their origin, in presence of 15 µM tubulin alone (right), with CEP97^CP110-

GFP cap (middle) and uncapped in presence of CEP97^CP110 (right). **(F–H)** Average PF lengths (F), average curvatures (G), and average terminal curvatures (H) of PFs with nonzero length for samples imaged in the presence of 15 μM soluble tubulin. Shown are average values within each MT (dots), their mean and SD (error bars); ****P < 0.0001, *P < 0.05; ns—nonsignificant with one-way ANOVA followed by Tukey's multiple comparison test; n is the number of MTs analyzed for each data set. Data distribution was assumed to be normal, but this was not formally tested. **(I)** Mean ± SEM of the curvature of PFs, aligned at their distal tips. The straight lines show the results of linear fitting. **(J)** Correlation between average curvature and average PF length per MT plus end. r, Pearson correlation coefficient; p, probability that the slope of the correlation is different from zero; and n showing number of MTs is mentioned in the figure.

allowed for unambiguous polarity determination of most MT ends (Fig. S3, A and B) (Chretien et al., 1996). Interestingly, in the presence of CEP97^CP110, we observed "caps" at MT ends, which were attached to a subset of PFs (partially capped) or blocking the whole MT lumen (fully capped) (Fig. 2, A and B; Fig. S3 B; and Video 2). Capping densities were observed much more frequently at MT plus ends (Fig. 2 C): 78% of plus ends carried a cap (38 out of 52), compared with only 9% of capped minus ends (5 out of 56). Some MT plus ends were attached to larger structures, which we also considered as full caps (Fig. S3 B). Out of three sample preparations with 15 μM soluble tubulin and CEP97^CP110, two were prepared with CEP97^CP110 added after the tubulin mix was subjected to high-speed centrifugation, and this led to the presence of large structures presumably formed by the chimeric protein (see Fig. S3 B for examples). In the sample with CEP97^CP110 added to the tubulin mix before centrifugation, we still observed caps predominantly at plus ends (50% capped plus ends and 9% capped minus ends); however, no full caps were seen in this sample. Therefore, fully capped MTs in our assays likely carry many more copies of CEP97^CP110 than determined by our TIRF assays (Fig. 1 H), which were performed after centrifugation of the tubulin-CEP97^CP110 mix. Importantly, most caps appeared to interact with the luminal side of the PFs (Fig. 2, A and B; Fig. S3 B; and Video 2).

To determine whether CEP97^CP110–mediated capping affected PF shapes at MT ends, we manually traced PFs in tomograms (Fig. 2, B and D). From these manually segmented 3D models, we obtained PF length (measured from the first segment bending away from the MT cylinder), curvature along the PF, and terminal curvature (Fig. 2 D). Contrary to a previous report (McIntosh et al., 2018), PFs in our samples frequently deviated from their planes (Fig. S3 C). Due to this difference, we modified the previously reported analysis to account for the full 3D coordinates of terminal PFs (Fig. 2 E, see Materials and methods for details).

The presence of a CEP97^CP110 cap correlated with shorter PFs at dynamic MT plus ends; PFs at non-capped MT ends in the presence of the protein were not different from those imaged in its absence (Fig. 2 F). The average curvatures and average terminal curvatures of the PFs at either plus or minus MT ends were not significantly altered by the presence of CEP97^CP110 (Fig. 2, G–I), and no significant correlation was observed between the curvatures and PF lengths (Fig. 2 J).

Since CEP97^CP110 blocked MT growth at the seed in our TIRF experiments (Fig. 1, D–F), we wondered whether the changes we observed in the lengths of the PFs were related to the tubulin incorporation into MT tips. To address this question, we

also analyzed MT end structures without soluble tubulin and at a low tubulin concentration (3 μM), with and without CEP97^CP110. In the absence of soluble tubulin, GMPCPP-stabilized seeds depolymerized, while the presence of 3 μM tubulin in solution protected the seeds without elongating them (Fig. 3 A). Also in the absence soluble tubulin, CEP97^CP110 formed tip-associated luminal structures (Fig. 3 B). The MT-capping frequency was the same as in the presence of free tubulin: 78% of GMPCPP seeds were capped or attached end on to large structures (38 out of 49) compared with 3% of capped minus ends (1 out of 30) (Fig. 2 C). For uncapped MTs, the shortest mean PF length was observed at 3 μM tubulin, while in the absence of soluble tubulin, uncapped MT ends displayed more pronounced peeling, probably due to their disassembly (Fig. 3, B and C). In contrast, the lengths of the PFs at the tips of CEP97^CP110–capped MTs remained unchanged regardless of tubulin concentration (Fig. 3 C), while average PF curvature was the same in all conditions (Fig. 3 D). We conclude that CEP97^CP110 reduces PF peeling at MT plus ends, to which it likely binds from the luminal side.

## CP110 directly binds to CPAP

Having established that CP110 binds to the inward-facing PF surface at the MT tip, we next wondered about its interplay with CPAP. Evidence for the interaction between the two mammalian proteins has been provided by proximity mapping (Gupta et al., 2015) and by yeast two-hybrid assays with their fly homologs (Galletta et al., 2016). We co-expressed in HEK293T cells full-length CP110 and CPAP or their fragments tagged with either GFP alone or GFP and a biotinylation tag together with biotin ligase BirA and performed streptavidin pull-down assays (Lansbergen et al., 2006). We found that full-length CP110 indeed associated with full-length CPAP (Fig. 4, A–E). The C-terminal 581–991 region of CP110, which contains a predicted coiled-coil domain (CP110-CC2), was sufficient for the interaction with the full-length CPAP (Fig. 4, A–D). A shorter C-terminal CP110 fragment 581–700 could still bind to CPAP, albeit weaker than the longer fragments (Fig. 4, A and D). Further, we found that the N-terminal part of CPAP mediates the binding to full-length CP110 (Fig. 4 E) and that the CPAP fragment 89–196, including its predicted coiled-coil domain (CPAP-CC1), is sufficient for the association with CP110 581–991 (Fig. 4, B and F).

Next, we analyzed the interaction between N-terminal CPAP and C-terminal CP110 fragments in more detail using biophysical and structural methods. We produced in bacteria a CPAP fragment 89–196 and a CP110 fragment 635–717, which from here onward will be referred to as CPAP-CC1 and CP110-CC2. The oligomerization state of these two domains as well as their

## A

**GMPCPP seeds** soluble tubulin

Soluble tubulin: 0 µM / 3 µM

10mins

5 µm

## B

**0 µM tubulin, free seed plus-ends**

**0 µM Tubulin + CEP97^CP110-GFP, capped seed plus-ends**

50 nm

## C

**GMPCPP seeds**

| | 0 µM tubulin | 3 µM tubulin |
|---|---|---|

Average PF length per MT (nm)

** ns ns

| n: | 24 | 13 | 4 | 18 | 12 | 19 | 18 | 7 |
|---|---|---|---|---|---|---|---|---|
| CEP97^CP110: | − | + | + | − | + | − | + | + |
| Capped: | − | + | − | − | − | − | + | − |
| | Plus ends | | | Minus ends | | Plus ends | | |

## D

**GMPCPP seeds**

| | 0 µM tubulin | 3 µM tubulin |
|---|---|---|

Average PF curvature per MT (deg/dimer)

ns ns

| n: | 23 | 12 | 3 | 16 | 11 | 17 | 17 | 6 |
|---|---|---|---|---|---|---|---|---|
| CEP97^CP110: | − | + | + | − | + | − | + | + |
| Capped: | − | + | − | − | − | − | + | − |
| | Plus ends | | | Minus ends | | Plus ends | | |

Figure 3. **CEP97^CP110 forms caps at the plus ends of GMPCPP-stabilized MTs. (A)** Kymographs showing GMPCPP-stabilized seeds (magenta) in absence (top row) or presence (bottom row) of 3 µM soluble tubulin labelled with TMR-tubulin (gray). **(B)** Slices through denoised tomograms containing plus ends of MT seeds in the absence or presence of 80 nM CEP97^CP110-GFP in the absence of free tubulin. **(C and D)** Average PF lengths (C) and average curvatures (D) of PFs with nonzero length for GMPCPP seeds imaged in the presence of 0 or 3 µM soluble tubulin. Shown are average values for each GMPCPP seed (dots), their mean and SD (error bars); **P < 0.01; ns—nonsignificant with one-way ANOVA followed by Tukey's multiple comparison test; and n is the number of GMPCPP seeds analyzed for each data set. Data distribution was assumed to be normal, but this was not formally tested.

combination was analyzed using size-exclusion chromatography coupled with multi-angle light scattering (SEC-MALS). For CPAP-CC1, these experiments revealed a single-elution peak, corresponding to a molecular mass of 13.0 ± 1.8 kDa, consistent with the presence of a monomer (calculated mass of the monomer: 12.5 kDa). In contrast, CP110-CC2 revealed a single-elution peak, corresponding to a molecular mass of 17.5 ± 1.0 kDa, consistent with the formation of a homodimer (calculated mass of the monomer: 10.0 kDa). When the two proteins were mixed in equimolar ratio, a single peak corresponding to a molecular mass of 19.7 ± 1.1 kDa was found, suggesting the formation of a CPAP-CC1/CP110-CC2 heterodimer (Fig. 4 G). Increasing the CPAP-CC1 concentration in the mixture by two- and threefold supported this conclusion (Fig. S4, A and B). These results suggest that two CPAP-CC1 monomers react with one CP110-CC2 dimer to form two stable CPAP-CC1/CP110-CC2 heterodimers (Fig. 4 H).

## CP110 and CPAP interact through an antiparallel coiled coil

Next, we analyzed the structure of CPAP-CC1, CP110-CC2, and CPAP-CC1/CP110-CC2 by circular dichroism (CD) spectroscopy. The far-UV CD spectrum of CPAP-CC1 recorded at 15°C, with the minima at 220 and 205 nm, was characteristic of proteins displaying a mixture of helical and random-coil secondary structure content. In contrast, CP110-CC2 and a 1:1 mixture of CPAP-CC1 and CP110-CC2 (monomer equivalents) revealed CD spectra characteristic of mostly α-helical proteins, with minima at 208 and 222 nm (Fig. 4 I). The stability of the proteins was subsequently tested by thermal-unfolding profiles monitored by CD at 222 nm. CPAP-CC1 revealed a broad, noncooperative-unfolding profile characteristic of a largely unfolded protein, whereas CP110-CC2 and a 1:1 mixture (monomer equivalents) of CPAP-CC1 and CP110-CC2 revealed sigmoidal and cooperative-unfolding profiles characteristic of well-folded, α-helical coiled-coil proteins (Fig. 4 J). These results suggest that CPAP-CC1 is

**Iyer et al.**
Control of microtubule dynamics by centriolar cap

**Journal of Cell Biology** 7 of 29

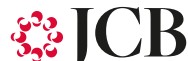

Figure 4. **Characterization of the CPAP–CP110 interaction. (A and B)** Schemes of CP110 and CPAP illustrating the deletion mutants used in this study. "+," interaction between CPAP and CP110; "−," no interaction between CPAP and CP110, and "+/−," weak interaction between CPAP and CP110. For CP110, CC1, and CC2 are the coiled-coil domains. For CPAP, CC1, and CC2 are coiled-coil domains; PN2–3, the tubulin-binding domain (Cormier et al., 2009); MBD, the MT-binding domain; and G-box, glycine-rich C-terminal domain forming an antiparallel β-sheet (Hatzopoulos et al., 2013). **(C and D)** Streptavidin pull-down assays with BioGFP-CP110 truncations as bait and full-length GFP-CPAP as prey. **(E and F)** Streptavidin pull-down assays with BioGFP-CPAP truncations as bait and full-length GFP-CP110 (E) or GFP-CP110 (581–991) (F) as prey. The assays in C–F were performed with extracts of HEK293T cells co-expressing the indicated

constructs and BirA and analyzed by western blotting with anti-GFP antibodies. **(G)** SEC-MALS analysis of CPAP-CC1 (magenta line), CP110-CC2 (green line), and an equimolar mixture of CPAP-CC1 and CP110-CC2 (black line). **(H)** Scheme illustrating the mechanism for CPAP-CC1 and CP110-CC2 association. **(I and J)** CD spectra (I) recorded at 15°C and thermal-unfolding profiles (J) recorded by CD at 222 nm. Proteins and colors as in G. **(K and L)** SAXS analysis of the CPAP-CC1/CP110-CC2 heterodimer. **(K)** Solution X-ray scattering intensity over scattering angle from a 1:1 mixture (monomer equivalents) of CPAP-CC1 and CP110-CC2. The fit to the data yielding the interatomic distance distribution is shown with a black line. **(L)** Surface representation of the X-ray scattering volume of CPAP-CC1/CP110-CC2, at 32 ± 3 Å estimated precision, derived from averaging 22 particle models calculated by ab initio fit to the scattering data. Source data are available for this figure: SourceData F4.

largely unfolded while CP110-CC2 and a mixture of CPAP-CC1 and CP110-CC2 form α-helical coiled-coil structures.

To assess whether the CPAP-CC1/CP110-CC2 complex forms a canonical, extended coiled coil and to further probe the dimerization of CP110-CC2, we performed SEC coupled with small angle X-ray scattering (SEC-SAXS) experiments. The obtained SAXS data were consistent with the presence of a monodisperse species in solution (Fig. 4 K and Fig. S4 C) with a radius of gyration of 3.5 nm as estimated by Guinier approximation (Fig. S4 E). To gain insight into the overall shape of CPAP-CC1/CP110-CC2 and CP110-CC2 in solution, we derived the pairwise distance distribution function, P(r), of these molecules. This suggested the presence of elongated particles in both cases, with a maximum dimension (interatomic distance) of ~12.5 nm (Fig. S4 E). This value is consistent with the calculated length of ~12.0 nm for a two-stranded α-helical coiled coil comprising ~80 amino acids. Accordingly, ab initio SAXS models derived from the P(r) distribution were consistent with the formation of extended coiled coils by CPAP-CC1/CP110-CC2 and CP110-CC2 (Fig. 4 L and Fig. S4 D).

To assess the orientation of the two chains in the CPAP-CC1/CP110-CC2 coiled-coil heterodimer, we performed chemical crosslinking combined with mass spectrometry, using a "zero-length crosslinking agent" (see Materials and methods for details). We found 28 inter-protein crosslinks between CPAP-CC1 and CP110-CC2. By normalizing the intensities of the interlinks to the intralinks and ranking them accordingly (Walzthoeni et al., 2015), we selected the nine most abundant interlinks, which represent the tightest crosslink-derived restraints (Fig. 5, A–C). Together with our CD results, the crosslinking data suggested that CPAP-CC1 and CP110-CC2 form an antiparallel coiled-coil structure (Fig. 5, B and C).

### Design of mutations that disrupt CPAP-CC1/CP110-CC2 coiled-coil formation

To perturb the interaction between CPAP and CP110, we mutated several conserved residues occupying either the predicted heptad a and d core positions and/or the e and g flanking positions of the coiled-coil regions (Fig. 5 C). We found that simultaneous mutation of L149 and K150 at the heptad positions d and e of the second heptad repeat of CPAP-CC1 to alanines (CPAP-CC1 L149A/K150A) disrupted CPAP-CC1/CP110-CC2 heterodimer formation; notably, K150 was among the most intensely crosslinked residues identified in our crosslinking experiments (Fig. 5 C). SEC-MALS of CPAP-CC1 L149A/K150A yielded an elution peak corresponding to a molecular mass of 12.5 ± 0.5 kDa, similar to WT CPAP-CC1 (Fig. 5 D and Fig. S4 E). Analysis of a 1:1 mixture of CPAP-CC1 L149A/K150A and CP110-CC2 (monomer equivalents) revealed two elution peaks, which

corresponded to molecular masses of 16.7 ± 0.4 kDa (CP110-CC2 homodimer) and 13.0 ± 0.5 kDa (CPAP-CC1 L149A/K150A monomer), respectively (Fig. 5 D). Finally, immunoprecipitation from HEK293T cells showed that the double L149A/K150A mutation in CPAP N terminus prevented its coprecipitation with CEP97^CP110 (Fig. 5 E, see Fig. 6 A for construct details).

We further found that mutating R656 and L659 at the heptad positions a and d of the second heptad repeat of CP110-CC2 to alanines (CP110-CC2 R656A/L659A) disrupts both CP110-CC2 homodimer as well as CPAP-CC1/CP110-CC2 heterodimer formation. Analytical SEC of CP110-CC2 R656A/L659A yielded a single-elution peak, which corresponded to the elution of a monomeric protein (Fig. 5 F). Consistent with this finding, CD experiments with CP110-CC2 R656A/L659A revealed a spectrum with minima at around 220 and 205 nm and a broad, noncooperative-unfolding profile (Fig. S4, F and G). A subsequent analytical SEC of a 1:1 mixture of CPAP-CC1 and CP110-CC2 R656A/L659A (monomer equivalents) revealed two elution peaks corresponding to monomers of CPAP-CC1 and CP110-CC2 R656A/L659A, respectively (Fig. 5 G). Taken together, these results demonstrate that residues at key positions of the heptad repeats of both CPAP-CC1 and CP110-CC2 coiled-coil domains are critical for mediating CP110-CC2 homo- and CPAP-CC1/CP110-CC2 heterodimer formation.

### The interaction between CP110 and CPAP promotes slow and processive MT growth in vitro

Having devised a way to perturb the interaction between CP110 and CPAP, we set out to test its functional significance. Since our previous work has shown that full-length CPAP does not behave well in vitro (Sharma et al., 2016), we fused the N-terminal 1–607 fragment of CPAP to a dimer-forming leucine zipper of GCN4 and an mCherry tag (Fig. 6 A; and Fig. S1, F and G). The resulting construct, termed CPAP-N$_{WT}$-mCh, encompassed the tubulin and the MT-binding domains of CPAP called PN2–3 and MBD, respectively, which were also a part of the previously used CPAP$_{mini}$ (Sharma et al., 2016) and also included the CP110-binding CC1 domain, which was absent in CPAP$_{mini}$. We have also generated a similar fusion bearing the L149A/K150A mutations called CPAP-N$_{MUT}$-mCh (Fig. 6 A and Fig. S1 F). Both the WT and mutant CPAP-N versions tracked growing MT plus ends, displayed similar accumulation at the MT tips, reduced MT plus-end growth rate, and inhibited catastrophes (Fig. 6, B and C), similarly to the previously described CPAP$_{mini}$ (Sharma et al., 2016).

Next, we combined 20 nM CEP97^CP110-GFP with 50 nM CPAP-N$_{WT}$-mCh or 50 nM CPAP-N$_{MUT}$-mCh (Fig. 6, D–H). At this concentration, CEP97^CP110 on its own blocked MT outgrowth from approximately half of the seeds and transiently

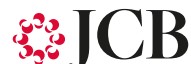

Figure 5. **Characterization of the mutations disrupting CP110–CPAP interaction. (A)** Schematic representation of the domain organization of full-length human CPAP and CP110. The minimal regions CPAP and CP110 that interact with each other are indicated. The domain nomenclature is as in Fig. 4, A and B. **(B and C)** Chemical crosslinking followed by mass spectrometry of CPAP-CC1/CP110-CC2. **(B)** Schematic representations of parallel (left) and antiparallel (right) arrangements of CPAP-CC1 and CP110-CC2 chains in the CPAP-CC1/CP110-CC2 heterodimer. Predicted heptad repeats or H are indicated in each chain. Observed inter-protein crosslinks between residues of CPAP-CC1 and CP110-CC2 are indicated by thin lines. **(C)** Normalized inter-protein crosslinks observed between CPAP-CC1 and CP110-CC2 in the CPAP-CC1/CP110-CC2 heterodimer. The heptad a and d position residues are shown in bold and are underlined. The CPAP-CC1 and CP110-CC2 residues that were mutated in this study are highlighted with asterisks. **(D)** SEC-MALS analysis of CPAP-CC1 L149A/K150A (magenta dashed lines), CP110-CC2 (green solid lines), and an equimolar mixture of CPAP-CC1 L149A/K150A and CP110-CC2 (black solid lines). **(E)** Co-immunoprecipitation of CEP97^CP110-GFP as bait and CPAP-N-mCh WT or mutant as prey from HEK293T cells using anti-GFP antibodies. **(F and G)** Analytical SEC analysis of CPAP-CC1 and CP110-CC2 variants. **(F)** Analytical SEC analysis of CP110-CC2 (green solid line) and CP110-CC2 R656A/L659A (dark green–dashed line). **(G)** Analytical SEC analysis of CPAP-CC1 (magenta line), CP110-CC2 R656A/L659A (dark green–dashed line), and an equimolar mixture of CPAP-CC1/CP110-CC2 R656A/L659A (black solid line). Source data are available for this figure: SourceData F5.

paused ~25% of the remaining MTs (Fig. 1 F), and MTs either grew with normal rates or did not grow at all, dependent on the presence of CEP97^CP110-GFP (Fig. 6, G and H). CPAP-N$_{WT}$ increased the percentage of time CEP97^CP110 was present at MT plus ends from 46% to 69%, although the increase was not statistically significant (Fig. 6 F). Remarkably, CPAP-N$_{WT}$ alleviated the growth block imposed by CEP97^CP110 and instead led to slow and persistent MT elongation with rates ranging between 0.05 and 0.1 µm/min, 20–40 times slower than those of control MTs in the same conditions (Fig. 6, D, G, and H; and Video 3). We

note that the rate of 0.05 µm/min is the lowest value we can detect in our current experimental conditions, and it is possible that MTs we regard as pausing are elongating at an even slower rate. When CEP97^CP110-GFP was not bound to MT plus ends, MTs mostly grew at a rate of 0.5–1.0 µm/min that is characteristic of CPAP-N$_{WT}$ alone (Fig. 6, D, G, and H). In contrast, the addition of CPAP-N$_{MUT}$-mCh decreased the percentage of time CEP97^CP110-GFP was bound to MTs plus ends to 26%, which was significantly less than with CPAP-N$_{WT}$-mCh (Fig. 6, E and F). Episodes of blocking at the seed, pausing, or very slow

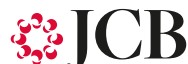

Figure 6. **The interaction between CPAP and CP110 promotes extremely slow and processive MT growth in vitro. (A)** Scheme illustrating the domains of full-length human CPAP and the shorter versions called CPAP-N$_{WT}$-mCh and CPAP-N$_{MUT}$-mCh. CPAP-N$_{WT}$ includes the first 607 amino acids of CPAP followed by a leucine zipper for dimerization and a mCherry (mCh) fluorescent tag. PN2–3, tubulin-binding domain; MBD, MT-binding domain. Two substitution mutations L149A and K150A in CPAP-N$_{MUT}$ disrupt the interaction with CP110. **(B)** Still images and kymographs illustrating MT dynamics in the presence of CPAP-N$_{WT}$-mCh and CPAP-N$_{MUT}$-mCh; arrowheads indicate CPAP on the plus ends of dynamic MTs (gray). **(C)** Top plot: Mean ± SEM of intensity of the

CPAP-N$_{WT}$-mCh ($n$ = 15 MTs) and CPAP-N$_{MUT}$-mCh ($n$ = 14 MTs) present on the MT plus end. Bottom plot: Catastrophe frequencies for CPAP-N$_{WT}$-mCh ($n$ = 151 MTs) and CPAP-N$_{MUT}$-mCh ($n$ = 129 MTs); nonsignificant (ns); P > 0.05, Mann–Whitney U test. **(D)** Kymographs showing slow and processive growth of MT plus end (gray) in presence of both CPAP-N$_{WT}$-mCh (magenta, white open arrowhead indicates plus end accumulation of CPAP-N$_{WT}$-mCh) and CEP97^CP110-GFP (green, white arrowhead indicates plus end accumulation of CEP97^CP110-GFP). CPAP-N$_{WT}$ is not visible as it is in the same channel as the bright GMPCPP-stabilized seed (magenta). **(E)** Kymographs representing MTs (gray) growing from the GMPCPP seed (magenta) in presence of CPAP-N$_{MUT}$-mCh (magenta, white open arrowhead shows plus end localization) and CEP97^CP110-GFP (green, binding event indicated by a white arrowhead). **(F)** Bar plot with mean-SEM of the percentage of total time that CEP97^CP110 is either bound (green bars) or unbound (gray bars) to the MT plus end with CEP97^CP110 alone ($n$ = 3 independent assays) or in combination with CPAP-N$_{WT}$-mCh ($n$ = 4 independent assays) or CPAP-N$_{MUT}$-mCh ($n$ = 3 independent assays). Nonsignificant (ns), P = 0.073, 0.156; **P = 0.0051, one-way ANOVA with Holm–Šídák's multiple comparisons test. Normality tested using Shapiro–Wilk test; P = 0.88. **(G)** MT plus-end growth rates in the indicated conditions. Upper panel, mean + SD; bottom panel, a cumulative histogram showing % of the total time spent by MT plus end growing at different growth rates, with X axis in a log scale. $n$, number of growth events analyzed, is indicated in the figure. Nonsignificant (ns), P = 0.108; ***P = 0.0004; ****P < 0.0001, Kruskal–Wallis ANOVA with Dunn's test for multiple comparisons. **(H)** MT plus-end growth rates for the samples 20 nM CEP97^CP110 alone, or in combination with 50 nM CPAP-N$_{WT}$ or CPAP-N$_{MUT}$, also shown in panel G, but with the values for the events where CEP97^CP110 is present at the tip shown in green and the events where it is absent in gray. The bottom part of the plot shows magnified view for the growth rate values between 0 and 0.4 µm/min; $n$, number of growth events analyzed, is indicated in the figure.

polymerization were still observed when CPAP-N$_{MUT}$-mCh and CEP97^CP110-GFP were combined but were less frequent (Fig. 6, E, G, and H). Most MTs grew with the rate characteristic of CPAP-N$_{MUT}$ alone, because CEP97^CP110 was absent from the MT tip (Fig. 6, E–H). These data indicate that the direct interaction between CP110 and CPAP promotes their colocalization at the MT tip and slow persistent MT growth.

Next, we used FRAP to investigate protein turnover at these slowly growing MT ends. CPAP-N$_{WT}$-mCh alone recovered at growing MT tips very rapidly (Fig. 7, A and B), indicating that it does not move processively with the growing MT end. This result is consistent with previous measurements of the dwell time of ~1.7 s for single CPAP$_{mini}$ molecules (Sharma et al., 2016). When CEP97^CP110-GFP and CPAP-N$_{WT}$-mCh colocalized at slowly growing or pausing ends, CEP97^CP110 did not exchange at all (Fig. 7, A and B). Also the turnover of CPAP-N$_{WT}$ was inhibited, though some exchange was still observed (Fig. 7, A and B), possibly because some PFs were free and could bind CPAP-N$_{WT}$ independently of its interaction with CEP97^CP110. These data indicate that when the complex of CPAP-N$_{WT}$ and CEP97^CP110 imparts slow processive growth, CEP97^CP110 tracks MT plus ends processively.

The shift from blocked MT ends with CEP97^CP110 alone to slowly growing ones when CPAP-N$_{WT}$ was included in the assay suggests that CPAP can act as polymerase that overcomes MT growth inhibition imposed by CP110. To investigate whether CPAP can relieve growth inhibition of a MT plus end already blocked by CEP97^CP110, we used flow-in assays. In these assays, MTs were initially incubated with 20 nM of CEP97^CP110-GFP alone, followed by flow in of a mixture of 50 nM CPAP-N$_{WT}$-mCh and 20 nM CEP97^CP110-GFP. MTs that had no CEP97^CP110 at the tip before the flow in rapidly switched from normal to slow growth (Fig. 7, C and D). However, the ends that were already blocked by CEP97^CP110 before flow in remained blocked (Fig. 7, C and D). Together, these data indicate that the CP110–CPAP complex that promotes slow tubulin addition likely forms in solution and not on MT tips already occluded by CP110.

**The interaction between CP110 and CPAP promotes centriole elongation in cells**

To test the importance of CP110–CPAP interaction in cells, we used U2OS cells with the inducible Flp-In T-REx gene expression system (Ward et al., 2011) to generate stable cell lines where the expression of either WT full-length GFP-CPAP or the L149A/K150A mutant could be induced with doxycycline (Fig. 8 A). In these cell lines, we knocked out the endogenous CPAP gene using a CRISPR/Cas9 approach with two single gRNAs (sgRNAs)-targeting sites located in the introns surrounding exon 3, while maintaining the expression of CPAP transgenes by culturing cells in the presence of doxycycline. CPAP knockout clones, obtained in the WT or mutant GFP-CPAP background, were screened for the deletion in the CPAP-encoding gene by sequencing (Fig. S5 A). Western blotting showed that even in the absence of doxycycline, all clones exhibited leaky expression of full-length GFP-CPAP that slightly exceeded endogenous CPAP levels (Fig. 8 B; and Fig. S5, B and C). We selected two pairs of clones, CPAP-FL$_{WT#3}$ and CPAP-FL$_{MUT#1}$ and CPAP-FL$_{WT#4}$ and CPAP-FL$_{MUT#5}$, with very similar levels of leaky expression of full-length GFP-CPAP, which exceeded endogenous CPAP levels by ~1.5-fold; an additional mutant clone (CPAP-FL$_{MUT#4}$) with approximately twofold higher overexpression level was also included in the analysis (Fig. 8 B).

We next analyzed centriole morphology and length in these clones as well as the host cell line stained with antibodies against acetylated tubulin, CP110 and GFP using Airyscan 2 confocal microscopy (Fig. 8 C). We found that CP110 signal at the distal centriole end was present in all analyzed cell lines (Fig. 8 C), indicating that CP110 localization is not dependent on its interaction with CPAP. Centriolar GFP-CPAP signal was observed in all transgenic lines, but the resolution was insufficient to determine protein localization. The length of the mother centriole in control U2OS cells and the host U2OS cells, blocked in G1/S with thymidine, was ~468 and ~475 nm, respectively, in line with previous work (Kong et al., 2020) (Fig. 8, C and D). Centrioles in the clone CPAP-FL$_{WT#4}$ were on average slightly longer (median of ~527 nm), and in both clones expressing WT GFP-CPAP overly long, irregularly shaped centrioles extending beyond 1 µm were observed in ~10% cells. These data confirm previous observations that elevated CPAP levels cause centriole overelongation (Kohlmaier et al., 2009; Schmidt et al., 2009; Tang et al., 2009). In contrast, in cells expressing full-length GFP-CPAP with L149A/K150A mutations, centrioles in thymidine-arrested cells were on average shorter (median of 309 nm for CPAP-FL$_{MUT#1}$, 325 nm for CPAP-FL$_{MUT#5}$, and 349

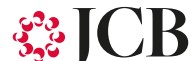

**Figure 7. Dynamics of CEP97^CP110 and CPAP on slowly growing MT plus ends. (A)** Upper panel: Kymographs representing bleaching of CPAP-$N_{WT}$-mCh (magenta) with a 561-nm laser and its quick recovery. Inset shows the bleaching moment with a white arrow. Bottom panel: Kymographs showing bleaching of both CEP97^CP110-GFP (green) and CPAP-$N_{WT}$-mCh (magenta) with a 488-nm laser (white arrow) illustrating that CEP97^CP110-GFP does not recover and CPAP-$N_{WT}$-mCh recovers slowly. **(B)** Upper plot: Mean + SD of the normalized intensity of CEP97^CP110-GFP (green line, $n = 30$) and CPAP-$N_{WT}$-mCh (magenta line, $n = 36$) over time on slowly growing MT plus ends. Bottom plot: Comparison of the recovery CPAP-$N_{WT}$-mCh alone ($n = 8$) and in the presence of CEP97^CP110-GFP (the latter data are the same as in the lot above). Black arrow marks the time point of photobleaching. **(C)** Scheme of flow-in assays. **(D)** Top: Kymographs representing a MT that was dynamic before flow in and switched to slow growth after the flow in of CEP97^CP110-GFP (green, white arrowhead) and CPAP-$N_{WT}$-mCh (magenta, white open arrowhead). Bottom: Kymographs showing a MT plus end blocked by CEP97^CP110-GFP (white arrowhead) before and after the flow in. The moment of flow in is indicated by a black arrow in both kymographs.

nm for CPAP-FL$_{MUT\#4}$), and centriole overgrowth was rare, ~0.3% in CPAP-FL$_{MUT\#3}$ and ~1% in CPAP-FL$_{MUT\#4}$, even though GFP-CPAP expression level in these clones was 1.5 and 2 times higher than endogenous CPAP levels (Fig. 8, B–D). In cells

arrested in G2/M with RO-3306, the mother centrioles were slightly shorter in cells expressing CPAP-FL$_{MUT}$, and the daughter centrioles were ~1.8 times shorter than in control and CPAP-FL$_{WT}$–expressing cells (Fig. 8, E and F). However, we did

**A** U2OS (**Control**) → Flp-In T-REx U2OS (**Host**) → Flippase mediated FRT site-specific integration → Transgenic cell lines → Endogenous CPAP KO

Single FRT site integration

Transfection

Expression of GFP -CPAP-FL_WT or GFP-CPAP-FL_MUT under CMV/tetO2 regulation

SV40 ATG FRT Zeocin

**Genome integration**
- CMV | TetR
- SV40 | ATG FRT | Zeocin

**pcDNA5/FRT/TO**
- FRT Hyg/Puro CMV-tetO2 GFP CPAP-FL_WT or MUT
- pOG44 — CMV Flippase

**Genome integration**
- CMV | TetR
- SV40 | ATG FRT | Hyg/Puro CMV/tetO2 GFP CPAP-FL_WT or MUT

**B**

Normalized CPAP intensity (western blot)

■ Cell lines used for quantification
▨ Other cell lines

ns (p2)
ns (p1)  ns
p=0.1  ns
ns

Control, Host, #1, #2, #3, #4, #5 (CPAP-FL_WT), #1, #2, #3, #4, #5 (CPAP-FL_MUT)

**D**

Mother centriole length at G1/S ($\mu$m)

**** **** **** ****
ns

Control, Host, #3, #4 (CPAP-FL_WT), #1, #5, #4 (CPAP-FL_MUT)

**F**

Centriole length at G2/M ($\mu$m)

■ Mother centriole
▨ Daughter centriole

CP110
Ace tub
DC
MC
Centriole length

**** ****
**** ****

Control, Host, #3, #4 (CPAP-FL_WT), #1, #5, #4 (CPAP-FL_MUT)

**C** Thymidine block 24h (G1/S)

Merge | Ace tub | CP110

Control
Host
CPAP-FL_MUT#4  GFP-CPAP

Merge | Ace tub | CP110 | GFP-CPAP

CPAP-FL_WT#3
CPAP-FL_MUT#1

Merge | Ace tub | CP110 | GFP-CPAP

CPAP-FL_WT#4
CPAP-FL_MUT#5

1 μm

**E** RO-3306 block 24h (G2/M)

Merge | Ace tub | CP110

Control
Host
CPAP-FL_MUT#4  GFP-CPAP

Merge | Ace tub | CP110 | GFP-CPAP

CPAP-FL_WT#3
CPAP-FL_MUT#1

Merge | Ace tub | CP110 | GFP-CPAP

CPAP-FL_WT#4
CPAP-FL_MUT#5

1 μm

Figure 8. **Characterization of the effects of disrupting CPAP–CP110 interaction on centriole length regulation at interphase. (A)** Scheme showing the generation of the inducible transgenic cell lines expressing either GFP-tagged WT full-length CPAP (CPAP-FL_WT) or full-length CPAP with L149A/K150A

mutation (CPAP-FL$_{MUT}$). U2OS cells (Control) were used to integrate with the Tet repressor, a single FRT site, and the lacZ-Zeocin fusion gene by lentivirus to generate the Flp-In T-REx U2OS host cell line (Host). pcDNA5/FRT/TO vectors for doxycycline-inducible expression of GFP-CPAP-FL$_{WT}$ or GFP-CPAP-FL$_{MUT}$ were co-transfected together with Flp recombinase-encoding pOG44 vector into the Flp-In T-REx U2OS host cell line to induce their integration into the FRT site of the host cell genome in a Flp recombinase-dependent manner. The expression of GFP-CPAP-FL$_{WT}$ or GFP-CPAP-FL$_{MUT}$ was controlled by the inducible hybrid human cytomegalovirus (CMV)/Tet operator 2 (TetO2) promoter. The endogenous CPAP gene was knocked out using a CRISPR/Cas9–based approach. **(B)** Mean ± SD of the normalized CPAP levels based on western blots shown in Fig. S5 C ($n = 3$ trials). Cell lines used for quantification are shown in magenta, where cell line pairs 1 and 2 (p1 and p2, respectively) are highlighted. Nonsignificant (ns), P > 0.05 calculated using an unpaired two-tailed Mann–Whitney U test. **(C and E)** Immunofluorescence images acquired using Airyscan 2 confocal microscope of centrioles at G1/S (C) and G2/M (E) and stained for acetylated tubulin (blue), CP110 (green), and GFP-CPAP (magenta). **(D)** Median ± IQR of mother centriole length at G1/S measured from proximal end of centriole (determined by acetylated tubulin) to distal end (determined by the geometric center of CP110 signal) (scheme in panel F). n, number of analyzed centrioles: control cell line, $n = 113$; host, $n = 105$; CPAP-FL$_{WT\#3}$, $n = 132$; CPAP-FL$_{WT\#4}$, $n = 131$; CPAP-FL$_{MUT\#1}$, $n = 84$; CPAP-FL$_{MUT\#5}$, $n = 170$; CPAP-FL$_{MUT\#4}$, $n = 81$; nonsignificant (ns); and ****P < 0.001 calculated using Kruskal–Wallis ANOVA test. **(F)** Median ± IQR of centriole length at G2/M measured as in D. n, number of analyzed mother centrioles (MC) and daughter centrioles (DC): control cells, $n = 80$ MC, 75 DC; host, $n = 72$ MC, 59 DC; CPAP-FL$_{WT\#3}$, $n = 67$ MC, 69 DC; CPAP-FL$_{WT\#4}$, $n = 64$ MC, 57 DC; CPAP-FL$_{MUT\#1}$, $n = 71$ MC, 80 DC; CPAP-FL$_{MUT\#5}$, $n = 78$ MC, 79 DC; CPAP-FL$_{MUT\#4}$, $n = 79$ MC, 77 DC; nonsignificant (ns); and ****P < 0.001 calculated using Kruskal–Wallis ANOVA test.

not observe major differences in centriole length in cells that were arrested for 24 h in mitosis using the inhibitor of kinesin-5 KIF11 STLC (Fig. S5, D and E). This indicated that centriole elongation in mitotically arrested cells occurs through a mechanism that does not require the CPAP–CP110 interaction.

We then extended the analysis of centriole morphology in thymidine-blocked CPAP-FL$_{WT\#3}$ and CPAP-FL$_{MUT\#1}$ cells using U-ExM. Compared with control cells, in cells expressing the mutant full-length GFP-CPAP, ∼21% and ∼4% of centrioles were strikingly shorter or incomplete, respectively. However, these phenotypes were very rare in the cells expressing WT full-length GFP-CPAP (Fig. 9, A and B). We observed a robust signal of CP110 at the distal end of mother and daughter centrioles in most studied cells, although the incidence of mother centrioles with low or undetectable CP110 signal was somewhat, albeit not significantly, higher in cells expressing mutant CPAP compared with the WT protein (3.8% compared with 0.4% for the mother and 1.5% compared with 0% for the daughter centriole) (Fig. 9 C). To study the effects on GFP-CPAP localization, we compared both anti-GFP and anti-CPAP antibodies and found that the latter staining was more robust and allowed better detection of CPAP localization after expansion (Fig. 9 D). Both WT and mutant GFP-CPAP were clearly visible around the daughter centriole, and weaker signals associated with the inner proximal centriole and the distal end of the mother centriole were also observed (Fig. 9 E). Even when centriole morphology was strongly perturbed, CPAP signal was still present at both the mother and the daughter centrioles, although the pool of CPAP at the distal end of the mother centriole was reduced or even undetectable in ∼31% and ∼20%, respectively, of the cells expressing mutant CPAP (Fig. 9, F and G). Similar phenotypes were observed when CPAP-FL$_{WT\#4}$ and CPAP-FL$_{MUT\#5}$ cell clones were compared with each other (Fig. S5, F and G).

Together, these results indicate that the CPAP–CP110 interaction is not essential for centriole formation or for localization of CPAP or CP110 to the centriole, but it does promote robust localization of CPAP to the distal centriole end, centriole elongation, as well as CPAP-driven centriole overelongation when CPAP is overexpressed. Disruption of the CPAP–CP110 interaction can cause centriole defects, albeit at low frequency, indicating that it makes centriole biogenesis more robust.

## Discussion

In this study, we have reconstituted in vitro the regulation of MT dynamics by the centriolar proteins CP110, CEP97, and CPAP. We showed that CP110 autonomously recognizes MT plus ends, inhibits their growth, and can induce pausing of dynamic MT ends, which means that it can also suppress plus-end shrinkage. These data provide a biochemical explanation for a large body of cell biological work showing that CP110 binds to distal ends of centrioles, prevents overgrowth and, in some cases, also shortening of centriolar MTs, and needs to be removed when centrioles are repurposed as ciliary basal bodies (Aydogan et al., 2018, 2022; Delgehyr et al., 2012; Dobbelaere et al., 2020; Franz et al., 2013; Goetz et al., 2012; Huang et al., 2018; Kleylein-Sohn et al., 2007; Kohlmaier et al., 2009; Le Guennec et al., 2020; Prosser and Morrison, 2015; Schmidt et al., 2009; Shoda et al., 2021; Spektor et al., 2007). In contrast, we found no evidence that CEP97 binds to MTs directly, suggesting that CEP97 affects centriolar MTs through other centriolar components, for example, by binding and regulating CP110 (Aydogan et al., 2022; Jiang et al., 2012; Sharma et al., 2021; Spektor et al., 2007) and CEP104 (Jiang et al., 2012; Rezabkova et al., 2016). Our results obtained with the CEP97^CP110 chimera are in line with this idea, as this fusion protein was better behaved in vitro than full-length CP110 or its fragments.

The MT-binding domain of CP110 resides in its C-terminal part containing a two-stranded coiled coil that interacts with CPAP, as well as several putative helical and disordered regions, which, based on AlphaFold predictions (Jumper et al., 2021; Varadi et al., 2022), are not expected to form folded protein domain(s). It is possible that this part of CP110 assumes a stable structure only upon binding to MTs. We showed that CP110 stably binds to MT ends even in the presence of a 1,000-fold molar excess of soluble tubulin, suggesting that its binding site is specific for the MT lattice and may be formed by more than one tubulin subunit. Furthermore, CP110 specifically blocks tubulin addition and removal at MT plus ends. Notably, our data argue against a strong competition with two other proteins binding to the plus-end–exposed tip of β-tubulin, DARPin, and CPAP, suggesting that the binding site of CP110 is distinct from these MT tip binders and thus does not fully rely on the longitudinal interface of β-tubulin. Our cryo-ET data indicate that CP110 interacts with the luminal side of the MT plus end. CP110 might

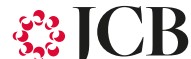

**Figure 9.** **Characterization of morphological defects of centrioles in cells with perturbed CPAP–CP110 interaction using U-ExM. (A and D–F)** U-ExM images of centrioles from host, CPAP-FL$_{WT\#3}$, and CPAP-FL$_{MUT\#1}$ blocked for 24 h in G1/S and stained for acetylated tubulin (blue) combined with CP110

(green) in (A), GFP (magenta) in (D) top image, or CPAP (magenta) in (D) bottom image and (E and F). **(A)** Normal centrioles from host and CPAP-FL$_{WT\#3}$ and incomplete centriole from CPAP-FL$_{MUT\#1}$ (white arrow). **(B)** Mean + SD of centriole morphology at G1/S categorized in normal, short, incomplete, and elongated. For host, $n$ = 159, CPAP-FL$_{WT\#3}$, $n$ = 171, and CPAP-FL$_{MUT\#1}$, $n$ = 251 centrioles; nonsignificant (ns); and *P = 0.019 and ****P < 0.0001 calculated using ordinary one-way ANOVA test. Normality tested using Shapiro–Wilk test; P = 0.77. **(C)** Mean + SD of CP110 cap on mother and daughter centrioles at G1/S categorized visually as present (clear, bright signal), low levels of (barely visible, low signal), and absent (no signal) CP110. For host, $n$ = 101, CPAP-FL$_{WT\#3}$, $n$ = 114, and CPAP-FL$_{MUT\#1}$, $n$ = 160 centrioles; nonsignificant (ns); and calculated using ordinary one-way ANOVA test. Normality tested using Shapiro–Wilk test; P = 0.83. **(D)** Comparison between antibody against GFP (top) or CPAP (bottom) to localize GFP-CPAP (magenta). CPAP is localized to daughter centriole (white arrowhead), inner, proximal (white arrow), and distal cap of mother centriole (black arrowhead). **(E)** Normal centrioles from host and CPAP-FL$_{WT\#3}$ and incomplete centriole from CPAP-FL$_{MUT\#1}$ (white arrow). **(F)** Gallery of short or incomplete centrioles from CPAP-FL$_{MUT\#1}$. CPAP population on daughter centriole was unchanged (white arrowhead), and example nine shows minor defect in centriole shaft (white arrow). **(G)** Mean + SD of CPAP localization on the distal and proximal ends of the mother and daughter centrioles at G1/S, categorized visually as present (clear, bright signal), low levels of (barely visible, low signal), and absent (no signal). For host, $n$ = 140, CPAP-FL$_{WT\#3}$, $n$ = 132, and CPAP-FL$_{MUT\#1}$, $n$ = 179; nonsignificant (ns); and *P = 0.031 or 0.012 and **P = 0.008 calculated using ordinary one-way ANOVA test. Normality tested using Shapiro–Wilk test; P = 0.13. Scale bar is corrected for ~4.5 expansion factor.

bind along individual curved PFs or to the luminal grooves between flared PFs. The specificity of CP110 for the plus ends might be due to the geometry of flared PFs, with β-tubulins separated somewhat further apart than α-tubulins. Such mechanism would be analogous to that of CAMSAPs, which recognize the transition zone between curved PFs and the regular MT lattice at the minus ends (Atherton et al., 2017). Through its binding, CP110 might hinder PF straightening or zippering, and thus block MT elongation, and at the same time, prevent PF peeling and disassembly, and thereby inhibit MT shortening. Binding either along or between MT PFs from the luminal side would be in line with our observation that CP110 decreases PF flaring and keeps MT tips in a relatively constant conformation irrespective of the concentration of soluble tubulin. Flared PFs that appear to be common for dynamic MT plus ends (Gudimchuk et al., 2020; McIntosh et al., 2018) might promote tubulin addition, whereas blunt MT ends might be more difficult to elongate. Alternatively, shorter and less curved PFs at MT plus ends could be a consequence, rather than the cause of a lower tubulin on-rate induced by CP110 binding.

MT growth inhibition imposed by CP110 was alleviated by CPAP, with which it directly interacts. Although by itself CPAP imparts a fourfold reduction in the MT growth rate (Sharma et al., 2016), in complex with CP110, it overcomes MT growth inhibition and allows some, albeit very slow, MT elongation. It is possible that the disruption of homodimerization of the CC2 domain of CP110 by the CC1 domain of CPAP, as suggested by our biochemical experiments (Fig. 10), alters the conformation of CP110, reduces its binding affinity for MTs, and promotes occasional tubulin addition and plus-end tracking. The growth rate observed with CP110 and CPAP together was 20–40 times slower than that observed with tubulin alone in the same conditions. Such slow MT elongation rate would be expected to be insufficient for the formation of a long GTP cap, which is necessary not only to stabilize the growing end but also to promote tubulin addition (Wieczorek et al., 2015). CPAP thus can be regarded as a MT polymerase responsible for tubulin polymerization at MT plus ends, which are unfavorable for tubulin incorporation (Fig. 10). Consistent with this notion, the domain organization of CPAP is reminiscent of another MT polymerase, XMAP215/chTOG (Widlund et al., 2011): CPAP contains a MT-binding domain, which can tether it to the MT wall, and a tubulin-binding domain (PN2–3), which could promote the addition of a tubulin dimer to the MT plus end.

Understanding the interaction mode between CP110 and CPAP allowed us to perturb it. Both in cells and in vitro, the two proteins can bind to MT tips independently of each other, and the interaction between the two proteins is not essential for centriole formation. However, this interaction promotes the localization of CPAP to the distal centriole end in cells and centriole elongation in the S phase. In vitro, the binding between the two proteins leads to long episodes of slow MT growth, whereby CP110 does not exchange, and the turnover of CPAP is inhibited, indicating that together, CPAP and CP110 form a persistent cap that promotes very slow tubulin addition. This observation is consistent with CPAP acting as a polymerase that can overcome MT growth inhibition imposed by CP110, explaining the opposing roles of the two proteins in controlling centriole length in mammalian cells (Kohlmaier et al., 2009; Schmidt et al., 2009; Tang et al., 2009). Perturbation of the CP110–CPAP interaction induced centriole defects in some cells, indicating that it underlies one of the mechanisms controlling robust centriole biogenesis. Since this interaction is not essential, additional mechanisms must exist, and other centriolar proteins likely regulate the localization and activities of CP110 and CPAP. Therefore, although in our in vitro experiments the complex of CPAP and CP110 that promoted slow MT elongation appeared to form in solution and not at the MT tips blocked by CP110, in cells, the binding and turnover of CP110 and CPAP could be regulated differently and are likely influenced by other centriole biogenesis factors.

The CP110-interacting domain of CPAP, CC1, is located N-terminally of the LID domain, which interacts with the longitudinal interface of β-tubulin and points with its N terminus toward MT lumen (Campanacci et al., 2022). Therefore, CPAP-CC1 is expected to be ideally positioned to interact with CP110, bound to the luminal side of the MT plus end (Fig. 10), and together, they can span both MT surfaces. Our recent work showed that a "plug-like" complex, like the one observed for CEP97^CP110 chimera, is also formed at MT plus ends by proteins controlling the elongation of ciliary tips (Saunders et al., 2024, Preprint). Although ciliary tip-associated and centriolar cap complexes consist of completely different proteins, some striking similarities are apparent: in both cases, MT growth-inhibiting proteins and MT polymerases are present, and slow MT polymerization is associated with the reduction in PF flaring. Furthermore, in both cases, protein complexes can potentially span both the outer and the inner MT surface, inhibit

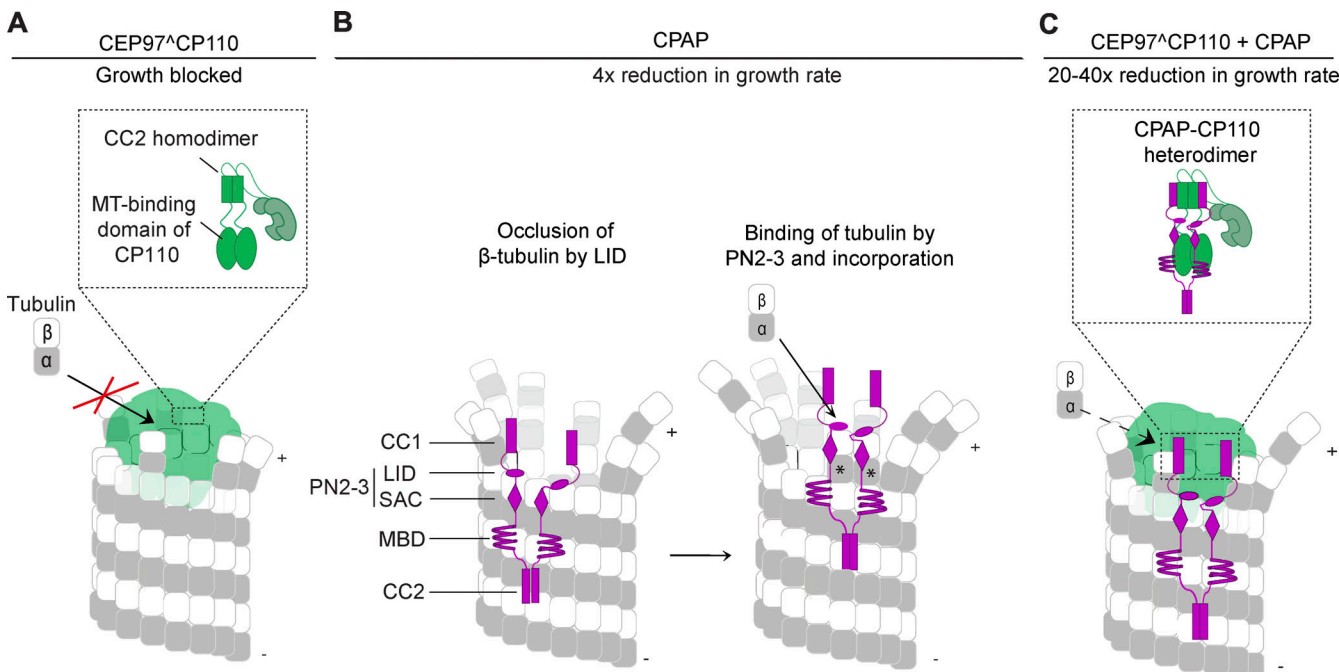

**Figure 10.** **Model for the combined action of CP110 and CPAP at MT plus ends.** A model illustrating the molecular mechanism of slow and processive MT growth observed with CEP97^CP110 and CPAP. **(A)** CEP97^CP110 (green) binds to the luminal side of the MT plus end (gray) through a MT-binding domain present at the C terminus of the CP110 moiety of CEP97^CP110. By doing so, it inhibits incorporation of new tubulin dimers into the MT plus end. **(B)** CPAP binds on the outside of the MT wall with its MT-binding domain (MBD); with the two parts of its tubulin-binding domain PN2–3, SAC, and LID, it binds to the side and longitudinal interface of tubulin at the PF tip, inhibits catastrophes, and leads to a 4× reduction in the plus-end growth rate. New tubulin dimers incorporated into the MT are shown with asterisks. **(C)** CEP97^CP110 and CPAP bind to each other through their coiled-coil domains (CPAP-CC1/CP110-CC2), and CPAP overcomes growth inhibition imposed by CP110. The complex of the two proteins leads to slow and processive growth of the MT plus end. Disruption of CP110 homodimerization by CPAP might contribute to alleviation of CP110-driven MT growth inhibition.

rapid tubulin addition, and at the same time prevent MT depolymerization in the absence of a long stabilizing GTP cap. These features, common for both centriole and ciliary MT tip regulators, help to explain how these stable, slowly growing MT-based structures are formed next to highly dynamic interphase MT arrays and mitotic spindles.

## Materials and methods
### DNA constructs
Human CPAP and CP110 constructs used here were described previously (Jiang et al., 2012; Sharma et al., 2016). To overexpress proteins in HEK293T cells, cDNAs of the human proteins were cloned into pTT5-based expression vectors (#52355; Addgene), which also had a SII and fluorescent tags (GFP or mCherry) to generate SII-GFP-CP110, CEP97-GFP-SII, CEP97^CP110-GFP-SII, CPAP-N$_{WT}$-mCherry-SII, and CPAP-N$_{MUT}$-mCherry-SII.

To mutate two amino acids (L149A, K150A) in the CPAP protein, two pairs of PCR primers bearing the desired mutations were designed to induce them into the CPAP cDNA by PCR. Then the two purified CPAP cDNA fragments with overlapping regions were cloned by Gibson Assembly (NEB) into the linear pcDNA5-FRT-TO vector (V652020; Thermo Fisher Scientific) digested by FastDigest BamHI and NotI (Thermo Fisher Scientific). The primers used for PCR are listed in the Table 1.

To knock out the endogenous CPAP, two sgRNAs were selected to target the flanking intron regions of exon 3. To generate the PX459 construct with sgRNA sequences, the vector pSpCas9(BB)–2A-Puro (PX459) V2.0 (Ran et al., 2013), purchased from Addgene, was digested with FastDigest BbsI (Thermo Fisher Scientific), and the annealing product of single-strand sgRNA-encoding oligonucleotides was inserted into the linear PX459 vector by T4 ligation (Thermo Fisher Scientific). To generate the PX459 with a blasticidin antibiotic cassette (PX459-BSD), the PX459 construct was digested with FastDigest EcoRI (Thermo Fisher Scientific) to cut out the puromycin resistance cassette, and the purified PCR product of the blasticidin fragment was inserted into the linear PX459 vector by Gibson Assembly (NEB). To change the restriction endonuclease site downstream of hU6 promoter, thereby avoiding an additional BbsI site due to the insertion of blasticidin cassette, BbsI (Thermo Fisher Scientific) was used to cut the PX459-BSD, and then the annealing product of oligonucleotides bearing two head-to-head aligned BsaI restriction endonuclease sites was inserted into the linear PX459-BSD vector by ligation with T4 ligase (Thermo Fisher Scientific). The sgRNA sequences were sgRNA-targeting CPAP intron 2 #1 5′-GTGGCTGATTTAGTT ACCTAG-3′ and sgRNA-targeting CPAP intron 3 #2 5′-GGT TCTTCAGCTGAACAGAG-3′. The sequencing primers for genotyping were CPAP KO sequencing primer F 5′-AAAAAG

Table 1.  **Key resources table**

| Reagent or resource | Source | Identifier |
| --- | --- | --- |
| **Antibodies** | | |
| Mouse anti-acetylated tubulin | Sigma-Aldrich | Cat# T7451, RRID:AB_609894 |
| Rabbit anti-CP110 | Proteintech | Cat# 12780-1-AP, RRID:AB_10638480 |
| Chicken anti-GFP | Aves Lab | Cat# GFP-100, RRID:AB_2307313 |
| Rabbit anti-CPAP | Proteintech | Cat# 11517-1-AP, RRID:AB_2244605 |
| Rabbit anti-CPAP | Kohlmaier et al. (2009) | |
| Mouse anti-GFP | Sigma-Aldrich | Cat# G1546 |
| Rabbit anti-RFP | Rockland Immunochemicals | Cat# 00–401-379 |
| Goat anti-mouse Alexa 594 | Thermo Fisher Scientific | Cat# A-11032, RRID:AB_2534091 |
| Goat anti-rabbit Atto 647N | Sigma-Aldrich | Cat# 40839, RRID:AB_1137669 |
| Goat anti-chicken dylight 488 | Thermo Fisher Scientific | Cat# SA5–10070, RRID:AB_2556650 |
| Mouse anti-Ku80 (clone 7) | BD Biosciences | Cat# 611360, RRID:AB_398882 |
| Goat anti-mouse IgG2a IRDye 680LT | LI-COR Biosciences | Cat#926–68051 |
| Goat anti-rabbit IgG IRDye 800 CW | LI-COR Biosciences | Cat#926–3221 |
| **Chemicals, peptides, and recombinant proteins** | | |
| Thymidine | Sigma-Aldrich | Cat # T9250-25G |
| RO-3306 | Selleckchem | Cat# S7747 |
| S-trityl-L-cysteine (STLC) | Sigma-Aldrich | Cat# 164739-5G |
| Puromycin | Invivogen | Cat# ant-pr-1 |
| Blasticidin | Invivogen | Cat# ant-bl-1 |
| StrepTactin sepharose high performance | GE Healthcare | Cat# 28–9355-99 |
| ChromoTek GFP-Trap magnetic particles M-270 | ProteinTech | Cat# gtma |
| Polyethyleneimine hydroschloride (PEI MAX) | Polysciences | Cat# 24765-1 |
| cOmplete, EDTA-free protease inhibitor cocktail | Roche | Cat# 4693116001 |
| PhosSTOP | Roche/Sigma-Aldrich | Cat# 4906845001 |
| Tubulin protein porcine brain | Cytoskeleten | Cat# T240-C |
| Tubulin protein (rhodamine) porcine brain | Cytoskeleten | Cat# TL590M-B |
| Tubulin protein (HiLyte 488 ) porcine brain | Cytoskeleten | Cat# TL488M-B |
| Tubulin protein (HiLyte 647 ) porcine brain | Cytoskeleten | Cat# TL670M-B |
| Tubulin protein biotin porcin brain | Cytoskeleten | Cat# T333P-B |
| GMPCPP | Jena Bioscience | Cat# NU-405L |
| GTP | Sigma-Aldrich | Cat# G8877 |
| Glucose oxidase | Sigma-Aldrich | Cat# G7141 |
| Catalase | Sigma-Aldrich | Cat# C9322 |
| DTT | Sigma-Aldrich | Cat# R0861 |
| K-casein | Sigma-Aldrich | Cat# C0406 |
| Neutravidin | Invitrogen | Cat# A-2666 |
| GFP-EB3 full length | Montenegro Gouveia et al. (2010) | N/A |
| DARPin (TM-3)$_2$ | Ahmad et al. (2016), Pecqueur et al. (2012) | N/A |
| HisTrap FF 5 ml | GE Healthcare | Cat #17531901 |

Table 1.  **Key resources table (Continued)**

| Reagent or resource | Source | Identifier |
|---|---|---|
| Superdex 75 10/30 GL | Cytiva | Cat #17517401 |
| HiLoad 16/60 superdex 75 pg column | GE Healthcare | Cat #28989333 |
| Superdex 200 increase 10/300 GL | GE Healthcare | Cat #28990944 |
| **Cell lines** | | |
| Human embryonic kidney 239T (HEK293T) cells | ATCC | Cat #CRL-321 |
| Flp-In T-Rex U2OS cells | Sharma et al. (2016) | |
| Flp-In U2OS GFP-CPAP-FL$_{WT}$ cells | This study | N/A |
| Flp-In U2OS GFP-CPAP-FL$_{MUT}$ cells | This study | N/A |
| Flp-In U2OS GFP-CPAP-FL$_{WT}$ cells-CPAP KO | This study | N/A |
| Flp-In U2OS GFP-CPAP-FL$_{MUT}$ cells-CPAP KO | This study | N/A |
| **Oligonucleotides** | | |
| sgRNA-targeting CPAP intron 2 #1 | This study | 5'-GTGGCTGATTTAGTTACCTAG-3' |
| sgRNA-targeting CPAP intron 3 #2 | This study | 5'-GGTTCTTCAGCTGAACAGAG-3' |
| CPAP KO-sequencing primer F | This study | 5'-AAAAAGGGACCACAGGTAGCG-3' |
| CPAP KO-sequencing primer R | This study | 5'-GCTCCAGGTCACATTTCCAGT-3' |
| Flp-In siCPA-presistant left part F | This study | 5'-ACTTAAGCTTGGTACCGAGCTCGGCCACCATGGTGAGCAAG-3' |
| Flp-In siCPA-presistant left part R | This study | 5'-CTCTTCGATCCTAGCTAACTCTTTTGCTTTCTGTTGTTCGAAGT-3' |
| Flp-In siCPA-presistant right part F | This study | 5'-AAAAGAGTTAGCTAGGATCGAAGAGTTTAAAAAGGAGGAGATGAGGA-3' |
| Flp-In siCPA-presistant right part R | This study | 5'-TTAAACGGGCCCTCTAGACTCGAGCTCACAGCTCCGTGTCCATTAGC-3' |
| BBNF-mut-CPAP-149–150 | This study | 5'-AAAAACTTGAACAGGCCGCCGAAGTACAACAGAAGAAGCAGGAACAATTGAA-3', |
| BBNR-mut-CPAP-149–150 | This study | 5'-ATGGGCAGGTAGCCGGATCAAGCGTATGCAGCCGCCGCAT-3' |
| INSF-mut-CPAP-149–-150 | This study | 5'-ATGCGGCGGCTGCATACGCTTGATCCGGCTACCTGCCCAT-3', |
| INSR-mut-CPAP-149–150 | This study | 5'-TGCTTCTTCTGTTGTACTTCGGCGGCCTGTTCAAGTTTTT-3' |
| PX459v2-P2A-BSD F | This study | 5'-GGCCAGGCAAAAAGAAAAAGGAATTCGGCAGTGGAGCTACTAACTTCAGCCTGCTGAAG-3' |
| PX459v2-P2A-BSD R | This study | 5'-GCTGATCAGCGAGCTCTAGTTAGAATTCTCAGCCCTCCCACACATAACC-3' |
| 2bsai-BSD F | This study | 5'-CACCTGAGACCGAGGTCTCT-3' |
| 2bsai-BSD R | This study | 5'-AAACAGAGACCTCGGTCTCA-3' |
| **Recombinant DNA** | | |
| pcDNA5 FRT TO | Thermo Fisher Scientific | Cat# V652020 |
| pcDNA5-FRT-TO-GFP-CPAP-FL$_{WT}$ (plasmid) | This study | N/A |
| pcDNA5-FRT-TO-GFP-CPAP-FL$_{MUT}$ (plasmid) | This study | N/A |
| pOG44 Flp-recombinase expression vector | Thermo Fisher Scientific | Cat#: V600520 |
| PX459-sgRNA–targeting CPAP intron 2 (plasmid) | This study | N/A |
| PX459-sgRNA targeting CPAP intron 3 (plasmid) | This study | N/A |
| pSpCas9(BB)–2A-Puro (PX459) V2.0 | Ran et al. (2013) | Addgene, RRID: Addgene_62988 |
| pSpCas9(BB)–2A-BSD (PX459-BSD) | This study | N/A |
| pTT5-SII-GFP-CP110 | This study | N/A |
| pTT5-CEP97-GFP-SII | This study | N/A |
| pTT5-CEP97(1-650)^CP110(581-991)-GFP-SII | This study | N/A |
| pTT5-CPAP(1-607)$_{WT}$-mCherry-SII | This study | N/A |
| pTT5-CPAP(1-607)L149A/K150A-mCherry-SII | This study | N/A |

Table 1. **Key resources table (Continued)**

| Reagent or resource | Source | Identifier |
|---|---|---|
| PSTCm9 | Olieric et al. (2010) | N/A |
| **Software and algorithms** | | |
| FIJI (ImageJ) | National Institutes of Health | https://fiji.sc/; https://imagej.net/ij/index.html; RRID: SCR_003070 |
| GraphPad prism 9.0.0 (121) | GraphPad Software | RRID: SCR_000306 |
| Adobe Illustrator 2023 | Adobe Systems | RRID: SCR_010279 |
| Huygens professional 21.04 | Scientific Volume Imaging B.V. | https://svi.nl |
| Metamorph | Molecular Devices | https://www.moleculardevices.com/products/cellular-imaging-systems/high-content-analysis/metamorph-microscopy |
| KymoResliceWide plugin | Eugene Katrukha | https://github.com/ekatrukha/KymoResliceWide |
| MATLAB code | Eugene Katrukha | https://gist.github.com/ekatrukha/8a8f336d44b7bef6523fc1acc3c71a19 |
| UCSF chimera | Pettersen et al. (2004) | https://www.cgl.ucsf.edu/chimera/ |
| Situs 3 | Kovacs et al. (2018) | |
| DAMAVER | Volkov and Svergun (2003) | https://www.embl-hamburg.de/biosaxs/damaver.html |
| DAMMIF | Franke and Svergun (2009) | https://www.embl-hamburg.de/biosaxs/dammif.html |
| Scatter3 | Forster et al. (2010) | |
| PyMol | Schrödinger | https://pymol.org/ |

GGACCACAGGTAGCG-3′ and CPAP KO sequencing primer R 5′-GCTCCAGGTCACATTTCCAGT-3′.

## Cell lines and cell culture

HEK293T cells (ATCC) and the Flp-In T-REx U2OS host cells (Sharma et al., 2016) were cultured in DMEM and Ham's F10 (1:1) supplemented with 10% FCS and 5 U/ml penicillin and 50 µg/ml streptomycin. The cell lines were routinely checked for mycoplasma contamination using the LT07–518 Mycoalert assay. Polyethyleneimine (PEI, Polysciences) was used to transfect HEK293T cells for StrepTactin protein purification and pull-down experiments.

FuGENE 6 (Promega) was used to transfect cells with plasmids for generating inducible cell lines and stable knockouts. For transfecting inducible cell lines, pcDNA5-FRT-TO-GFP-CPAP-FL$_{WT}$ or pcDNA5-FRT-TO-GFP-CPAP-FL$_{MUT}$ together with the flippase-encoding vector pOG44 at a ratio of 1: 8 (pcDNA5: pOG44 = 0.5 µg: 4 µg) were transiently co-transfected per well of a 6-well plate at ~80% confluence. To generate stable knockout cell lines, the inducible cell lines were co-transfected with two different PX459 plasmids bearing the different sgRNAs and different drug resistance genes, puromycin and blasticidin. 1 µg for each plasmid was used for the transfection of one well of 6-well plate at ~40% confluence, together with 6 µl FuGENE 6 according to the manufacturer's instructions.

## Generation of inducible transgenic cell lines

To generate the inducible cell lines stably expressing full-length GFP-CPAP-FL$_{WT}$ or GFP-CPAP-FL$_{MUT}$, the Flp-In T-REx U2OS cells (host) with ~80% confluence in 6-well plate were transiently co-transfected with pcDNA5-FRT-TO-GFP-CPAP-FL$_{WT}$ or pcDNA5-FRT-TO-GFP-CPAP-FL$_{MUT}$ together with a plasmid-encoding flippase. After incubation overnight, cells were moved to 10-cm plates and grown for another 2 days. Cells were then cultured with medium containing 1 µg/ml puromycin until untransfected cells died (~4 days later). A portion of the remaining cells was used to test for GFP expression after culturing in medium containing 1 µg/ml doxycycline for 1 day. Subsequently, ~200 cells were sorted into 96-well plates and cultured for 2–3 wk. Around 50 single colonies were selected and transferred into 24-well plate. 1 wk later, the cells were transferred to 6-well plates and grown for ~3 days, and then to 10-cm plates until full confluence. Their CPAP expression levels were compared with normal control cells and host cells by western blotting, and cells with similar, low CPAP levels were used in the experiments.

## Generation of CRISPR/Cas-9 mediated stable knockout cell lines

To stably knock out the endogenous CPAP from the cell lines described above, two constructs bearing two sgRNAs that target the intron regions flanking exon 3 were transiently transfected into cells with ~40% confluence in a 6-well plate. To improve the screening efficiency of generating stable knockout cell lines, the two sgRNAs were cloned into two different vectors, bearing genes conferring resistance to two different drugs, puromycin and blasticidin. Immediately after transfection, cells were maintained in medium containing 1 µg/ml doxycycline to insure expression of the GFP-CPAP transgenes. After 24 h, cells were transferred to 10-cm plates and grown for another 2 days. Cells

were then cultured in medium containing 1 μg/ml puromycin and 5 μg/ml blasticidin until the non-transfected control cells died 4–5 days later. The subsequent cell sorting and cell expansion were performed as described above. When the stable cell lines are generated, their genomic DNA was extracted and used as a template for PCR, and the obtained PCR products were purified and sequenced. Their CPAP expression levels (leaky expression, without doxycycline induction) were compared by western blotting, and cell lines with similar, low CPAP levels were used in the experiments.

**Pull-down assays, immunoprecipitation, and western blotting**
For the streptavidin pull-down assays, 6-well plates of HEK293T cells with about 80–90% confluency were transfected with plasmid DNA and PEI (Polysciences) in 1:3 wt/wt ratio per well. Equal amounts of the bait, prey, and BirA biotin ligase DNA was used. 1 day after transfection, the medium was refreshed, and cells were harvested on the second day. Each sample was washed with ice-cold PBS and lysed on ice for 15 min with 100 μl lysis buffer (50 mM HEPES, 150 mM NaCl, and 1% Triton X-100) supplemented with protease and phosphatase inhibitor cocktail (Roche). 10% of the soluble fraction of the lysate was boiled with 2X Laemmli sample buffer. Dynabeads (Thermo Fisher Scientific) were blocked with 0.1% albumin from chicken egg white (Sigma-Aldrich) for 30 min and washed three times with the wash buffer (50 mM HEPES, 150 mM NaCl, and 0.1% Triton X-100, pH 7.4). DynaMag-2 (Invitrogen) magnets were used for washing the beads. The remaining 90% of the soluble fraction was incubated with the beads at 4°C for 1 h with continuous shaking. After three washes, the beads were boiled in a 2X Laemmli sample buffer. All samples were loaded on SDS-PAGE gels with a chosen percentage (6–9%) according to the protein size.

For GFP pull-down assays, 10-cm dishes of HEK293T with 70–80% confluency were transfected with equal amounts of bait and prey plasmid DNA and PEI in a 1:3 wt/wt ratio. 1 day after transfection, the cells were washed with ice-cold PBS and harvested. The cells were then lysed on ice for 30 min in 200 μl lysis buffer (10 mM Tris HCl pH 7.5, 150 mM NaCl, 0.5 mM EDTA, 0.5% IGEPAL CA-630, and 1 mM DTT) supplemented with EDTA-free protease inhibitor cocktail and PhosSTOP phosphatase inhibitor cocktail (Roche). The lysates were clarified by spinning for 20 min at 14,000 × *g*. The clarified lysate was diluted with 300 μl dilution buffer (10 mM Tris HCl pH 7.5, 150 mM NaCl, 0.5 mM EDTA, and 1 mM DTT) supplemented with EDTA-free protease inhibitor cocktail (Roche), and ~10% of this diluted lysate was mixed with 4X Laemmli sample buffer to be loaded as input. ChromoTek GFP-Trap Magnetic particles M-270 beads (Proteintech) were washed three times using the wash buffer (10 mM Tris HCl pH 7.5, 150 mM NaCl, 0.5 mM EDTA, 0.05% IGEPAL CA-630, and 1 mM DTT) and DynaMag-2 (Invitrogen) magnets were used for washing the beads. The remaining clarified lysate was incubated with the beads for 30 min at 4°C with continuous shaking. After three washes with wash buffer, the beads with the proteins were boiled with 2X Laemmli sample buffer. All samples were then loaded onto an SDS-PAGE gel.

For western blotting, unstained SDS-PAGE gels were transferred to a nitrocellulose membrane using either a wet transfer cell for 2 h at 250 V or a semidry transfer cell (Bio-rad) for 2 h at 12 Vs. Membranes were blocked for 30–60 min with 2% BSA before adding the primary antibody to incubate overnight at 4°C (primary antibodies mentioned in the Table 1). The membrane was washed three times for 5 min in PBS containing 0.05% Tween-20 (PBST) before adding the secondary antibody (secondary antibodies are mentioned in the Table 1). After 1 h incubation, again three washes with PBST were performed before imaging. Imaging was done on Odyssey CLx infrared imager (Li-Cor Biosciences).

For western blotting of overexpressed GFP-CPAP from U2OS lines, cells were harvested from 10-cm dishes at 90% confluence and protein extracts were prepared using RIPA buffer (50 mM Tris-HCl pH 7.5, 150 mM NaCl, 1% Triton X-100, and 0.5% sodium deoxycholate) supplemented with protease inhibitors and phosphatase inhibitors (Roche) and followed by three 3 s ultrasound-assisted protein extractions on ice. Proteins were transferred to a nitrocellulose membrane by wet transfer for 3 h at 100 V. The other steps are the same as mentioned above. The antibodies used for western blotting are listed in the Table 1.

**Immunofluorescence staining**
For immunofluorescence cell staining, cultured cells were fixed with –20°C methanol for 5 min, rinsed in PBS for 5 min, permeabilized with 0.15% Triton X-100 in PBS for 2 min, washed three times for 5 min with 0.05% Tween-20 in PBS, sequentially incubated for 20 min in the blocking buffer (2% BSA and 0.05% Tween-20 in PBS), then for 1 h with primary antibodies in the blocking buffer, washed three times for 5 min with 0.05% Tween-20 in PBS, then for 1 h in secondary antibodies in the blocking buffer, washed three times for 5 min with 0.05% Tween-20 in PBS, and air-dried after a quick wash in 96% ethanol. Cells were mounted with ProLong Glass Antifade Mountant (Life Technologies). Antibodies used for immunostaining are listed in the Table 1.

**Ultrastructure expansion microscopy**
Cells were incubated on ice for 30 min and pre-extracted for 90 s using 0.1% Triton X-100 in MRB80 (80 mM PIPES pH 6.8, 1 mM EGTA, and 4 mM MgCl$_2$) and prefixed for 8 min with MeOH at –20°C. Then, cells were washed twice with PBS and fixed for 5 h with 1.4% PFA and 2.0% acrylamide in PBS at 37°C. After a short wash with PBS, a gelation solution (23% wt/vol sodium acrylate, 10% wt/vol acrylamide, 0.1% N, N'-methylenebisacrylamide, 1X PBS, 0.5% tetramethylethylenediamine, and ammonium persulfate) was added to a rubber gelation chamber on a parafilm-covered glass slide and sealed with the cover glass containing the cells. The gel was allowed to polymerize for 1 h at 37°C. After a short wash with PBS, disruption buffer (0.2 M SDS, 0.2 M NaCl, and 50 mM Tris calibrated to pH 9.0) was added and incubated for 15 min at room temperature with agitation before further disruption for 90 min at 95°C. Then, the gel was expanded in MQ for at least 30 min before shrinking it back in multiple rounds of PBS washes. To stain the sample, a quarter of the gel was incubated overnight at 4°C in primary antibody in 1% BSA 0.1%

Triton X-100 in a total volume of 600 µl. Gels were washed extensively by multiple rounds: 30 min in 1% BSA 0.1% Triton X-100 in 1X PBS, 30 min in 0.1% Triton X-100 in 1X PBS, and 3 × 1 h wash in 0.1% Triton X-100 in 1X PBS. Gels were incubated overnight at 4°C in secondary antibody in 1% BSA 0.1% Triton X-100 in a total volume of 600 µl. Then, gels were washed similar to the wash steps after primary antibody incubation. To expand the gels, they were transferred to a 10-cm petri dish and incubated for at least 4 h in water at room temperature. Prior to imaging, the cells were trimmed and mounted onto a plasma-cleaned, poly-L-lysine–treated cover glass. A complete list of primary and secondary antibodies, including their dilution, are listed in Table 1. Microscopy images of centrioles were deconvolved using Huygens Professional 21.04.

## Protein purification from HEK293T cells for in vitro reconstitutions

Proteins were purified from HEK293T cells using the Strep-Tactin affinity purification as previously described in Sharma et al. (2016). The plasmid DNA was mixed with PEI (stock 1 mg/ml) in a ratio of 1:3 wt/wt in antibiotics-free Ham's F10 (Gibco) and incubated for 20 min at room temperature. The PEI-DNA mixture was then gently added to the adherent HEK293T cells in complete DMEM and incubated at 37°C in a 5% $CO_2$ incubator. Cells were harvested 1 or 2 days after transfection depending on the protein. The cells from one 15-cm dish were lysed in 500 µl lysis buffer (50 mM HEPES, 300 mM NaCl, and 0.5% Triton X-100 pH 7.4 with or without 1 mM EGTA and 1 mM $MgCl_2$, depending on the protein) supplemented with protease cOmplete inhibitors cocktail (Roche). After clearing debris by centrifugation at 14,000 × $g$ for 20 min, cell lysates were incubated with StrepTactin beads (GE Healthcare) for 45 min. Beads were washed five times with lysis buffer without protease inhibitors. The proteins were eluted in elution buffer (50 mM HEPES, 150 mM NaCl, 1 mM DTT, 2.5 mM d-desthiobiotin, and 0.05% Triton X-100, pH 7.4 with or without 1 mM EGTA and 1 mM $MgCl_2$, depending on the protein). All purified proteins were snap frozen in liquid nitrogen and stored at –80°C.

## Protein expression and purification from *E. coli* for biophysical and structural studies

CPAP-CC1 (residues 89–196) and CP110-CC2 (residues 635–717) were amplified by PCR and cloned into the bacterial expression vector PSPCm9 (Olieric et al., 2010) containing N-terminal thioredoxin, a 6x His-tag, and a PreScission cleavage site. Mutants of CPAP-CC1 and CP110-CC2 were generated using a PCR-based site-directed mutagenesis approach. The DNA sequences of all the established constructs were validated via sequencing.

Protein expression was performed in the *E. coli* strain BL21(DE3). In brief, LB medium containing 50 mg/ml of kanamycin was used for growing the transfected *E. coli* cells at 37°C. Once cell cultures reached an $OD_{600}$ of 0.6, they were cooled down to 18°C and then induced with 0.4 mM IPTG. Proteins were expressed overnight at 18°C. The next day, cells were harvested by centrifugation, washed in cold PBS buffer, and lysed via sonication in a lysis buffer (20 mM HEPES pH 7.5, 1 M NaCl, 10% glycerol, 30 mM imidazole pH 8.0, and 5 mM β-mercaptoethanol), supplemented with protease cOmplete inhibitor cocktail (Roche) and DNase (Sigma-Aldrich). After high-speed centrifugation at 18,000 × $g$, the supernatants were collected and applied onto a HiTrap Ni-NTA column (Cytiva) for immobilized metal-affinity chromatography (IMAC) purification at 4°C. The bound proteins were washed extensively with IMAC buffer to remove nonspecifically bound proteins. Bound proteins were eluted by increasing the concentration of imidazole to 500 mM. To cleave off the N-terminal thioredoxin-His fusion tag, the eluted fractions were pooled and incubated in the presence of His-tagged HRV 3C protease (Cordingley et al., 1990) overnight at 4°C in IMAC buffer. The cleaved samples were separated from non-cleaved proteins and HRV 3C protease via a HiTrap Ni-NTA purification step. Cleaved proteins were concentrated and loaded onto a SEC HiLoad Superdex 75 16/60 column (Cytiva) for final purification in a final buffer (20 mM HEPES pH 7.5, 150 mM NaCl, and 2 mM β-mercaptoethanol). The quality and identity of proteins were assessed by SDS-PAGE and mass spectrometry before storing at –80°C for further experiments.

## Mass spectrometry of purified proteins

To confirm the identity of purified proteins, purified protein samples were digested using S-TRAP microfilters (ProtiFi) according to the manufacturer's protocol. Briefly, 4 µg of protein sample was denatured in 5% SDS buffer and reduced and alkylated using DTT (20 mM, 10 min, 95°C) and IAA (40 mM, 30 min). Next, samples were acidified, and proteins were precipitated using a methanol TEAB buffer before loading on the S-TRAP column. Trapped proteins were washed four times with the methanol TEAB buffer and then digested overnight at 37°C using 1 µg trypsin (Promega). Digested peptides were eluted and dried in a vacuum centrifuge before LC-MS analysis.

Samples were analyzed by reversed-phase nLC-MS/MS using an Ultimate 3000 UHPLC coupled to an Orbitrap Q Exactive HF-X mass spectrometer (Thermo Fisher Scientific). Digested peptides were separated using a 50-cm reversed-phase column packed in-house (Agilent Poroshell EC-C18, 2.7 µm, 50 cm × 75 µm) and were eluted at a flow rate of 300 nl/min using a linear gradient with buffer A (0.1% FA) and buffer B (80% ACN, 0.1% FA) ranging from 13 to 44% B over 38 min, followed by a column wash and re-equilibration step. The total data acquisition time was 55 min. MS data were acquired using a DDA method with the following MS1 scan parameters: 60,000 resolution, AGC target equal to 3E6, maximum injection time of 20 ms, the scan range of 375–1,600 m/z, and acquired in profile mode. The MS2 method was set at 15,000 resolution, with an AGC target set to standard, an automatic maximum injection time, and an isolation window of 1.4 m/z. Scans were acquired using a fixed first mass of 120 m/z and a mass range of 200–2,000, and an NCE of 28. Precursor ions were selected for fragmentation using a 1 s scan cycle, a dynamic exclusion time set to 10 s, and a precursor charge selection filter for ions possessing +2 to +6 charges.

Raw files were processed using Proteome Discoverer (version 2.4; Thermo Fisher Scientific). MSMS fragment spectra were searched using Sequest HT against a human database (UniProt,

year 2020) that was modified to contain protein sequences from our cloning constructs and a common contaminants database. The search parameters were set using a precursor mass tolerance of 20 ppm and a fragment mass tolerance of 0.06 Da. Trypsin digestion was selected with a maximum of two missed cleavages. Variable modifications were set as methionine oxidation and protein N-term acetylation and fixed modifications were set to carbamidomethylation. Percolator was used to assign a 1% false discovery rate (FDR) for peptide spectral matches, and a 1% FDR was applied to peptide and protein assemblies. An additional filter requiring a minimum Sequest score of 2.0 was set for PSM inclusion. MS1-based quantification was performed using the Precursor Ion Quantifier node with default settings applied. Precursor ion feature matching was enabled using the Feature Mapper node. Proteins matching the common contaminate database were filtered out from the results table.

## CD spectroscopy

Far-UV CD spectra of proteins samples were recorded at 5°C using a Chirascan-Plus spectrophotometer (Applied Photophysics Ltd.), equipped with a computer-controlled Peltier element. A 400 µl of protein sample with the final concentration of 0.2 mg/ml in PBS was loaded into a quartz cuvette of 1-mm optical path length. The thermal stability of each protein sample was analyzed by monitoring their CD spectrum at 222 nm using constant heating from 5°C to 85°C with 1°C per min intervals. The apparent midpoint of the transition, referring to the melting temperature was determined by fitting the data points with the GraphPad Prism 7 by choosing the nonlinear least-square fitting function based on a sigmoid model.

## Size-exclusion chromatography coupled with multi-angle light scattering

SEC-MALS was done at 20°C using a Superdex S75 10/30 or a Superdex S200 10/30 column (Cytiva). The system was purged and equilibrated overnight using an Agilent UltiMate3000 HPLC in the buffer (20 mM HEPES pH 7.5, 150 mM NaCl, and 2 mM DTT) with a flow rate of 0.5 ml/min. For each experiment, 15 µl of protein sample was loaded onto the respective SEC column at a concentration of ~7 mg/ml. The molecular mass of protein samples was determined using the miniDAWN TREOS and Optilab T-rEX refractive index detectors (Wyatt Technology). For the data fitting, the Zimm model was selected in the ASTRA 6 software.

## SAXS data collection and analysis

SAXS data were collected at the small-angle scattering beamline B21 of the Diamond Light Source. Protein samples (in 50 mM HEPES pH 7.5, 100 mM NaCl, 1 mM DTT, and 1 mM MgCl$_2$) were passed through a Shodex KW402.5-4F SEC column in line to the X-ray scattering measurement cell. Samples of 20 and 10 mg/ml protein concentrations and 80 µl volume were used; however, only data from the lower concentration samples were analyzed due to superior homogeneity, as judged by the SEC profile. Buffer subtraction, summation of scattering intensities across peaks in size-exclusion chromatograms, calculation of the radius of gyration from Guinier plots, estimation of molecular weight

from scattering volume-of-correlation plots, and evaluation of distance distribution functions (P(r)) were performed using Scatter3 (Forster et al., 2010). Ab initio calculation of molecular volumes from P(r) distributions was performed using DAMMIF (Franke and Svergun, 2009). For each dataset, 23 bead-based models were derived using random starting seeds and assuming no internal volume symmetry (P1). Pairwise cross-correlation and averaging of models was performed by DAMAVER (Volkov and Svergun, 2003). The final CP110-CC2 envelope derives from averaging of 22 calculated models with NSD 0.67 ± 0.05, while the CPAP-CC1/CP110-CC2 envelope is the average of 22 models with NSD 0.68 ± 0.04. Bead models were converted to volumetric envelopes using Situs 3 (Kovacs et al., 2018); graphical representations were created in UCSF Chimera (Pettersen et al., 2004).

## Chemical crosslinking combined with mass spectrometry

CP110-CC2 homodimers and CPAP-CC1/CP110-CC2 heterodimers were crosslinked using the coupling reagent 4-(4,6-dimethoxy-1,3,5-triazin-2-yl)-4-methylmorpholinium chloride (Sigma-Aldrich), which introduces zero-length crosslinks by coupling the carboxyl groups of aspartate and glutamate side chains to the primary amines of lysine side chains (Leitner et al., 2014). Crosslinking was performed for 6 min at 25°C and 1,200 rpm at a final concentration of 60 mM. The reaction was quenched using a desalting column (Thermo Fisher Scientific), followed by the addition of ammonium bicarbonate.

Crosslinked samples were denatured by adding two sample volumes of 8 M urea, reduced with 5 mM TCEP (Thermo Fisher Scientific), and alkylated by adding 10 mM iodoacetamide (Sigma-Aldrich) for 40 min at room temperature. Digestion was performed with lysyl endopeptidase (1:50 wt/wt; Wako) for 2 h followed by a second digest with trypsin at 35°C overnight at 1,200 rpm (1:50 ratio wt/wt; Promega). Proteolysis was stopped by the addition of 1% (vol/vol) TFA. Crosslinked peptides were purified by reversed-phase chromatography using C18 cartridges (Sep-Pak, Waters) and enriched on a Superdex Peptide PC 3.2/30 column (300 × 3.2 mm).

Fractions of crosslinked peptides were analyzed by liquid chromatography coupled to tandem mass spectrometry using an LTQ Orbitrap Elite (Thermo Fisher Scientific) instrument (Herzog et al., 2012). Crosslinked peptides were identified using xQuest (Walzthoeni et al., 2012). The results were filtered with an MS1 tolerance window of –4 to 4 ppm and score ≥22 followed by manual validation. The intensities of the identified crosslinks were extracted and normalized by using a modified protocol of the previously published software xTract (Walzthoeni et al., 2015).

## In vitro reconstitution of MT dynamics

The in vitro assays with dynamic MTs were performed under the same conditions as described previously by Sharma et al. (2016). Briefly, in vitro flow chambers for TIRF microscopy were assembled on microscopic slides by two strips of double-sided tape with plasma-cleaned glass coverslips. Flow chambers were functionalized by sequential incubation with 0.2 mg/ml PLL-PEG-biotin (Susos AG) and 0.83 mg/ml neutravidin

(Invitrogen) dissolved in MRB80 buffer Next, GMPCPP-stabilized MT seeds were attached to the coverslips through biotin–neutravidin interactions. The flow chambers were further blocked with 1 mg/ml κ-casein. The reaction mix constituting of MRB80 buffer supplemented with 14.5 µM porcine brain tubulin, 0.5 µM (X-rhodamine/HiLyte 488/HiLyte 647)-labelled tubulin, 75 mM KCl, 1 mM GTP, 0.5 mg/ml κ-casein, 0.1% methylcellulose, and oxygen scavenger mix (50 mM glucose, 400 µg/ml glucose oxidase, 200 µg/ml catalase, and 4 mM DT), along with different concentrations and combinations of the respective purified proteins was prepared and spun in an Airfuge for 5 min at 119,000 × $g$. Finally, the spun reaction was flowed into the chamber, and the chamber was sealed with vacuum grease. Dynamic MTs were imaged immediately at 30°C using a TIRF microscope. For flow-in assays, the chambers were not sealed, and the new reaction was flowed into the chamber during acquisition. All tubulin products used are from Cytoskeleton Inc.

### TIRF microscopy
TIRF imaging was performed on a microscope setup (inverted research microscope; Nikon Eclipse Ti-E), equipped with the perfect focus system (Nikon) and a Nikon CFI Apo TIRF 100/1.49 numerical aperture oil objective (Nikon). The microscope was supplemented with a TIRF-E motorized TIRF illuminator, modified by Roper Scientific/PICT-IBiSA Institut Curie, and a stage-top incubator (model no. INUBG2E-ZILCS; Tokai Hit) to maintain the temperature of the sample at 30°C. Image acquisition was performed using either a Photometrics Evolve 512 EMCCD camera (Roper Scientific) or a Photometrics CoolSNAP HQ2 CCD camera (Roper Scientific) and controlled with MetaMorph7.10.2.240 software (Molecular Devices). The Evolve EMCCD camera's final resolution was 0.064 µm/pixel, while with the CoolSNAP Myo CCD camera, it was 0.045 µm/pixel. For excitation lasers, 491-nm 100-mW Stradus (Vortran), 561-nm 100-mW Jive (Cobolt), and 642-nm 110-mW Stradus (Vortran) were used. An ET-GFP 49002 filter set (Chroma) was used for imaging proteins tagged with GFP, an ET-mCherry 49008 filter set (Chroma) for imaging X-rhodamine-labelled tubulin or mCherry-tagged proteins, and an ET647 for imaging Alexa 647-labelled tubulin. Time-lapse movies of dynamic MTs with different protein combinations were taken with sequential imaging of channels and a 3 s time interval for 10 min unless mentioned otherwise.

### Analysis of MT plus end dynamics in vitro
Kymographs were generated using the ImageJ plugin KymoResliceWide v.0.4 (https://github.com/ekatrukha/KymoResliceWide). MT dynamics parameters were obtained from the kymographs.

For the experiments determining the proportion of MTs blocked or paused, kymographs were generated and MTs were manually sorted as completely blocked, occasionally paused, or with no visible effects on dynamic MTs. Only pause events that lasted for 15 s (5 pixels) or longer were included in the analysis.

For determining the MT dynamics, MT plus ends were manually traced from the kymographs (regardless of whether they were growing, blocked, or paused) in Fiji, and the angles and length of the line traces were extracted. This information was used to further calculate the growth durations, growth rates, pause durations, and catastrophe frequencies in various assay conditions. This dataset was also used to generate the cumulative histogram showing % of total time that MT plus ends spent growing at different growth rates using a MATLAB code (https://gist.github.com/ekatrukha/8a8f336d44b7bef6523fc1acc3c71a19). The duration of every event was divided by the sum of durations of all the events. The quantitative data reported for each experiment were collected in at least three independent assays.

### Single molecule counting and fluorescence intensity analysis
The single molecule counting and fluorescence intensity analysis was done as described in van Riel et al. (2017). Briefly, three parallel flow chambers were made on the same plasma-cleaned coverslip and then appropriate dilutions of purified monomeric GFP, dimeric GFP-EB3, and CEP97^CP110-GFP in MRB80 buffer were flowed into each of these chambers. After protein addition, the flow chambers were washed with MRB80 buffer, sealed with vacuum grease, and immediately imaged with a TIRF microscope. About 40 images of unexposed coverslip areas were acquired with 100-ms exposure time and low-laser power. Single-molecule fluorescence spots were detected and fitted with 2D Gaussian function using custom-written ImageJ plugin DoM_Utrecht v.1.2.2 (https://github.com/ekatrukha/DoM_Utrecht). The fitted peak intensity values were used to build fluorescence intensity histograms. The histograms were fitted to Gaussian distributions using GraphPad Prism 9. To estimate the number of CEP97^CP110-GFP molecules that might be causing the observed pausing or blocking of MT plus end, appropriate dilutions of monomeric GFP were added to in vitro chambers such that single GFP molecules would be immobilized on the coverslip. In a parallel chamber on the same coverslip, in vitro reconstitution assay with different concentrations of CEP97^CP110-GFP was added. Images of single-unbleached molecules of GFP as well as CEP97^CP110-GFP in the in vitro reaction were acquired first. After this, time-lapse imaging was performed on the in vitro assay using the same illumination parameters. The CEP97^CP110-GFP accumulations completely blocking or pausing dynamic MTs were manually located as regions of interest in each frame and fitted with 2D Gaussian as described above. For building the distributions of molecules at the MT tip, each CEP97^CP110-GFP intensity value at the MT plus end was normalized by the average GFP single molecules intensity from the adjacent chamber. The same procedure as described above was followed to compare the intensities of CPAP-N$_{WT}$ and CPAP-N$_{MUT}$ molecules on MT plus ends and to examine the influence of (TM-3)$_2$ DARPin on CEP97^CP110 intensity on the MT plus end.

### FRAP assay
The FRAP experiment was done on the TIRF microscope equipped with an iLas system (Roper scientific/PICT-IBiSA) as described previously (Rai et al., 2020). Photobleaching of CEP97^CP110-GFP on a blocked end or a slow-growing end was performed with a 488-nm laser. Time-lapse movies were taken

for 2 min, followed by bleaching with 488-nm laser, followed by 8 min imaging after bleaching. A 561-nm laser was used for bleaching CPAP-N$_{WT}$-mCherry, and the same imaging-bleaching paradigm was used. Additionally, three assays were performed where both CEP97^CP110-GFP and CPAP-N$_{WT}$-mCherry were bleached together using a 488-nm laser. Bleached MT plus ends were traced on the kymographs to generate plot profiles of the intensity. The line was traced only till when the MT end was slow growing or stationary. If the MT end started growing at a visibly faster speed, it was not taken into consideration for recovery analysis.

### Cryo-electron tomography

MTs were grown by incubating GMPCPP-stabilized doubly cycled seeds with 15 µM porcine brain tubulin in the MRB80 buffer. The reaction mix was centrifuged in Beckman Airfuge for 5 min at 119,000 × *g* prior to mixing with seeds in a tube. In samples with CEP97^CP110-GFP present, 80 nM of the protein was added to the reaction mix before centrifugation (one grid) or after centrifugation (two grids). After incubation for 6–20 min at 37°C, 5-nm gold particles were added to the mix, and then 3.5 µl was transferred to a recently glow-discharged, lacey carbon grid suspended in the chamber of Leica EM GP2 plunge freezer, equilibrated at 37°C and 98% relative humidity. The grid was immediately blotted for 4 s and plunge frozen in liquid ethane.

Images were recorded on a JEM3200FSC microscope (JEOL) equipped with a K2 Summit direct electron detector (Gatan) and an in-column energy filter operated in zero-loss imaging mode with a 30-eV slit. Images were recorded at 300 kV with a nominal magnification of 10,000, resulting in a pixel size of 3.668 Å at the specimen level. Imaging was performed using SerialEM software (Mastronarde, 2005), recording bidirectional tilt series starting from 0° ± 60°; tilt increment 2°; total dose of 80–100 e–/Å²; and target defocus –4 µm.

### Analysis of cryo-ET images and 3D volume reconstruction

Tomographic tilt series were processed as outlined in Fig. S6 (analysis flowchart). Direct electron detector movie frames were aligned using MotionCor2 (Zheng et al., 2017) and then split into full, even, and odd stacks. Tilt series alignment and tomographic reconstructions were performed on sums of full stacks with the IMOD software package using gold beads as fiducial markers (Kremer et al., 1996). Final tomographic volumes were binned by two, corrected for contrast transfer function, and the densities of gold beads were erased in IMOD. 3D volumes were subsequently denoised using the cryoCARE procedure (Buchholz et al., 2019). This procedure enhanced the signal-to-noise ratio in the reconstructed 3D volumes and significantly improved the segmentation of individual PFs at the ends of MTs and their manual tracing. For this, 3D reconstruction was performed on odd and even aligned stacks with the IMOD parameters identified for full stacks. We trained two to three denoiser models for each acquisition series and then applied one model to the rest of the tomograms in this series. Splitting of movie frames, reconstructing even and odd volumes, training data generation, model training, and denoising was performed

on a cluster of graphics processing units using python scripts (available at https://github.com/NemoAndrea/cryoCARE-hpc04).

Sub-volumes containing MT ends were manually extracted from denoised tomographic volumes and processed further. First, the polarity of MTs was determined on summed projections using moiré patterns of images Fourier filtered at the origin using Fiji (Chretien et al., 1996; Schindelin et al., 2012). Following the previously published procedure to obtain PF coordinates (McIntosh et al., 2018), 3D models were manually built for each MT end in 3dmod (Kremer et al., 1996). Each PF was stored as a separate contour, the first point in a contour was placed on a MT wall, the second point at the last segment of the PF that was still in the MT cylinder, and the following points were placed every 2–4 nm along the bending part of the PF. Accuracy of manual segmentation was constantly monitored in the Isosurface view of 3dmod, which contained both the rendered 3D representation of the tomographic volume and the manually built 3D model. This procedure resulted in 3D models such as those presented in Fig. 2, B and D. Coordinates of the PFs were then extracted using the "howflared" program in IMOD.

PF coordinates were further analyzed using MATLAB scripts available at https://github.com/ngudimchuk/Process-PFs. These scripts are based on the previously published ones (Gudimchuk et al., 2020; McIntosh et al., 2018), but they were modified to account for PF shapes that deviated from 2D planes. As reported previously, the sampling along the PF was made uniform by interpolation and then smoothed using quadratic LOESS with a window of 10 points. Curvature was calculated as the angle between consecutive pairs of line segments in LOESS-smoothed traces.

To segment the denoised densities into tubulin and cap, we used the Tomoseg module of EMAN2.2 (Chen et al., 2017). Using a full denoised tomogram containing MTs grown in the presence of CEP97^CP110, we boxed reference regions sets containing (1) MT walls, (2) bent PFs at MT ends and soluble tubulin oligomers, (3) caps at MT ends, and (4) "bad" regions containing carbon support, gold particles, ice contamination, etc. These boxed sets were then manually segmented, and three neural networks were trained: 1 versus 4, 2 versus 4, and 3 versus 4. The resulting neural networks were applied to sub-volumes containing MT ends, and the resulting segmentations were used to mask tomographic densities in UCSF Chimera (Pettersen et al., 2004). Masked densities were imported into Blender to make visualizations.

### Statistical analysis

Statistical analyses were performed using GraphPad 9/10. Additional details about the tests performed, significance levels, and number of measurements are reported in the figure legends.

### Online supplemental material

Fig. S1 illustrates the characterization of purified proteins used in this study. Fig. S2 shows the subcellular localization of the constructs encoding CP110 and its fragments, CEP97, and CEP97^CP110 chimera. Fig. S3 provides the additional images of MT ends by cryo-ET. Fig. S4 shows the biophysical

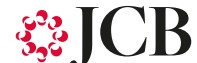

characterization of CPAP-CC1, CP110-CC2, and their complex. Fig. S5 provides the additional data on the characterization of stable cell lines expressing WT or mutant GFP-CPAP. Fig. S6 presents a flowchart illustrating the pipeline for 3D reconstruction, denoising, segmentation, and visualization of tomographic volumes, described in Materials and methods. Video 1 demonstrates that a dynamic MT plus end becomes blocked upon flow in of CEP97^CP110-GFP. Video 2 shows a 3D view of a MT plus end in the presence of CEP97^CP110-GFP. Video 3 illustrates the slow MT plus-end growth in the presence of CEP97^CP110-GFP and CPAP-N$_{WT}$-mCh.

### Data availability

The data that support the conclusions are available in the manuscript; the original fluorescence microscopy datasets are available upon request to A. Akhmanova. Tomography data are available from EMDB using the following accession codes: MTs in presence of tubulin and GMPCPP-stabilized seeds (EMD-14101 and EMD-14102), MTs in presence of CEP97^CP110, tubulin, and GMPCPP-stabilized seeds (EMD-14103, EMD-14104, and EMD-14105). SAXS data and models are deposited in SASBDB: CPAP-CC1/CP110-CC2 heterodimer, accession code SASDNA3; CP110-CC2 homodimer, accession code SASDNB3. Scripts used for data analysis are available at https://github.com/ekatrukha/KymoResliceWide, https://github.com/ekatrukha/DoM_Utrecht, https://github.com/NemoAndrea/cryoCARE-hpc04, and https://github.com/ngudimchuk/Process-PFs.

## Acknowledgments

We thank R. Stucchi and A.F.M. Altelaar (Utrecht University) for the help with mass spectrometry, W. Evers and A. Jakobi lab (TU Delft) for the help with cryo-ET. J.R. McIntosh (University of Colorado) for the feedback on PF shape analysis, and D. Mastronarde (University of Colorado) for modifying the Howflared program to extract 3D coordinates from PF models. We thank the Diamond Light Source for the provision of synchrotron radiation facilities and the staff of Beamline B21 for their help.

This work was supported by the European Research Council Synergy grant 609822 to M. Dogterom and A. Akhmanova, the Netherlands Organization for Scientific Research ALWOP.440 grant to A. Akhmanova, the China Scholarship Council scholarship to F. Chen. The Medical Research Council UK (MR/N009274/1) grant to I. Vakonakis, the European Molecular Biology Organization Long Term Fellowship ALTF-840-2018 to F.E. Ogunmolu, the Marie Curie COFUND Fellowship (Marie Skłodowska-Curie grant agreement 701647) to S. Moradi, and the grants from the Swiss National Science Foundation (31003A_166608 and 310030_192566 to M.O. Steinmetz and 310030M_215014 to A. Akhmanova). Analysis of the shapes of tubulin PFs was supported by the Russian Science Foundation grant 21-74-20035 to N. Gudimchuk.

Author contributions: S.S. Iyer: conceptualization, data curation, formal analysis, investigation, methodology, validation, visualization, and writing - original draft, review, and editing. F. Chen: conceptualization, data curation, formal analysis, funding acquisition, investigation, methodology, resources, validation, visualization, and writing—original draft. F.E. Ogunmolu: conceptualization, data curation, formal analysis, funding acquisition, investigation, methodology, project administration, validation, visualization, and writing—original draft. S. Moradi: conceptualization, data curation, formal analysis, investigation, methodology, validation, visualization, and writing—original draft, review, and editing. V.A. Volkov: conceptualization, data curation, formal analysis, investigation, methodology, validation, visualization, and writing—review and editing. E.J. van Grinsven: conceptualization, formal analysis, investigation, methodology, visualization, and writing—original draft, review, and editing. C. van Hoorn: investigation and visualization. J. Wu: conceptualization and investigation. N. Andrea: software. S. Hua: resources. K. Jiang: resources. I. Vakonakis: formal analysis, investigation, methodology, and writing—review and editing. M. Potočnjak: data curation, formal analysis, investigation, validation, visualization, and writing—original draft. F. Herzog: data curation and formal analysis. B. Gigant: resources. N. Gudimchuk: formal analysis and software. K.E. Stecker: data curation, formal analysis, and investigation. M. Dogterom: funding acquisition and supervision. M.O. Steinmetz: conceptualization, data curation, formal analysis, funding acquisition, supervision, and writing—original draft, review, and editing. A. Akhmanova: conceptualization, funding acquisition, project administration, supervision, and writing—original draft, review, and editing.

Disclosures: The authors declare no competing interests exist.

Submitted: 12 June 2024

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

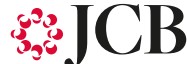

# Supplemental material

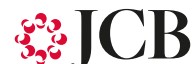

## A

SII-GFP-CP110 — CEP97-GFP-SII — CEP97^CP110-GFP-SII

## B Analysis of CP110, CEP97 and CEP97^CP110 by mass spectrometry

### GFP-CP110

| Accession | Description | % Coverage | # PSMs |
|---|---|---|---|
| O43303 | Centriolar coiled-protein 110 kDa [CP110] | 75 | 1377 |
| P0DMV9 | Heat shock 70 kDa protein 1B [HSPA1B] | 72 | 697 |
| P07437 | Tubulin β - chain [TUBB] | 75 | 551 |
| P68371 | Tubulin β - 4B chain [TUBB4B] | 77 | 517 |
| Q9BVA1 | Tubulin β - 2B chain [TUBB2B] | 70 | 455 |
| Q13885 | Tubulin β - 2A chain [TUBB2A] | 70 | 455 |
| Q71U36 | Tubulin α - 1A chain [TUBA1A] | 71 | 413 |
| Q9BQE3 | Tubulin α - 1C chain [TUBA1C] | 71 | 400 |
| P04350 | Tubulin β - 4A chain [TUBB4A] | 75 | 400 |
| P11142 | Heat shock cognate 71 kDa protein [HSP7C] | 62 | 373 |

### CEP97-GFP

| Accession | Description | % Coverage | # PSMs |
|---|---|---|---|
| Q8IW35 | Centrosomal protein of 97 kDa [CEP97] | 92 | 3278 |
| P0DMV9 | Heat shock 70 kDa protein 1B [HSPA1B] | 85 | 868 |
| P07437 | Tubulin β - chain [TUBB] | 84 | 311 |
| P68371 | Tubulin β - 4B chain [TUBB4B] | 84 | 289 |
| P34931 | Heat shock 70 kDa protein [HSPA1] | 34 | 276 |
| P11142 | Heat shock cognate 71 kDa protein [HSP7C] | 68 | 260 |
| Q9BVA1 | Tubulin β - 2B chain [TUBB2B] | 77 | 258 |
| Q13885 | Tubulin β - 2A chain [TUBB2A] | 68 | 255 |
| P04350 | Tubulin β - 4A chain [TUBB4A] | 82 | 237 |
| P38363 | Tubulin α - 1B chain [TUBA1B] | 69 | 237 |

### CEP97^CP110-GFP

| Accession | Description | % Coverage | # PSMs |
|---|---|---|---|
| Chimera | CEP97^CP110 | 81 | 4641 |
| Q8IW35 | Centrosomal protein of 97 kDa [CEP97] | 85 | 2118 |
| Q43303 | Centriolar coiled-coil protein of 110 kDa [CP110] | 59 | 1479 |
| P0DMV9 | Heat shock 70 kDa protein 1B [HSPA1B] | 88 | 1053 |
| P07437 | Tubulin β - chain [TUBB] | 85 | 448 |
| P68363 | Tubulin α - 1B chain [TUBA1B] | 77 | 400 |
| Q71U36 | Tubulin α - 1A chain [TUBA1A] | 77 | 379 |
| P68371 | Tubulin β - 4B chain [TUBB4B] | 84 | 367 |
| Q9BQE3 | Tubulin α - 1C chain [TUBA1C] | 77 | 358 |
| Q9BVA1 | Tubulin β - 2B chain [TUBB2B] | 78 | 333 |

## C

## D Tubulin (15 µM) CEP97-GFP (50 nM)

## E

## G Analysis of CPAP by mass spectrometry

### CPAPNWT-mCherry

| Accession | Description | % Coverage | # PSMs |
|---|---|---|---|
| Q9HC77 | Centromere protein J [CPAPN-WT] | 43 | 2317 |
| P68371 | Tubulin β - 4B chain [TUBB4B] | 86 | 1527 |
| P07437 | Tubulin β - chain [TUBB] | 86 | 1517 |
| P04350 | Tubulin β - 4A chain [TUBB4A] | 86 | 1336 |
| P0DMV9 | Heat shock 70 kDa protein [HSPA1B] | 90 | 1303 |
| Q71U36 | Tubulin α - 1A chain [TUBA1A] | 86 | 1191 |
| Q9BVA1 | Tubulin β - 2B chain [TUBB2B] | 85 | 1177 |
| Q13885 | Tubulin β - 2A chain [TUBB2A] | 77 | 1174 |
| Q9BQE3 | Tubulin α - 1C chain [TUBA1C] | 86 | 1124 |

### CPAPNMUT-mCherry

| Accession | Description | % Coverage | # PSMs |
|---|---|---|---|
| Q9HC77 | Centromere protein J [CPAPN-MUT] | 43 | 2079 |
| P68371 | Tubulin β - 4B chain [TUBB4B] | 86 | 1243 |
| P07437 | Tubulin β - chain [TUBB] | 86 | 1217 |
| P0DMV9 | Heat shock 70 kDa protein [HSPA1B] | 90 | 1087 |
| P04350 | Tubulin β - 4A chain [TUBB4A] | 86 | 1085 |
| Q71U36 | Tubulin α - 1A chain [TUBA1A] | 80 | 978 |
| Q9BQE3 | Tubulin α - 1C chain [TUBA1C] | 80 | 931 |
| Q9BVA1 | Tubulin β - 2B chain [TUBB2B] | 85 | 927 |
| Q13885 | Tubulin β - 2A chain [TUBB2A] | 77 | 925 |
| Q13509 | Tubulin β - 3 chain [TUBB3] | 54 | 753 |

## F CPAP-N_WT mCh-SII — CPAP-N_MUT mCh-SII

Figure S1.   **Characterization of purified proteins used in this study. (A)** SDS-PAGE gel of GFP-CP110, CEP97-GFP, and CEP97^CP110-GFP, purified from HEK293T cells. Gels were stained with Coomassie brilliant blue R250. **(B)** Analysis of purified GFP-CP110, CEP97-GFP, and CEP97^CP110-GFP by mass spectrometry. **(C)** The proportion of fully blocked MTs with increasing concentrations of GFP-CP110 in in vitro reconstitution assays. $n$ = 91, 28, 142, 105, and 140 MT plus ends for 5, 10, 20, 30, and 50 nM GFP-CP110. **(D)** A still image and a kymograph representing dynamic MT (blue) behavior in the presence of 50 nM CEP97-GFP (green, no binding). **(E)** Bar plot showing that CEP97-GFP does not affect the plus end blocking of dynamic MTs in vitro by GFP-CP110. The numbers of analyzed MTs are indicated on the bar plots. **(F)** SDS-PAGE of CPAP-N_WT-mCh and CPAP-N_MUT-mCh, purified from HEK293T cells. Gels were stained with Coomassie brilliant blue R250. **(G)** Analysis of purified CPAP-N_WT-mCh and CPAP-N_MUT-mCh by mass spectrometry. Source data are available for this figure: SourceData FS1.

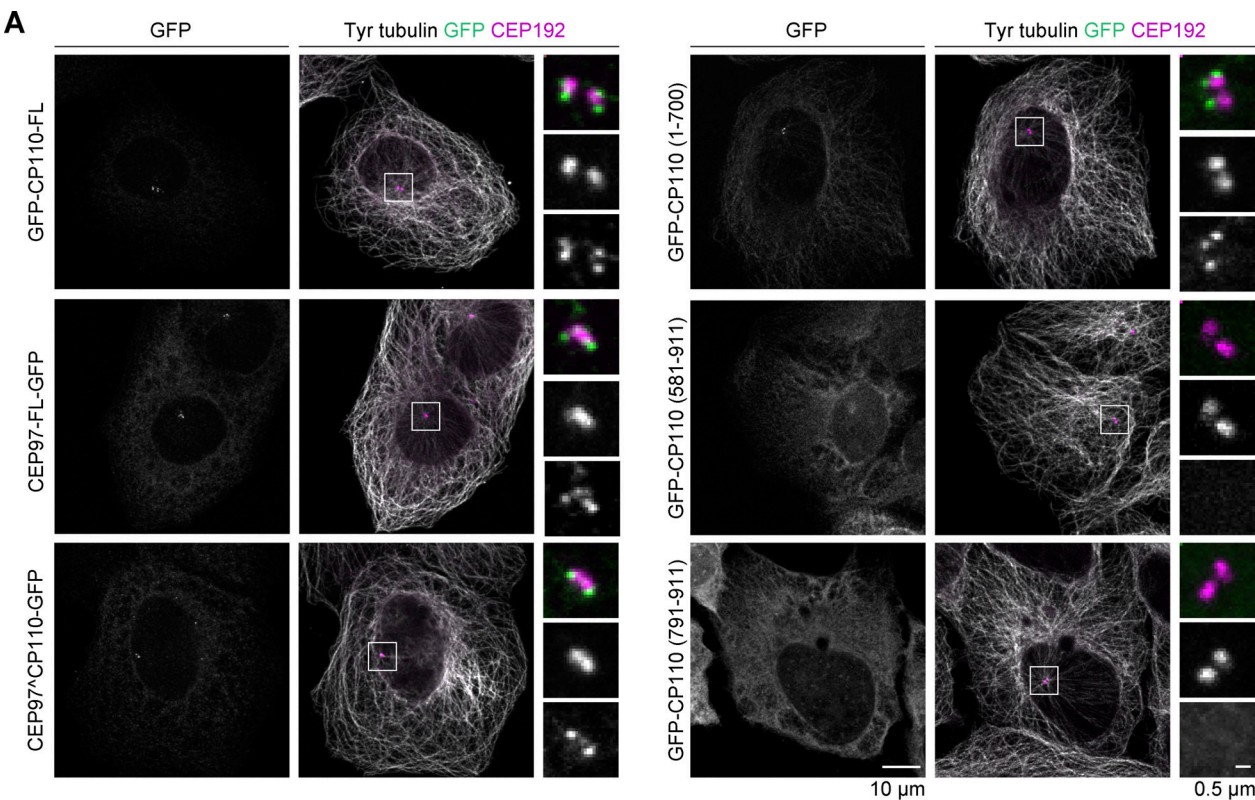

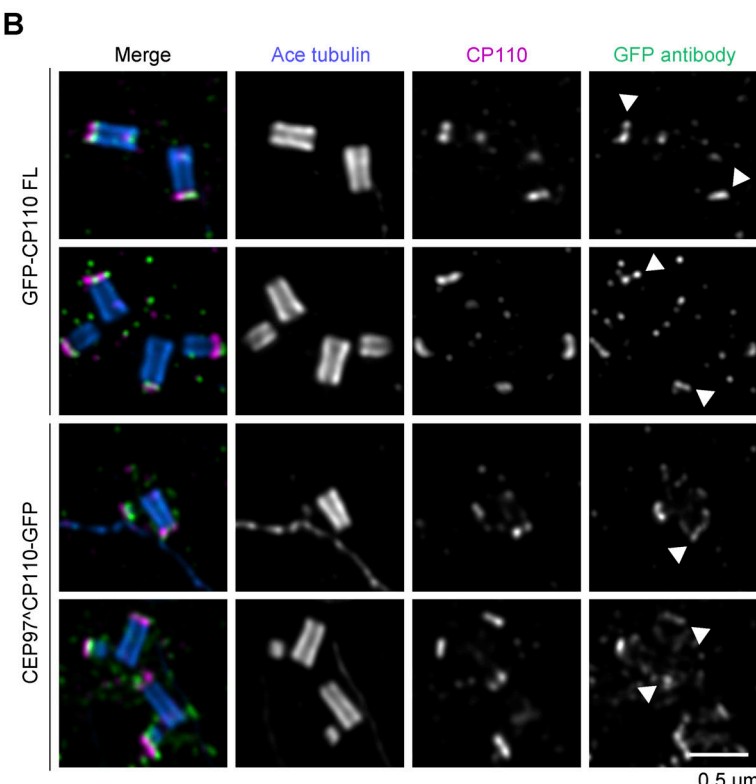

**Figure S2. Subcellular localization of GFP-tagged CP110 and its fragments, CEP97, and CEP97^CP110 chimera. (A)** U2OS transiently transfected with the indicated GFP-tagged constructs were fixed and stained with antibodies against CEP192 (magenta), GFP (green), and tyrosinated tubulin (gray). White box highlights region with centrioles, which are enlarged in zoom. **(B)** U-ExM images of centrioles from U2OS cells overexpressing the indicated constructs and stained for acetylated tubulin (blue), CP110 (magenta), and GFP (green). CP110 full-length and CEP97^CP110 both localize to the distal cap of the mother centriole (white arrowhead) and distal cap of the daughter centriole.



**A**

14 PF, plus-end

14 PF, minus-end

13 PF, unclear polarity

50 nm

**B**

free microtubule plus-ends

free microtubule minus-ends

uncapped plus-ends + CEP97^CP110-GFP

capped plus-ends + CEP97^CP110-GFP

50 nm

**C**

14 PF, plus-end
clockwise twist (84%)

14 PF, minus-end
counter clockwise twist (63%)

13 PF, unclear polarity

25 nm

Figure S3. **Characterization of MT ends by cryo-ET. (A)** Determination of MT polarity. For each MT: sum of slices containing the MT (top) and the same image Fourier filtered at origin (bottom). **(B)** Gallery of MT ends—plus and minus, capped by CEP97^CP110-GFP and uncapped, in the presence of 15 µM tubulin. Scale bar: 50 nm. **(C)** Sum of slices obtained from the tomograms rotated 90° to illustrate the end-on view of PF flares. Plus ends typically show clockwise twist pattern, while minus ends typically show counterclockwise pattern. The twist pattern is also observed for 13-PF MT ends.

## A

1X CPAP-CC1 + 1X CP110-CC2
2X CPAP-CC1 + 1x CP110-CC2
3X CPAP-CC1 + 1x CP110-CC2
1X CPAP-CC1
1X CP110-CC2

F1  F2  F3  F4  F5

## B

Coomassie-stained SDS
PAGE of the fractions F1

i) 1x CPAP-CC1+
1x CP110-CC2

CPAP-CC1
CP110-CC2

ii) 2x CPAP-CC1+
1x CP110-CC2

CPAP-CC1
CP110-CC2

iii) 3x CPAP-CC1+
1x CP110-CC2

CPAP-CC1
CP110-CC2

F1 F2 F3 F4 F5

## C

I
SigI
Fit

## D

CP110-CC2 ab initio model

~135 Å

90°

~30 Å

## E

| | CPAP-CC1 | CP110-CC2 | Mix |
|---|---|---|---|
| Predicted molecular mass (kDa) | 12.5 | 10.0 | - |
| Determined molecular mass by MALS (kDa) | 13.0 ± 1.8 | 17.5 ± 1.0 | 20.0 ± 1.1 |
| Determined molecular mass by SAXS (kDa) | - | 18 | 20 |
| SEC elution volume (ml) | 15.1 | 14.5 | 14.5 |
| $T_m$ determined by CD at 222 nm (°C) | - | 48 | 48 |
| $D_{max}$ determined by SAXS (nm) | - | 12.5 | 12.5 |
| $R_g$ determined by SAXS (nm) | - | 3.5 | 3.5 |

## F

CP110-CC2(R656A/L659A)

## G

CP110-CC2(R656A/L659A)

Figure S4. **Biophysical characterization of CPAP-CC1, CP110-CC2, and CPAP-CC1/CP110-CC2 complex. (A)** SEC-MALS analyses of CPAP-CC1 (magenta lines) and CP110-CC2 (green lines) alone, and mixtures of CPAP-CC1 with CP110-CC2 at molar ratios of 1:1 (black line), 2:1 (light blue line), and 3:1 (dark blue line). **(B)** Coomassie-stained SDS-PAGE of the fractions F1–F5 indicated in panel A and collected from SEC-MALS runs obtained with mixtures of CPAP-CC1 and CP110-CC2. SDS-PAGE analysis of the elution peak fractions centered at around 14.3 ml (corresponding to the molecular weight of CPAP-CC1/CP110-CC2 heterodimer) of the various mixtures revealed equally intense protein bands corresponding to CPAP-CC1 and CP110-CC2. **(C and D)** SAXS analysis of the CP110-CC2 homodimer. **(C)** Solution X-ray scattering intensity over scattering angle from CP110-CC2. The fit to the data yielding the interatomic distance distribution is shown with a black line. **(D)** Surface representation of the X-ray scattering volume of CP110-CC2, at 30 ± 2 Å estimated precision, derived from averaging 22 particle models calculated by ab initio fit to the scattering data. **(E)** Table summarizing biophysical parameters of CPAP-CC1, CP110-CC2, and an equimolar mixture of CPAP-CC1 and CP110-CC2 obtained by SEC-MALS, CD, and SAXS. **(F and G)** CD spectrum (F) recorded at 15°C and thermal-unfolding profiles (G) recorded by CD at 222 nm of CP110-CC2 R656A/L659A (light green dashed lines). Source data are available for this figure: SourceData FS4.

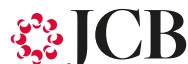

Figure S5. **Generation and characterization of stable cell lines expressing WT or mutant GFP-CPAP. (A)** Sequencing results of the genomic mutation using gel-purified PCR products. **(B)** Western blots illustrating that the Flp-In–induced protein expression system has a low level of leaky expression, where CPAP endogenous (endo) is compared with CPAP overexpression (OE). **(C)** Western blots illustrating the CPAP expression levels in control, host, and different GFP-CPAP-FL$_{WT}$ and GFP-CPAP-FL$_{MUT}$ cells lines without doxycycline induction. **(D)** Immunofluorescence images taken with Airyscan 2 confocal microscope of centrioles of cells blocked for 24 h in mitosis with S-trityl-L-cysteine (STLC) and stained for the acetylated tubulin (blue), CP110 (green), and GFP-CPAP (magenta). **(E)** Median ± IQR of centriole length in mitotically blocked cells by STLC, measured as in Fig. 8 F. Number of analyzed mother centrioles (MC) and daughter centrioles (DC): control cells, $n = 74$ MC, 53 DC; host, $n = 71$ MC, 44 DC; CPAP-FL$_{WT#3}$, $n = 66$ MC, 69 DC, and CPAP-FL$_{MUT#1}$, $n = 50$ MC, 40 DC; and nonsignificant (ns) calculated using Kruskal–Wallis ANOVA test. **(F and G)** U-ExM images of centrioles from host, CPAP-FL$_{WT#4}$, and CPAP-FL$_{MUT#5}$ cells blocked in G1/S and stained for acetylated tubulin (blue) combined with CP110 (green) in F and CPAP (magenta) in G. **(F)** Normal centrioles from host and CPAP-FL$_{WT#4}$ and incomplete centriole from CPAP-FL$_{MUT#5}$. **(G)** Normal centrioles from host and CPAP-FL$_{WT#4}$ and incomplete centriole from CPAP-FL$_{MUT#5}$. Scale bar is corrected for ~4.5 expansion factor. Source data are available for this figure: SourceData FS5.

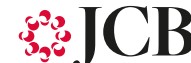

**Figure S6.** **Schematic flowchart illustrating the pipeline for 3D reconstruction, denoising, segmentation, and visualization of tomographic volumes, related to Materials and methods.**

**Video 1.** **Dynamic MT is blocked at the plus end upon flow in of CEP97^CP110-GFP.** Time-lapse movie acquired with a TIRF microscope showing in vitro–reconstituted MTs growing from GMPCPP-stabilized seeds (blue) in the presence of 15 μM tubulin and 20 nM mCherry-EB3 (gray). MT plus end is on the right side of the movie. After 5 min, CEP97^CP110-GFP (green) is flowed into the chamber along with 15 μM tubulin and 20 nM mCherry-EB3 (gray) and blocks plus-end growth. The movie is displayed at 20 frames per second. Sequential triple-color acquisition was done at an interval of 3 s over the course of 10 min. Scale bar, 2 μm.

**Video 2.** **3D view of MT plus ends in the presence of CEP97^CP110-GFP.** The video shows a MT plus end in the presence of CEP97^CP110-GFP. The denoised densities were segmented into tubulin and MTs (blue) and all other densities (green) as described in Materials and methods. Manually segmented models with coordinates of tubulin PFs for each of the plus ends are shown in orange.

**Video 3.** **Dynamic MT growing slowly in presence of CEP97^CP110-GFP and CPAP-N$_{WT}$-mCh.** TIRF microscopy movie showing in vitro–reconstituted MTs growing from GMPCPP-stabilized seeds (magenta) in presence of 15 μM tubulin (gray), 20 nM CEP97^CP110-GFP (green), and 50 nM CPAP-N$_{WT}$-mCh (magenta). The plus end, which is on the right side, is growing slowly (gray) with CEP97^CP110-GFP tracking the growing plus end (green). CPAP-N$_{WT}$-mCh is not visible on the slow growing plus ends because of the bright seed in the same channel. The movie is displayed at 20 frames per second. Sequential triple-color acquisition was done at an interval of 3 s over the course of 10 min. Scale bar, 2 μm.

