## [Peer Review File · The Journal of Cell Biology]

Centriolar cap proteins CP110 and CPAP control slow elongation of microtubule plus ends

Saishree Iyer, Fangrui Chen, Funso Ogunmolu, Shoeib Moradi, Vladimir Volkov, Emma van Grinsven, Chris van Hoorn, Jingchao Wu, Nemo Andrea, Shasha Hua, Kai Jiang, Ioannis Vakonakis, Mia Potočnjak, Franz Herzog, Benoît Gigant, Nikita Gudimchuk, Kelly Stecker, Marileen Dogterom, Michel Steinmetz, and Anna Akhmanova

Corresponding Author(s): Anna Akhmanova, Utrecht University

Review Timeline:

Submission Date:	2024-06-12
Editorial Decision:	2024-07-26
Revision Received:	2024-10-24
Editorial Decision:	2024-11-18
Revision Received:	2024-12-01

Monitoring Editor: Alexey Khodjakov

Scientific Editor: Tim Fessenden

Transaction Report:

DOI: <https://doi.org/10.1083/jcb.202406061>

July 26, 2024

Re: JCB manuscript #202406061

Dr. Anna Akhmanova
Utrecht University
Biology
Cell Biology, Neurobiology and Biophysics, Department of Biology, Faculty of Science Utrecht University Padualaan 8
Utrecht 3584 CH
Netherlands

Dear Dr. Akhmanova,

Thank you for submitting your manuscript entitled "Centriolar cap proteins CP110 and CPAP control slow elongation of microtubule plus ends". The manuscript was assessed by expert reviewers, whose comments are appended to this letter. We invite you to submit a revision if you can address the reviewers' key concerns, as outlined here.

As you will see, reviewers commended the combination of structural and functional insights towards resolving a longstanding and important question of centriolar growth regulated by CP110 and CEP97. All reviewers make important suggestions to clarify the claims made as well as requests to include details on methods. Reviewer 2 made additional requests, in some cases without concisely stipulating the new data they feel are needed. Nonetheless, we agree that a revised manuscript should include, if possible, additional observations with full length CP110 as well as a rationale for the truncation used. Reviewers 2 and 3 both felt the presentation and interpretation of data in Figure 4 should improve. We agree and encourage you to furnish additional details, however you see fit, on precisely how CPAP interacts with CEP97-CP110 at microtubule ends to accomplish their slow growth.

GENERAL GUIDELINES:

Text limits: Character count for an Article is < 40,000, not including spaces. Count includes title page, abstract, introduction, results, discussion, and acknowledgments. Count does not include materials and methods, figure legends, references, tables, or supplemental legends.

Figures: Articles may have up to 10 main text figures. Figures must be prepared according to the policies outlined in our Instructions to Authors, under Data Presentation, <https://jcb.rupress.org/site/misc/ifora.xhtml>. All figures in accepted manuscripts will be screened prior to publication.

*****IMPORTANT:** It is JCB policy that if requested, original data images must be made available. Failure to provide original images upon request will result in unavoidable delays in publication. Please ensure that you have access to all original microscopy and blot data images before submitting your revision. ***

Supplemental information: There are strict limits on the allowable amount of supplemental data. Articles may have up to 5 supplemental figures. Up to 10 supplemental videos or flash animations are allowed. A summary of all supplemental material should appear at the end of the Materials and methods section.

Please note that JCB now requires authors to submit Source Data used to generate figures containing gels and Western blots with all revised manuscripts. This Source Data consists of fully uncropped and unprocessed images for each gel/blot displayed in the main and supplemental figures. Since your paper includes cropped gel and/or blot images, please be sure to provide one Source Data file for each figure that contains gels and/or blots along with your revised manuscript files. File names for Source Data figures should be alphanumeric without any spaces or special characters (i.e., SourceDataF#, where F# refers to the associated main figure number or SourceDataFS# for those associated with Supplementary figures). The lanes of the gels/blots should be labeled as they are in the associated figure, the place where cropping was applied should be marked (with a box), and molecular weight/size standards should be labeled wherever possible.

The typical timeframe for revisions is three to four months. While most universities and institutes have reopened labs and allowed researchers to begin working at nearly pre-pandemic levels, we at JCB realize that the lingering effects of the COVID-19 pandemic may still be impacting some aspects of your work, including the acquisition of equipment and reagents. Therefore, if you anticipate any difficulties in meeting this aforementioned revision time limit, please contact us and we can work with you to find an appropriate time frame for resubmission. Please note that papers are generally considered through only one revision cycle, so any revised manuscript will likely be either accepted or rejected.

Thank you for this interesting contribution to Journal of Cell Biology. You can contact us at the journal office with any questions at cellbio@rockefeller.edu.

Sincerely,

Alexey Khodjakov
Monitoring Editor
Journal of Cell Biology

Tim Fessenden
Scientific Editor
Journal of Cell Biology

Reviewer #1 (Comments to the Authors (Required)):

This is a remarkable paper, pulling together an impressive range of methods to tackle an interesting problem in centriole biogenesis: how is slow microtubule (MT) growth achieved. Most of the data presented have been obtained from an in vitro reconstitution approach, and some come from observations on engineered cells. In sum, the data presented make a strong case for the basic conclusion of this paper, that centriole MT elongation is controlled by the interaction of a growth-promoting protein (CPAP) with a growth inhibiting protein (CP110) at or near the growing MT end, where tubulin protofilaments (PFs) flare out from the MT axis. The paper is so complex and dense that it is hard to read, but it comes across as a reliable and important contribution to the literature on centriole growth. There are several additional issues raised and discussed during the course of the paper, such as the shape of tubulin protofilaments (PFs), which is one reason it is hard to read. Elimination of those extras would make for a trimmer paper, but those issues are all contributions, making the paper, over-all, an excellent piece of work, suitable for publication after attention to some details. My specific comments are listed below.

Abstract

In 4, elongation: (there should be a colon, not a comma)

Results

P5, para 2 The rationale for the chimera used to obtain soluble CP110 is not well explained. The fact of interaction seems a vague criterion for choosing to make the chimera that was used. Were other things tried, such as Maltose binding protein? The assumption that this chimera is producing wild type activity is not a favorable feature of this paper, and it is never discussed.

P5 In -2. I don't know what "average duration of ~0.28-0.76 mins" means. How many observations? What do you state a range for an average?

P6 In 1 Were there any efforts to establish how "similar" the binding of CP110 alone is to that of the chimera? This important point is simply glossed over. (Note it should read "similarly")

Fig. 1C In this image, the rates of MT growth strongly imply that the image is constructed with the seed's plus end pointing to the right. In other images in other figures, the +/- orientation is far from obvious, and is again unstated. I infer that all images are made with + ends on the right side of the seed, but this issue should be stated explicitly.

Fig. 11,J This single figure is convincing, but the paper should include some indication of how many times the experiment was

done, and there should be some evidence to show that photobleaching wasn't simply killing the system through radiation damage.

Fig. S1 A,B The proteins purified from HEK cells are not impressively poor. The mass spec. assays identify what are probably the major contaminants. The fact that nothing is said about these components suggests that the authors presume that readers will surmise that the contaminants have no function in the assays performed. This opinion should be stated explicitly.

Fig. 1 is convincing about the blocking effect of CH110 and the chimera. However, I had trouble relating Fig. S1D to your descriptions of CEP97. Can this figure be labeled or better explained?

Figs. 2 and S2. The denoised images of MT ends are beautiful, and the analysis of PF curvature is convincing. There seems to be an error in the labeling of the ordinate in 2F: The axis is displaying PF length, but the units are "deg/dimer." Should be nanometers? The description of the "caps", however, leaves something to be desired. The material associated with the MT ends looks like an amorous mesh. The information on the molecular structure of parts of this chimera from CD spectroscopy is hard to relate to the fuzz seen in by CryoET. Some structural analysis of this chimera and related proteins by EM might help to bridge this gap. Shadow images of material freeze-dried out of glycerol or perhaps negative straining might allow the reader to relate the "caps" to the structure of the protein(s) that are altering polymerization dynamics. The images shown are very clear, but they don't fit easily into one's experience of other proteins binding to MTs. Moreover, no explanation is offered for the large and variable size of the caps seen by CryoET. Does their size relate to the number of copies of capping proteins involved? How do these large caps relate to the rather small number of copies of the chimeric protein seen in the light microscope? Have you tried taking the molecular mass from biochemistry and the stoichiometry from light microscopy to estimate the amount of material you would expect to see in the CryoEM? The presentation as currently stated leaves me with a feeling of puzzlement about these data, which weakens the impact of the paper. Some thoughtful work on the mass of the caps seen in CryoET and of the EM structure of the protomer could really turn this weakness around.

Fig. S2C. You give handedness of the helical paths of the protofilament curls at each MT end, but you don't state the orientation of viewing, which for planar displays is important.

P5 lower paragraph, middle. The justification for making this chimera, as opposed to finding an alternative way to make CP110 more soluble is not very convincing. You made this chimera and it works to slow growth, but was there some clear logic that let you to it?

Fig S3A The streaky character of the tubulin staining is intriguing. What's going on? In the legend to S3C, the name of the test includes the man's name: Tukey, not Turkey.

P9, para2, ln 1 I don't understand, "The outermost MT plus end". The chimera commonly (but not always) binds to inward-facing surface of bending PFs at the MT tip. Is this what you mean?

Fig. S4 A and B, for the reader's convenience, the legend should again spell out the meaning of CC, PN, MBD, and G-box

P10 last line Define DMTMM

Fig. 4 (legend on p46, ln -7) This statement would be clearer if it read "(white arrow) showing their lack of recovery on a plus end until a rapid growth begins". Fig. 4H. Is it possible to compress the time axis of this graph, so the start of fast recovery for CPAP can be shown on the same recovery graph? It would make a more dramatic display of the phenomena implies in 4G.

P 46 ln -2 I can't see a gray arrow. Do you mean the black-appearing arrow at the very right side of the images?

P12 para 2, ln4 This point is made directly in Fig 4D, left, so you might mention that too.

Fig. S7 legend ln2 You state, "full-length CPAP" but then say "CPAPwt", which is not full length. Which did you use? If not Wild Type, why not?

P14, para 2 ln 9 The descriptions of centriole length are strange. You give 3 significant figures preceded by ~ to indicate approximately, you don't mention the number of centrioles scored, nor do you give a range of lengths for each measurement. Strange.

Discussion and Fig. 7. The case the CP110 binds to the inward-facing surface of tubule is pretty strong in the data presented. The binding shown is characteristically at the wide end of a set of flared PFs at a MT end. It is therefore something of a mystery that Fig. 7 deals only with cylindrical MT walls, failing to represent the data accurately. By not confronting the geometry of the MT tip that is described in this paper and in several publications by others, the modeling seems naïve and misplaced. The current figures fail to display the possibility that CP110 may interfere with PF straightening, which in turn might alter the affinity of the terminal tubulin dimer for dimers in solution, or the like. It would be worth mentioning in discussion that the wad-like images of the CP110 chimera at MT plus ends makes it difficult to assign a specific geometry to the binding and inhibition. The

text mentions a "wedge" model, but the image data look more like simply a binding to the inward-facing surface of the bent PF, interfering, perhaps, with its ability to straighten. I saw no indication that the CP110 chimera fit in between PFs. Modeling would be much easier if the paper conveyed a firm idea of the shape of the hybrid CP110 in solution.

P18 second para. This text suggests by implication how nice it would be to have CryoET images of MTs growing in the presence of CP110 and CPAP. I expect you tried to get them, but without success. The paper includes so much information already that it would be silly to ask for more, but this gap is noticeable.

P20 last para The similarity in network texture between the CryoET images of CP110 and the "carrots" at the tips of flagellar MTs is interesting and might be worth mentioning.

Reviewer #2 (Comments to the Authors (Required)):

The manuscript by Lyer et al. entitled "Centriolar cap proteins CP110 and CPAP control slow elongation of microtubule plus ends" investigate the cooperative activities of CP110 and CPAP in establishing the slow, processive microtubule growth characteristic of the centriole. The authors use advanced in vitro methods to determine how these molecules interact and localize to the growing tips of microtubules. They show that a chimeric Cep97-CP110 protein localizes to microtubule lumens and caps microtubules to promote microtubule stalling. When mixed with a fragment of CPAP, the combined effect causes extremely slow microtubule growth. However, CPAP cannot overcome CP110-mediated stalling when added to pre-"capped" microtubules. In cells, the authors show that replacement of CPAP with a CP110-binding mutant results in shorter centrioles with less CPAP. The authors conclude that regulation of a plug-like complex of CP110 ensures the slow growth of centriolar microtubules.

This manuscript addresses an impactful, key question in the field of centriole assembly (i.e., how centriole microtubules grow?). Overall, the experiments appear to be performed and interpreted well, particularly the in vitro reconstructions. I believe that this manuscript will be a great fit for the Journal of Cell Biology after revision.

Major concerns:

1. Regarding the in vitro experiments, it is unclear how the authors determined purity of their protein. The protein was purified from human cells with a strep tag (the placement is not shown in Figs 1A and 4A) and the stated methods suggest that these proteins would retain endogenous interactions. Figure S1A shows multiple proteins pull down with GFP-CP110. I'm assuming that S1B is mass spectrometry of the entire pull-down mixture instead of bands cut from gels (this should be made clear). However, it is unclear the extent of this list. What are the identities of the bands that are clearly visible on the protein gels? Without knowing if there are contaminating proteins that bind to CP110, Cep97 and CPAP, it is difficult to determine that their effects in the in vitro experiments are specific. Can the authors explain why there is so much tubulin co-purifying in these pull-downs? CP110 isn't expected to bind soluble tubulin heterodimers and the authors show that Cep97 doesn't bind microtubules at all.
2. It is unfortunate that the in vitro aspects of this study could not be conducted with purified full-length CP110, especially since the authors demonstrated in Figs 1B and 1C that it could be possible. Regarding the CEP97^ΔCP110 chimera protein introduced in Figure 1. How were the regions of these two proteins chosen? Why didn't the authors map the site(s) of microtubule binding in CP110 first? Can the authors show that the CP110 NT cannot bind microtubules? The authors excluded any contribution of the CP110 N-terminus to microtubule binding/dynamics. Instead of fusing the C-terminus of CP110 to the N-terminus of Cep97, why not fuse CP110 to Maltose-binding protein or some other large soluble monomeric protein? Using Cep97 as the fusion partner only invites additional co-purifying centriole proteins in the protein purification prep. Again, the rationale for making this particular fusion protein is not at all clear but it should be made clear. The authors state that CEP97^ΔCP110 localizes to centrioles in U2OS cells but do not show the data. This should be shown as well as any effects on centrioles (length, number and morphology)? Does the chimera localize specifically to the distal tips of centrioles? If the chimeric protein localizes normally to centrioles, then shouldn't it retain the ability to bind other centriole proteins, and wouldn't those proteins contaminate the protein purification prep? Lastly, the authors need to show that CEP97^ΔCP110 can bind the CPAP^{wt} protein but not the CPAP^{mut} protein.
3. Results in Figure 4G and conclusions from Figure 4 appear to be incongruent. In the right panel of 4G, the microtubule at the start of the kymograph is capped by Cep97-CP110 and growth appears inhibited. After FRAP, neither protein recovers until new CPAP is recruited, resulting in slow growth. Only after the microtubule pauses, then do we see more Cep97-CP110 recruited to the stalled microtubule. Is the interpretation that the bleached cap was removed when CPAP is recruited, thus triggering microtubule growth and that CEP97^ΔCP110 is recruited to stalled microtubule ends? This appears similar to Figure S6D, where loss of CPAP/CEP97^ΔCP110 precedes microtubule regrowth followed by recruitment of these proteins to the growing microtubule.

Overall, I do not believe that there is enough evidence to conclude that CPAP allows Cep97-CP110 to track the tips of growing microtubules. Rather, it seems equally likely that 1) CPAP is displacing the cap to promote growth, 2) CPAP recruits Cep97-

CP110 when present, or 3) microtubule shape is determining if the cap proteins are recruited.

Do the authors see an increase in pausing and restarting when CPAP is added? Only about 20% of microtubules exhibit this in 20nM CEP97^ΔCP110, although pausing followed by catastrophe does not appear to be separated. On microtubules that display a continuous slow growth rate, do the authors see that fluorescent intensity never recovers? If the authors stabilized microtubule ends with (taxol?), would they see an increased recruitment of Cep97-CP110 to these ends?

Additional concerns:

Page 13 - 3rd line from the top "MTs grew at a rate..." not "MTs that grew at a rate"

Scale appears incorrect in S7H.

Misspelled CP110 as CEP110 in Discussion (pg 20).

Figures 1B, 1D, 4B, 4C, 4D: use different colors for the arrowheads instead of red and green.

Figure 4E: is there a significant difference between CPAPwt and CEP97^ΔCP110+CPAPwt.

Addition of CPAP to microtubules causes the microtubule growth rate to decrease. Why do the authors then refer to CPAP as a microtubule polymerase?

The authors should consider changing the title of the paper because CPAP is not recognized as a centriolar capping protein.

Figure 4F: is the decrease in CEP97^ΔCP110 localization to microtubules with CPAPmut significantly different compared to microtubules with no CPAP? Why would it be less if it can't bind CPAP?

Figures 6C and 6G: what is significant in these graphs and what is not? This is important in order to make any conclusions.

Missing reference. Direct binding between CP110 and Sas4 has been shown with Drosophila protein fragments (Galletta et al., Nat Communications, 2016). Specifically, the N-terminus of CP110 interacts via yeast 2-hybrid with Sas4 NT.

Page 12, paragraph 3, sentence 2: Did the authors mean to say CPAPmut in this sentence?

Page 13, paragraph 1, last sentence: The authors should rephrase their concluding sentence here. Perhaps, "these data indicate that a mixture of CPAP and CP110^ΔCEP97 causes an even slower rate of MT growth compared to CPAP alone". These data do not show that this requires direct interaction because a direct interaction with these new protein constructs has not been shown.

Page 15, paragraph 3, sentences 2-3: from the graph in Figure 6C, it appears that CPAP binding is not important for robust CP110 localization since >90% of CP110 localizes properly and there are no measurements of intensity to say otherwise.

Reviewer #3 (Comments to the Authors (Required)):

Centrioles are small organelles whose distinguishing feature is an outer wall composed of a 9-fold array of stabilized microtubule blades. Remarkably, during centriole assembly, centriolar microtubules elongate very slowly to yield centrioles that are within a reproducible length range. How this is achieved is not well understood. Two microtubule-binding regulators that have been implicated in this process are CPAP, which is thought to promote the growth of centriolar microtubules, and CP110/CEP97, which is involved in capping the ends of centriolar microtubules to limit their extension. In this paper, the authors identify forms of these proteins that enable characterization in vitro and use a combination of in vitro assays, structural work, and in vivo approaches to dissect the roles of CPAP and CP110 and their collaboration in the growth of centriolar microtubules. They identify a CEP97^ΔCP110 chimera that does not aggregate in vitro and is suitable for in vitro assays. In vitro assays show that this chimera dimerizes, specifically binds to microtubule plus ends, and when recruited sub-stoichiometrically to the number of protofilaments blocks further microtubule elongation. Using cryo-electron tomography, they show that CEP97^ΔCP110 forms a cap or partial cap that sits in the MT lumen. They next go on to examine the interplay between CEP97^ΔCP110 and CPAP. Through a very nice biochemical/structural analysis they identify a coiled-coil region in CPAP (CC1) that interacts in an antiparallel fashion with a coiled-coil region in CP110 and specific mutations that disrupt this interaction. They show that whereas microtubule plus ends are capped when CEP97^ΔCP110 is added, simultaneous addition of CPAP allows these capped ends to slowly grow. By introducing the mutation that prevents the interaction of the CP110 and CPAP coiled-coil regions in vivo, they show that centrioles exhibit structural defects, with one prominent defect being that the centrioles are too short in the presence of the mutant. Overall, the work provides significant insight into key players that control the growth and stability of centriolar microtubules and is appropriate for publication in the JCB. However, in a few cases, which I discuss below, I think that the presentation of the data could be improved to allow the main points to come through more clearly.

Main points:

1. Although Figure 2A-C are quite effective, the points made in Fig. 2D-I are much less clear. Fig. 2F makes the point that protofilament length, defined as the region after the protofilament first bends outwards, is shorter in the microtubules capped by CEP97^ΔCP110 than in controls. First, the y-axis label on this graph does not seem to be correct - why is the graph labeled deg/dimer if this is a measure of protofilament length? Figs. 2H-I are also confusing. For Fig. 2H, the question arises of whether it is fair to compare average curvature of the whole protofilament extension of the capped and non-capped ends, given that the extensions are almost twice as long in the uncapped compared to the capped cases. It is mentally hard to figure out what this comparison means. It might be better just to show the graph currently in 2G. However, the issue with the graph in Fig. 2G is that it ignores the fact that most of the protofilament extensions are less than 15 nm long in the capped case compared to more than 15 nm long in the uncapped case. For short distances, like 5 nm, it does seem like the capped microtubule protofilaments show less curvature than the protofilaments in the uncapped microtubules. Since protofilament length is only measured after the point where the protofilament first bends away from the MT cylinder, is this analysis biased due to the fact that it ignores protofilaments that don't bend away from the MT cylinder? What if all protofilament lengths and curvatures were measured from the point when the first protofilament bent away from the MT cylinder-would a much higher percentage of the other protofilaments exhibit reduced curvature for longer in the capped than uncapped cases?

2. The presentation of the data in Fig. 4E/F is not very effective at conveying the main points. The left graph should potentially be labeled "Mean growth rate". The right graph is harder to understand. The left axis is labeled % of time spent by MT plus end. However, this seems to be a cumulative distribution plot, in which what is plotted is actually the % of time that the microtubule end spent going at the indicated rate or slower. It might be worth thinking about the best way to present this data, as I am not sure that the general readers can easily grasp what is being shown here and what the main point is. The key takeaway from this is that CPAP prevents CEP97^ΔCP110 from irreversibly capping MT plus ends-converting these ends to ends that are instead slowly growing. It would be nice if there could be a graph of a summary statistic that makes this point more directly in the main figure. For example, a plot of mean growth rate of microtubule ends with associated CEP97^ΔCP110, which presumably is undetectable (given the assay limitations) in the absence of CPAP and then a low positive rate in the presence of either form of CPAP.

The second point made by these panels is that the CPAP mutation that prevents CEP97^ΔCP110 binding reduces the amount of time that CEP97^ΔCP110 spends at microtubule ends, so the conclusion here would be that when CPAP is present it would suppress the binding of CEP97^ΔCP110 to microtubule ends if not for its CC1 interaction region, which instead allows CPAP to increase the number of microtubule ends with CEP97^ΔCP110 present.

To better convey these points to the reader, it would be good to put F before E to highlight point 2-that CPAP promotes the association of CEP97^ΔCP110 in a fashion dependent on the identified coiled-coil region first (making the point that its presence would inhibit CEP97^ΔCP110 if it were not for the coiled-coil interaction region). This will provide a really nice transition from the data in Figures 2 and 3. After this, have a panel that provides statistics for mean growth rates when the CEP97^ΔCP110 cap is present or not for the different conditions-maybe make one graph for CEP97^ΔCP110 cap present and one graph for CEP97^ΔCP110 cap absent so that the main point comes through-when microtubules have a CEP97^ΔCP110 cap, the presence of CPAP allows them to continue to grow at a slow rate.

3. The in vivo analysis is nicely done. The main point of this is that centrioles are shorter in the presence of the mutant protein than in the presence of the WT protein. However, it was a bit unsatisfying in the sense that the paper did not do a good job of bringing things together to explain (or suggest an explanation for) why this could/would be the main phenotype observed.

JCB manuscript #202406061

Response to Reviewers

Reviewer 1

This is a remarkable paper, pulling together an impressive range of methods to tackle an interesting problem in centriole biogenesis: how is slow microtubule (MT) growth achieved. Most of the data presented have been obtained from an in vitro reconstitution approach, and some come from observations on engineered cells. In sum, the data presented make a strong case for the basic conclusion of this paper, that centriole MT elongation is controlled by the interaction of a growth-promoting protein (CPAP) with a growth inhibiting protein (CP110) at or near the growing MT end, where tubulin protofilaments (PFs) flare out from the MT axis. The paper is so complex and dense that it is hard to read, but it comes across as a reliable and important contribution to the literature on centriole growth. There are several additional issues raised and discussed during the course of the paper, such as the shape of tubulin protofilaments (PFs), which is one reason it is hard to read. Elimination of those extras would make for a trimmer paper, but those issues are all contributions, making the paper, over-all, an excellent piece of work, suitable for publication after attention to some details.

We thank the reviewer for the positive and thoughtful comments on our manuscript.

Specific comments:

1. Abstract

In 4, elongation: (there should be a colon, not a comma)

Thank you for pointing this out. We have corrected this. The revised text in abstract line 4 reads as: “the proteins that cap distal centriole ends and control their elongation: CP110, CEP97 and CPAP/SAS-4.”

2. Results

P5, para 2 The rationale for the chimera used to obtain soluble CP110 is not well explained. The fact of interaction seems a vague criterion for choosing to make the chimera that was used. Were other things tried, such as Maltose binding protein? The assumption that this chimera is producing wild type activity is not a favorable feature of this paper, and it is never discussed.

We have now explained our reasoning for making the chimeric CEP97-CP110 protein better (page 5 of the revised paper, see below) and added data showing the localization of CP110 fragments and the CEP97[^]CP110 chimera in cells (new Figure S2). We note that fusing of a small soluble protein, GFP, to C-terminal fragments of CP110 has not made them functional in vitro, and we have not tested any fusions with other unrelated proteins. Instead, we generated an array of CEP97[^]CP110 fusions and investigated their ability to localize to the distal centriole end in cells, because this provided an easy functional test. We chose the best-behaved one, which we used for subsequent experiments. The localization of this fusion protein to distal centriole ends by Expansion Microscopy is now illustrated in the new Fig. S2B.

“We next generated different GFP-tagged CP110 deletion mutants. Since the N-terminus of CP110 is known to bind to the middle part of CEP97 (Spektor et al., 2007), we hypothesized that the C-terminus of CP110 is responsible for blocking MT growth. When expressed in cells,

full-length CP110 and CEP97, as well as the N-terminal CP110 fragment 1-700 localized to centrosomes. However, GFP-tagged C-terminal CP110 fragments 581-991 and 791-991 did not associate with either centrosomes or microtubules (Fig.S2A), and we were not successful in purifying them from HEK293 cells. Since CP110 normally functions in association with CEP97, we reasoned that a direct fusion with CEP97 might improve the folding or stability of the CP110. We screened different fusion constructs by their localization in U2OS cells and found that a protein containing residues 1-650 of CEP97 and residues 581-991 of CP110 (termed here CEP97^CP110, Fig. 1A) localized to centrosomes (Fig. S2A). Ultrastructure expansion microscopy (U-ExM), which entails 4.5x expansion of cells, followed by confocal imaging (Gambarotto et al., 2019) demonstrated that CEP97^CP110-GFP bound to the distal centriole ends, similar to CP110 (Fig. S2B). In vitro, SII and GFP-tagged CEP97^CP110 blocked MT seed elongation and was much less aggregation-prone than full-length CP110 (Fig. 1D-F, Video S1)."

Figure S2:

3. *P5 In -2. I don't know what "average duration of ~0.28-0.76 mins" means. How many observations? What do you state a range for an average?*

The statistics for quantifications of pause duration is reported in the figure legend of Figure 1G, where these pause durations are plotted. We state a range for the mean because the average pause durations are different at different concentrations of CEP97^ΔCP110 in the assay. At 2 nM, the average pause duration is 0.28 min and at 40 nM, it is 0.76 min.

4. *P6 In 1 Were there any efforts to establish how "similar" the binding of CP110 alone is to that of the chimera? This important point is simply glossed over. (Note it should read "similarly")*

A quantification of the percentage of blocked microtubule ends for one experimental series with GFP-CP110 is shown in Fig. S1C. A more detailed quantification of these assays (the percentage of microtubules that are growing, pausing or blocked) with increasing concentrations of GFP-CP110 is shown below. The titration of GFP-CP110 doesn't give linear results because GFP-CP110 aggregates in the assays, reducing the consistency of the results. However, both GFP-CP110 and CEP97^ΔCP110-GFP consistently induce transient pauses followed by catastrophes at low concentrations and block microtubule plus ends at the seed at higher concentrations, without affecting microtubule minus ends. Their effects on microtubule dynamics are thus similar, as stated.

5. *Fig. 1C In this image, the rates of MT growth strongly imply that the image is constructed with the seed's plus end pointing to the right. In other images in other figures, the +/- orientation is far from obvious, and is again unstated. I infer that all images are made with + ends on the right side of the seed, but this issue should be stated explicitly.*

We apologize for this omission. We have now indicated the plus end of dynamic microtubules in all kymographs with a "+" and, where the dynamic minus end is visible, also with a "-".

6. *Fig. 1I,J This single figure is convincing, but the paper should include some indication of how many times the experiment was done, and there should be some evidence to show that photobleaching wasn't simply killing the system through radiation damage.*

The experiment was performed independently three times, and this is indicated in the legend to Figure 1J. The laser power used in Figure 1 I, J and the Figure 7A, B of the revised manuscript are comparable. In Figure 7A, the kymograph shows that the microtubule keeps growing after photobleaching, and new molecules of CPAP-N_{WT} can bind to the plus end. This shows that the laser power is not high enough to kill the system, but just leads to bleaching of the molecules at the plus end.

7. *Fig. S1 A,B The proteins purified from HEK cells are not impressively poor. The mass spec. assays identify what are probably the major contaminants. The fact that nothing is said about these components suggests that the authors presume that readers will surmise that the contaminants have no function in the assays performed. This opinion should be stated explicitly.*

We have now discussed the contaminants in our protein preparations on page 5 of the revised manuscript: “The main contaminants detected by mass spectrometry (Fig. S1B) were tubulins and the heat shock protein Hsp70, which we often observe in our protein preparations and which to our knowledge have no effect in MT dynamics (van den Berg et al., 2023)”.

8. *Fig. 1 is convincing about the blocking effect of CH110 and the chimera. However, I had trouble relating Fig. S1D to your descriptions of CEP97. Can this figure be labelled or better explained?*

Figure S1D shows the absence of green signal of CEP97-GFP on microtubules. Since the green signal is absent, there is nothing to label.

9. *Figs. 2 and S2. The denoised images of MT ends are beautiful, and the analysis of PF curvature is convincing. There seems to be an error in the labeling of the ordinate in 2F: The axis is displaying PF length, but the units are "deg/dimer." Should it be nanometers?*

We apologize for the mistake. We have updated the figure with the correct axis label, which should indeed be in nanometers.

10. *The description of the "caps", however, leaves something to be desired. The material associated with the MT ends looks like an amorous mesh. The information on the molecular structure of parts of this chimera from CD spectroscopy is hard to relate to the fuzz seen in by CryoET. Some structural analysis of this chimera and related proteins by EM might help to bridge this gap. Shadow images of material freeze-dried out of glycerol or perhaps negative straining might allow the reader to relate the "caps" to the structure of the protein(s) that are altering polymerization dynamics. The images shown are very clear, but they don't fit easily into one's experience of other proteins binding to MTs. Moreover, no explanation is offered for the large and variable size of the caps seen by CryoET. Does their size relate to the number of copies of capping proteins involved? How do these large caps relate to the rather small number of copies of the chimeric protein seen in the light microscope? Have you tried taking the molecular mass from biochemistry and the stoichiometry from light microscopy to estimate the amount of material you would expect to see in the CryoEM? The presentation as currently stated leaves me with a feeling of puzzlement about these data, which weakens the impact of the paper. Some thoughtful work on the mass of the caps seen in CryoET and of the EM structure of the protomer could really turn this weakness around.*

Based on structural predictions, the CEP97[^]CP110 contains alpha-helical domains and unstructured regions, but no large globular or long helical domains, so its structural analysis will be challenging and would be a separate project. For a protein with a large part of its sequence disordered, we find it hard to predict the volume one molecule occupies in tomographic volumes. Furthermore, as indicated in the text on page 8, large amorphous caps were observed when CEP97[^]CP110 was added after the tubulin mix was subjected to high-speed centrifugation, and these caps very likely represent multiple copies of the protein.

When the protein was added before centrifugation, the caps were smaller. Therefore, fully capped microtubules in our assays likely carry many more copies of CEP97[^]CP110 than determined by our TIRF assays (Fig. 1H), which were performed after centrifugation of the tubulin-CEP97[^]CP110 mix. Another key difference between our sample preparations for light microscopy and cryo-ET is the process of grid blotting that is present in the latter and absent in the former. It is possible that during the 3-4 s that we used for blotting the CEP97[^]CP110 chimera might have been recruited to some pre-formed plus-end caps due to quickly decreasing sample volume.

We agree that our Cryo-ET data do not provide structural insight into the CP110-microtubule interaction, but we think that they are still very useful, because they support the notion that the binding of CEP97[^]CP110 is plus-end-specific and demonstrate that the protein interacts with protofilaments from the luminal side and makes them blunter.

11. *Fig. S2C. You give handedness of the helical paths of the protofilament curls at each MT end, but you don't state the orientation of viewing, which for planar displays is important.*

The orientation of viewing in this figure (now Fig. S3C) is looking into the microtubule from the end (end-on view, as stated in the legend).

12. *P5 lower paragraph, middle. The justification for making this chimera, as opposed to finding an alternative way to make CP110 more soluble is not very convincing. You made this chimera and it works to slow growth, but was there some clear logic that let you to it?*

The logic of making the chimera is now explained better on page 5; please see our answers to comment 2 above.

13. *Fig S3A The streaky character of the tubulin staining is intriguing. What's going on?*

In the upper panel of the former Fig S3A (Fig.3A of the revised manuscript), the seeds gradually depolymerize, and the fluorescent signal is getting narrower. In the bottom panel, soluble tubulin shows some noise and some incorporation in the seeds. As in all other kymographs, fluorescent tubulin signal is irregular, because only a small percentage of tubulin subunits used to generate the seeds or dynamic lattices is fluorescently labeled.

14. *In the legend to S3C, the name of the test includes the man's name: Tukey, not Turkey.*

We apologize for the mistake. We have corrected the legend which now reads: “** P < 0.01; ns – non-significant with one-way ANOVA followed by Tukey’s multiple comparison test...”

15. *P9, para2, In 1 I don't understand, "The outermost MT plus end". The chimera commonly (but not always) binds to inward-facing surface of bending PFs at the MT tip. Is this what you mean?*

Thank you for the excellent suggestion, we have changed the description to “inward-facing PF surface at the MT tip”.

16. *Fig. S4 A and B, for the reader's convenience, the legend should again spell out the meaning of CC, PN, MBD, and G-box*

We have updated the legend of Fig 4A and B to explain CPAP domains. The legend reads as:

“For CP110, CC1 and CC2 are the coiled coil domains. For CPAP, CC1 and CC2 are coiled coil domains; PN2-3, the tubulin-binding domain (Cormier et al., 2009); MBD, the MT-binding

domain; G-box, glycine-rich C-terminal domain forming an antiparallel β -sheet (Hatzopoulos et al., 2013)."

We note that the origin of the name of PN2-3 domain is not an acronym but just a name of deletion mutant.

17. P10 last line Define DMTMM

We have updated the text, which now reads as follows: "We used DMTMM (4-(4,6-dimethoxy-1,3,5-triazin-2-yl)-4-methylmorpholinium chloride), which introduces 'zero-length' crosslinks..." Since we had to shorten the text of the article to adhere to the strict length limitations of the J Cell Biol, the details of cross-linking experiment were moved to the Methods section.

18. Fig. 4 (legend on p46, ln -7) This statement would be clearer if it read "(white arrow) showing their lack of recovery on a plus end until a rapid growth begins".

We included in the revised paper two better examples of FRAP on slowly growing MT plus ends (Figure 7A bottom panel of the revised paper. The legend now reads: "Bottom panel: kymographs showing bleaching of both CEP97^ΔCP110-GFP (green) and CPAP-N_{WT}-mCherry (magenta) with a 488-nm laser (white arrow) illustrating that CEP97^ΔCP110-GFP does not recover and CPAP-N_{WT}-mCh recovers slowly."

19. Fig. 4H. Is it possible to compress the time axis of this graph, so the start of fast recovery for CPAP can be shown on the same recovery graph? It would make a more dramatic display of the phenomena implies in 4G.

We have added a new graph to the figure (which became Fig. 7B in the new version of the manuscript), which zooms into the time axis and shows the recovery of CPAP-N_{WT} alone and in combination with CEP97^ΔCP110.

20. *P 46 In -2 I can't see a gray arrow. Do you mean the black-appearing arrow at the very right side of the images?*

The multiple arrows in this figure panel (Fig. 7D of the revised paper) were confusing, and we removed them. Now the figure just has arrowheads showing the localization of the proteins and a black arrow on the right that shows the timing of flow-in. We have modified the legend accordingly.

21. *P12 para 2, In4 This point is made directly in Fig 4D, left, so you might mention that too.*

Since there were some comments about Figure 4, this figure has been modified; since the manuscript was strongly edited to reduce the number of supplements and increase the number of main figures in accordance with J Cell Biol guidelines, the data that were present in Figure 4 are now distributed between Figures 6 and 7.

22. *Fig. S7 legend In2 You state, "full-length CPAP" but then say "CPAPwt", which is not full length. Which did you use? If not Wild Type, why not?*

Thank you for pointing this out. We have modified the nomenclature to eliminate the confusion. The construct we referred to in Fig. S7 legend is now called "CPAP-FL_{WT}" and it is the full length, wild type CPAP. The construct used for the in vitro reconstitutions is called "CPAP-N_{WT}" and it is the first 607 amino acids of CPAP, fused to a GCN4 leucine zipper, a mCherry tag and a SII tag.

23. *P14, para 2 In 9 The descriptions of centriole length are strange. You give 3 significant figures preceded by ~ to indicate approximately, you don't mention the number of centrioles scored, nor do you give a range of lengths for each measurement. Strange.*

From this and previous comments of this reviewer we understand that they would prefer to have all statistics included in the main text. However, we strongly prefer to describe the statistics in figure legends, because it makes the text more readable and facilitates staying within the strict length limits of a J Cell Biol article. In this case, statistics are described in the legends to the Figure 8D,F of the revised article.

24. *Discussion and Fig. 7. The case the CP110 binds to the inward-facing surface of tubule is pretty strong in the data presented. The binding shown is characteristically at the wide end of a set of flared PFs at a MT end. It is therefore something of a mystery that Fig. 7 deals only with cylindrical MT walls, failing to represent the data accurately. By not confronting the geometry of the MT tip that is described in this paper and in several publications by others, the modeling seems naïve and misplaced. The current figures fail to display the possibility that CP110 may interfere with PF straightening, which in turn might alter the affinity of the terminal tubulin dimer for dimers in solution, or the like. It would be worth mentioning in discussion that the wad-like images of the CP110 chimera at MT plus ends makes it difficult to assign a specific geometry to the binding and inhibition. The text mentions a "wedge" model, but the image data look more like simply a binding to the inward-facing surface of the bent PF, interfering,*

perhaps, with its ability to straighten. I saw no indication that the CP110 chimera fit in between PFs. Modeling would be much easier if the paper conveyed a firm idea of the shape of the hybrid CP110 in solution.

We fully agree with the reviewer that better structural EM data would make for a much better molecular model. Unfortunately, we are currently not able to obtain such data, and therefore, we do not know whether CP110 acts by preventing protofilament straightening or by occluding some protofilament surfaces, causing wedging or preventing incoming tubulin dimers from binding. We have changed the model to show that the terminal protofilaments are bent and removed the discussion of the wedge model from the text.

25. *P18 second para. This text suggests by implication how nice it would be to have CryoET images of MTs growing in the presence of CP110 and CPAP. I expect you tried to get them, but without success. The paper includes so much information already that it would be silly to ask for more, but this gap is noticeable.*

For meaningful analysis of Cryo-ET experiments, the majority of the plus- or minus ends of microtubules should be in the same state (e.g., fast growth, pausing, etc). Unfortunately, CEP97^ΔCP110+CPAP-N samples were less uniform, and the obtained results would be much more difficult to interpret. We are currently still searching for conditions that would convert all microtubule ends in the sample with these two proteins into the slow growing state. The analysis of such samples will be the subject of a future paper.

26. *P20 last para The similarity in network texture between the CryoET images of CP110 and the "carrots" at the tips of flagellar MTs is interesting and might be worth mentioning.*

We have mentioned the similarity of cryo-ET images of CP110 and the ciliary tip module proteins in in vitro reconstitution assays, which are shown in our preprint Saunders et al, 2024 (<https://doi.org/10.1101/2024.03.25.586532>), where we have also observed "plugs" at the tips of slowly growing microtubules. Unfortunately, the stringent space limitation of J Cell Biol precludes more extensive citation of the literature on cilia and flagella, which is peripheral to the current story.

Reviewer 2:

The manuscript by Lyer et al. entitled "Centriolar cap proteins CP110 and CPAP control slow elongation of microtubule plus ends" investigate the cooperative activities of CP110 and CPAP in establishing the slow, processive microtubule growth characteristic of the centriole. The authors use advanced in vitro methods to determine how these molecules interact and localize to the growing tips of microtubules. They show that a chimeric Cep97-CP110 protein localizes to microtubule lumens and caps microtubules to promotes microtubule stalling. When mixed with a fragment of CPAP, the combined effect causes extremely slow microtubule growth. However, CPAP cannot overcome CP110-mediated stalling when added to pre-"capped" microtubules. In cells, the authors show that replacement of CPAP with a CP110-binding mutant results in shorter centrioles with less CPAP. The authors conclude that regulation of a plug-like complex of CP110 ensures the slow growth of centriolar microtubules.

This manuscript addresses an impactful, key question in the field of centriole assembly (i.e., how centriole microtubules grow?). Overall, the experiments appear to be performed and interpreted well, particularly the in vitro reconstructions. I believe that this manuscript will be a great fit for the Journal of Cell Biology after revision.

We thank the reviewer for the positive and thoughtful comments on our manuscript.

Major concerns

1. Regarding the in vitro experiments, it is unclear how the authors determined purity of their protein. The protein was purified from human cells with a strep tag (the placement is not shown in Figs 1A and 4A)

We have updated all our protein schemes to show the position of the SII tag used for purification from mammalian cells. This can be seen here:

And the stated methods suggest that these proteins would retain endogenous interactions. Figure S1A shows multiple proteins pull down with GFP-CP110. I'm assuming that S1B is mass spectrometry of the entire pull-down mixture instead of bands cut from gels (this should be made clear). However, it is unclear the extent of this list. What are the identities of the bands that are clearly visible on the protein gels? Without knowing if there are contaminating proteins that bind to CP110, Cep97 and CPAP, it is difficult to determine that their effects in the in vitro experiments are specific. Can the authors explain why there is so much tubulin co-purifying in these pull-downs? CP110 isn't expected to bind soluble tubulin heterodimers, and the authors show that Cep97 doesn't bind microtubules at all.

All the mass spectrometry data were acquired using the whole purified protein preparations and not from bands cut from gels. Since current mass spectrometry approaches are very sensitive, isolation of bands from gel is no longer necessary. During protein purification, the cells are lysed and the isolated proteins washed in a buffer containing 300 mM NaCl. Preserving interactions with endogenous proteins would require much lower ionic strength of the buffer, and indeed the only major contaminants we see in our preparations are the heat shock protein HSP70 and tubulins, which are also visible on gels at 70 and ~50 kDa, respectively. We now mention this explicitly on page 5 of revised manuscript: "The main contaminants detected by mass spectrometry (Fig. S1B) were tubulins and the heat shock protein Hsp70, which we often observe in our protein preparations and which to our knowledge have no effect in MT dynamics (van den Berg et al., 2023)." Tubulin is one of the most abundant proteins in cell lysates, and apparently the proteins we study have a weak affinity for it. Its presence should not perturb our assays which include 15 uM tubulin, anyway.

- 2. It is unfortunate that the in vitro aspects of this study could not be conducted with purified full-length CP110, especially since the authors demonstrated in Figs 1B and 1C that it could be possible. Regarding the CEP97^CP110 chimera protein introduced in Figure 1. How were the regions of these two proteins chosen? Why didn't the authors map the site(s) of microtubule binding in CP110 first? Can the authors show that the CP110 NT cannot bind microtubules? The authors excluded any contribution of the CP110 N-terminus to microtubule binding/dynamics. Instead of fusing the C-terminus of CP110 to the N-terminus of Cep97, why not fuse CP110 to Maltose-binding protein or some other large soluble monomeric protein? Using Cep97 as the fusion partner only invites additional co-purifying centriole proteins in the protein purification prep. Again, the rationale for making this particular fusion protein is not at all clear but it should be made clear. The authors state that CEP97^CP110 localizes to centrioles in U2OS cells but do not show the data. This should be shown as well as any effects on centrioles (length, number and morphology)? Does the chimera localize specifically to the distal tips of centrioles? If the chimeric protein localizes normally to centrioles, then shouldn't it retain the ability to bind other centriole proteins, and wouldn't those proteins contaminate the protein purification prep?*

We have now explained our reasoning for making the chimeric CEP97-CP110 protein better (page 5 of the revised paper, see below) and added data showing the localization of CP110 fragments and the CEP97^CP110 chimera in cells (new Figure S2, see next page).

“We next generated different GFP-tagged CP110 deletion mutants. Since the N-terminus of CP110 is known to bind to the middle part of CEP97 (Spektor et al., 2007), we hypothesized that the C-terminus of CP110 is responsible for blocking MT growth. When expressed in cells, full-length CP110 and CEP97, as well as the N-terminal CP110 fragment 1-700 localized to centrosomes. However, GFP-tagged C-terminal CP110 fragments 581-991 and 791-991 did not associate with either centrosomes or microtubules (Fig.S2A), and we were not successful in purifying them from HEK293 cells. Since CP110 normally functions in association with CEP97, we reasoned that a direct fusion with CEP97 might improve the folding or stability of the CP110. We screened different fusion constructs by their localization in U2OS cells and found that a protein containing residues 1-650 of CEP97 and residues 581-991 of CP110 (termed here CEP97^CP110, Fig. 1A) localized to centrioles (Fig. S2A). Ultrastructure expansion microscopy (U-ExM), which entails 4.5x expansion of cells, followed by confocal imaging (Gambarotto et al., 2019) demonstrated that CEP97^CP110-GFP bound to the distal centriole ends, similar to CP110 (Fig. S2B). In vitro, SII and GFP-tagged CEP97^CP110 blocked MT seed elongation and was much less aggregation-prone than full-length CP110 (Fig. 1D-F, Video S1).”

We note that fusing of a small soluble protein, GFP, to C-terminal fragments of CP110 has not made them functional in vitro, and we have not tested any fusions with other unrelated proteins. Instead, we generated an array of CEP97^CP110 fusions and investigated their ability to localize to the distal centriole end in cells, because this provided an easy functional test. We chose the best-behaved one, which we used for subsequent experiments. The localization of this fusion protein to distal centriole ends by Expansion Microscopy is now illustrated in the new Fig. S2B (see next page). Furthermore, our mass spectrometry analysis shows that in purification and washing conditions we used, the chimeric CEP97^CP110 protein is not contaminated with other centriolar proteins, such as CPAP, to which it can specifically bind (new Fig. 5E).

Overall, we fully understand that the approach we have chosen feels weird and unconventional. Unfortunately, CP110 and CPAP are very difficult to work with. Our lab has extensive experience with working with purified microtubule regulators (EBs, MCAK, CLASP, CAMSAPs, katanin, ASPM, chTOG, TOGARAM1, CEP104, etc) and making deletion mutants of these proteins. For example, for our studies of CLASP2, we made >20 different deletion mutants, and they all worked (Aher et al, Dev Cell 2018). Unfortunately, working with centriolar proteins is very difficult, possibly because they evolved to function in complexes with multiple partners that specifically concentrate them at the centrioles. We certainly do not exclude that it would be possible to make other functional fusions of the CP110 C-terminus, but within this very long-going project, we ultimately chose to focus on an approach that does work. We cannot exclude that the N-terminus of CP110 contributes to microtubule regulation, but we do show that the effects of full-length CP110 and the CEP97^CP110 chimera in vitro are quite similar, and the binding site for CPAP is preserved in the chimeric protein.

Supplementary figure S2

Lastly, the authors need to show that CEP97^CP110 can bind the CPAPwt protein but not the CPAPmut protein.

Thank you for the good suggestion. We have included the Western blot in the new Fig. 5E which shows that CEP97^CP110-GFP co-immunoprecipitates with CPAP-N_{WT} but not CPAP-N_{MUT}.

3. Results in Figure 4G and conclusions from Figure 4 appear to be incongruent. In the right panel of 4G, the microtubule at the start of the kymograph is capped by Cep97-CP110 and growth appears inhibited. After FRAP, neither protein recovers until new CPAP is recruited, resulting in slow growth. Only after the microtubule pauses, then do we see more Cep97-CP110 recruited to the stalled microtubule. Is the interpretation that the bleached cap was removed when CPAP is recruited, thus triggering microtubule growth and that CEP97^ΔCP110 is recruited to stalled microtubule ends? This appears similar to Figure S6D, where loss of CPAP/CEP97^ΔCP110 precedes microtubule regrowth followed by recruitment of these proteins to the growing microtubule. Overall, I do not believe that there is enough evidence to conclude that CPAP allows Cep97-CP110 to track the tips of growing microtubules. Rather, it seems equally likely that 1) CPAP is displacing the cap to promote growth, 2) CPAP recruits Cep97-CP110 when present, or 3) microtubule shape is determining if the cap proteins are recruited. Do the authors see an increase in pausing and restarting when CPAP is added? Only about 20% of microtubules exhibit this in 20nM CEP97^ΔCP110, although pausing followed by catastrophe does not appear to be separated. On microtubules that display a continuous slow growth rate, do the authors see that fluorescent intensity never recovers? If the authors stabilized microtubule ends with (taxol?), would they see an increased recruitment of Cep97-CP110 to these ends?

The reviewer is completely right that all fast growth episodes in the presence of CEP97^ΔCP110 and CPAP are due to the dissociation of the chimera. On rapidly growing microtubule ends, CPAP exchanges rapidly, as shown in Fig.7A of the revised paper. The important question is, however, what happens at the slowly growing ends, such as those shown in Fig. 6D of the revised manuscript, when CEP97^ΔCP110 is present. FRAP experiments show no recovery in the CEP97^ΔCP110 channel and reduced recovery in the CPAP channel (see updated Fig.7A,B). These data argue in favor of our interpretation. Experiments when CPAP is added to the assay where CEP97^ΔCP110 is already present are shown in Fig. 7D of the revised paper. CPAP cannot displace the chimera from the blocked seed end, but when pre-mixed with the chimera, it can trigger slow growth of microtubule ends that were not blocked. This suggests that the complex of CP110 and CPAP promoting processive slow MT growth is likely formed in solution and not on the microtubule tips blocked by CP110.

Furthermore, we agree with the reviewer that protofilament shapes at microtubule tips, cap binding and growth are likely to be related to each other, but the exact relationship requires further clarification. It is important to note that both CP110 and CPAP can affect microtubule dynamics when their number at the tip is not sufficient to bind to each protofilament end,

and different protofilaments at the same MT tip can have different conformations and likely also different polymerization states. This strongly complicates the analysis and the potential conclusions derived from this analysis, and would require additional cryo-ET studies.

Adding Taxol sounds like an interesting thing to try but is actually impossible to do in our dynamic assays, because at concentrations that would trigger not just very transient pausing but microtubule stabilization, in the presence of free tubulin, there will be massive microtubule nucleation in solution, and the TIRF assay will look like a “microtubule rain”. The paper includes cryo-ET data with GMPCPP-stabilized seeds in the absence of free tubulin (cryo-ET data in Figure 3 of the revised manuscript). CEP97^ΔCP110 chimera binds to such seeds, and the frequency of plus-end capping is the same with and without free tubulin, 78% (Fig. 2C), but comparing recruitment of a protein to microtubule ends with and without free tubulin is problematic, because free tubulin has a very strong effect on the interactions of microtubule-associated proteins with microtubules. For example, in the absence of free tubulin, CPAP strongly binds along the microtubule shaft and loses its preference for microtubule plus ends.

Additional concerns:

4. *Page13 - 3rd line from the top "MTs grew at a rate..." not "MTs that grew at a rate"*

Thank you for pointing this out, we have corrected the text.

5. *Scale appears incorrect in S7H.*

We apologize for the mistake which has been corrected in the revised version of the paper (where this figure is now Fig. S5E).

6. *Misspelled CP110 as CEP110 in Discussion (pg 20).*

We apologize for the mistake which has been corrected.

7. *Figures 1B, 1D, 4B, 4C, 4D: use different colors for the arrowheads instead of red and green.*

We have changed the color of these arrowheads to white.

8. *Figure 4E: is there a significant difference between CPAPwt and CEP97^ΔCP110+CPAPwt.*

In this figure (Fig 6G of the revised manuscript), we used Brown-Forsythe and Welch ANOVA, corrected with Games-Howell test for multiple comparisons, to calculate if there are any significant differences between the mean growth rates of different conditions. The mean growth rate of CPAP-N_{WT} and CPAP-N_{WT} + CEP97^ΔCP110 is significantly different (****) with a p value < 0.0001.

9. *Addition of CPAP to microtubules causes the microtubule growth rate to decrease. Why do the authors then refer to CPAP as a microtubule polymerase?*

CPAP structure has hallmarks of a microtubule polymerase, similar to chTOG: it has a tubulin-binding domain (PN2-3) and microtubule-binding domain. CPAP indeed slows down growth of free-growing microtubule plus ends, and this is due to the fact that through its LID domain,

it can transiently occlude the outward-facing surface of beta-tubulin. However, in the context of the centriolar cap, CPAP acts as a polymerase: it overcomes the block of microtubule growth imposed by CP110 and imparts slow but processive tubulin addition to an end which would be lacking a long stabilizing GTP cap. This fully aligns with the well-known ability of CPAP, also illustrated by our data (Fig. 8D,F of the revised manuscript), to cause centriole overelongation at elevated expression levels.

10. *The authors should consider changing the title of the paper because CPAP is not recognized as a centriolar capping protein.*

For a long time, it was thought that CPAP is only present at the proximal centriole end, making the observation that CPAP overexpression causes centriole overelongation very puzzling. However, recent studies from the Loncarek lab (Vasquez-Limeta et al., 2022) and Hamel&Guichard lab (Laporte et al., 2024) have convincingly shown that CPAP localizes to the distal end of the procentriole along with other capping proteins like CP110 and CEP97. Even though it doesn't block centriolar growth, CPAP is present in the distal centriole protein cap and controls centriole elongation, making it a functional part of the cap.

11. *Figure 4F: is the decrease in CEP97^CP110 localization to microtubules with CPAPmut significantly different compared microtubules with no CPAP? Why would it be less if it can't bind CPAP?*

At 20 nM CEP97^CP110 present in vitro, CEP97^CP110 bound to the microtubule plus ends for 46% of total time. When CPAP-N_{WT} was present along with CEP97^CP110, the total dwell time of CEP97^CP110 increased to 69% and together with CPAP-N_{WT}, the dwell time decreased to 26%. Whereas the differences with CEP97^CP110 alone were not statistically significant, there was a significant difference in the dwell time of CEP97^CP110 between the conditions with CPAP-N_{WT} and CPAP-N_{MUT} (Fig. 6F of the revised paper). The observed effect is relatively mild likely because the binding sites for the two proteins are not saturated in our experimental conditions. When CPAP cannot bind to CP110, it could interfere with the binding of the chimera by altering the conformation of the protofilaments at microtubule tips. Further structural work would be needed to resolve this question.

12. *Figures 6C and 6G: what is significant in these graphs and what is not? This is important in order to make any conclusions.*

We fully agree. Statistical comparisons have been included in the figure and the corresponding legend (Figure 9B,C,G of the revised manuscript). The differences in centriole length (Fig. 9B) and CPAP localization (Fig. 9G), which form the basis of the main conclusions in the paper are significant for the cells expressing CPAP mutant.

13. *Missing reference. Direct binding between CP110 and Sas4 has been shown with Drosophila protein fragments (Galletta et al., Nat Communications, 2016). Specifically, the N-terminus of CP110 interacts via yeast 2-hybrid with Sas4 NT.*

We apologize for the omission and thank the reviewer for pointing out this paper. We have included the citation on pages 4 and 9.

14. Page 12, paragraph 3, sentence 2: Did the authors mean to say CPAPmut in this sentence?

Thank you for pointing out the mistakes which we have corrected in the revised paper.

15. Page 13, paragraph 1, last sentence: The authors should rephrase their concluding sentence here. Perhaps, "these data indicate that a mixture of CPAP and CP110^ΔCEP97 causes an even slower rate of MT growth compared to CPAP alone". These data do not show that this requires direct interaction because a direct interaction with these new protein constructs has not been shown.

Thank you for the good suggestion. We have now included a Western blot in Fig. 5E which shows the co-IP data of CEP97^ΔCP110 and CPAP variants. The blot shows that CEP97^ΔCP110-GFP can interact with CPAP-N_{WT} but not CPAP-N_{MUT}.

16. Page 15, paragraph 3, sentences 2-3: from the graph in Figure 6C, it appears that CPAP binding is not important for robust CP110 localization since >90% of CP110 localizes properly and there are no measurements of intensity to say otherwise.

We agree that the differences between the cells expressing wild type and mutant CPAP with respect to CP110 localization are minor and removed the statement about robust localization of CP110.

Reviewer 3:

Centrioles are small organelles whose distinguishing feature is an outer wall composed of a 9-fold array of stabilized microtubule blades. Remarkably, during centriole assembly, centriolar microtubules elongate very slowly to yield centrioles that are within a reproducible length range. How this is achieved is not well understood. Two microtubule-binding regulators that have been implicated in this process are CPAP, which is thought to promote the growth of centriolar microtubules, and CP110/CEP97, which is involved in capping the ends of centriolar microtubules to limit their extension. In this paper, the authors identify forms of these proteins that enable characterization in vitro and use a combination of in vitro assays, structural work, and in vivo approaches to dissect the roles of CPAP and CP110 and their collaboration in the growth of centriolar microtubules. They identify a CEP97^CP110 chimera that does not aggregate in vitro and is suitable for in vitro assays. In vitro assays show that this chimera dimerizes, specifically binds to microtubule plus ends, and when recruited stoichiometrically to the number of protofilaments blocks further microtubule elongation. Using cryo-electron tomography, they show that CEP97^CP110 forms a cap or partial cap that sits in the MT lumen. They next go on to examine the interplay between CEP97^CP110 and CPAP. Through a very nice biochemical/structural analysis they identify a coiled-coil region in CPAP (CC1) that interacts in an antiparallel fashion with a coiled-coil region in CP110 and specific mutations that disrupt this interaction. They show that whereas microtubule plus ends are capped when CEP97^CP110 is added, simultaneous addition of CPAP allows these capped ends to slowly grow. By introducing the mutation that prevents the interaction of the CP110 and CPAP coiled-coil regions in vivo, they show that centrioles exhibit structural defects, with one prominent defect being that the centrioles are too short in the presence of the mutant. Overall, the work provides significant insight into key players that control the growth and stability of centriolar microtubules and is appropriate for publication in the JCB. However, in a few cases, which I discuss below, I think that the presentation of the data could be improved to allow the main points to come through more clearly.

We thank the reviewer for the positive and thoughtful comments on our manuscript.

Main points:

1. *Although Figure 2A-C are quite effective, the points made in Fig. 2D-I are much less clear. Fig. 2F makes the point that protofilament length, defined as the region after the protofilament first bends outwards, is shorter in the microtubules capped by CEP97^CP110 than in controls. First, the y-axis label on this graph does not seem to be correct - why is the graph labeled deg/dimer if this is a measure of protofilament length?*

Thank you for pointing out the mistake in the axis of the graph, which has been corrected to nm in the revised manuscript.

Figs. 2H-I are also confusing. For Fig. 2H, the question arises of whether it is fair to compare average curvature of the whole protofilament extension of the capped and non-capped ends, given that the extensions are almost twice as long in the uncapped compared to the capped cases. It is mentally hard to figure out what this comparison means. It might be better just to show the graph currently in 2G. However, the issue with the graph in Fig. 2G is that it ignores the fact that most of the protofilament extensions are less than 15 nm long in the capped case compared to more than 15 nm long in the uncapped case. For short

distances, like 5 nm, it does seem like the capped microtubule protofilaments show less curvature than the protofilaments in the uncapped microtubules. Since protofilament length is only measured after the point where the protofilament first bends away from the MT cylinder, is this analysis biased due to the fact that it ignores protofilaments that don't bend away from the MT cylinder? What if all protofilament lengths and curvatures were measured from the point when the first protofilament bent away from the MT cylinder-would a much higher percentage of the other protofilaments exhibit reduced curvature for longer in the capped than uncapped cases?

We have tried to analyze protofilament shapes at microtubule ends in our samples in different ways, but all analyses boil down to the conclusion that in the presence of CEP97^ΔCP110 cap, the length of protofilaments at the plus ends deviating from the microtubule cylinder is shorter, whereas the curvature of protofilaments, once they deviate from the microtubule shaft, is similar. To further illustrate this point, we have now included an additional quantification where we measured the curvature of terminal protofilament segment (new Figure 2H), which is similar to the average curvature and does not differ per condition.

- 2. The presentation of the data in Fig. 4E/F is not very effective at conveying the main points. The left graph should potentially be labeled "Mean growth rate". The left axis is labeled % of time spent by MT plus end. However, this seems to be a cumulative distribution plot, in which what is plotted is actually the % of time that the microtubule end spent going at the indicated rate or slower.*

We have corrected the left graph in this panel (Fig. 6G of the revised paper), and the axis now reads "Mean growth rate".

It might be worth thinking about the best way to present this data, as I am not sure that the general readers can easily grasp what is being shown here and what the main point is. The key takeaway from this is that CPAP prevents CEP97^ΔCP110 from irreversibly capping MT plus ends-converting these ends to ends that are instead slowly growing. It would be nice if there could be a graph of a summary statistic that makes this point more directly in the main figure. For example, a plot of mean growth rate of microtubule ends with associated CEP97^ΔCP110, which presumably is undetectable (given the assay limitations) in the absence of CPAP and then a low positive rate in the presence of either form of CPAP. The second point made by these panels is that the CPAP mutation that prevents CEP97^ΔCP110 binding reduces the amount of time that CEP97^ΔCP110 spends at microtubule ends, so the conclusion here would be that when CPAP is present it would suppress the binding of CEP97^ΔCP110 to microtubule ends if not for its CC1 interaction region, which instead allows CPAP to increase the number of microtubule ends with CEP97^ΔCP110 present.

To better convey these points to the reader, it would be good to put F before E to highlight point 2-that CPAP promotes the association of CEP97^ΔCP110 in a fashion dependent on the identified coiled-coil region first (making the point that its presence would inhibit CEP97^ΔCP110 if it were not for the coiled-coil interaction region). This will provide a really nice transition from the data in Figures 2 and 3. After this, have a panel that provides statistics for mean growth rates when the CEP97^ΔCP110 cap is present or not for the

different conditions-maybe make one graph for CEP97^CP110 cap present and one graph for CEP97^CP110 cap absent so that the main point comes through-when microtubules have a CEP97^CP110 cap, the presence of CPAP allows them to continue to grow at a slow rate.

We agree that the plots are not easy to understand, and discussed them with different colleagues, but found that different people find different plots more intuitive or less intuitive. At the end, we have followed the excellent suggestion to change the flow of the plots in the figure (Fig. 6 of the revised manuscript). Further, based on the reviewers' suggestion, we have now also included separate plots that provide statistics for mean growth rates when the fluorescent signal of CEP97^CP110 cap is either present or absent (new Fig. 6H). However, since some people, including the authors of this paper, are used to interpreting cumulative plots, we included these as well (bottom panel of Fig. 6G of the revised manuscript).

- 3. The in vivo analysis is nicely done. The main point of this is that centrioles are shorter in the presence of the mutant protein than in the presence of the WT protein. However, it was a bit unsatisfying in the sense that the paper did not do a good job of bringing things together to explain (or suggest an explanation for) why this could/would be the main phenotype observed.*

We bring things together at the end of the Discussion by stating: "Both in cells and in vitro, the two proteins can bind to MT tips independently of each other, and the interaction between the two proteins is not essential for centriole formation. However, this interaction promotes the localization of CPAP to the distal centriole end in cells and centriole elongation in the S phase. In vitro, the binding between the two proteins leads to long episodes of slow MT growth, whereby CP110 does not exchange, and the turnover of CPAP is inhibited, indicating that together, CPAP and CP110 form a persistent "cap" that promotes very slow tubulin addition. This observation is consistent with CPAP acting as a polymerase that can overcome MT growth inhibition imposed by CP110, explaining the opposing roles of the two proteins in controlling centriole length in mammalian cells (Kohlmaier et al., 2009; Schmidt et al., 2009; Tang et al., 2009). Perturbation of the CP110-CPAP interaction induced centriole defects in some cells, indicating that it underlies one of the mechanisms controlling robust centriole biogenesis."

November 18, 2024

RE: JCB Manuscript #202406061R

Anna Akhmanova
Utrecht University

Dear Dr. Akhmanova:

Thank you for submitting your revised manuscript entitled "Centriolar cap proteins CP110 and CPAP control slow elongation of microtubule plus ends". Reviewers commended the new data provided in this revision and unanimously support publication. Reviewer 2 made one suggestion concerning your discussion of CP110-CPAP complex formation, which we invite you to clarify in the text. In light of this strong reviewer enthusiasm we would be happy to publish your paper in JCB pending final revisions necessary to meet our formatting guidelines (see details below).

1) Text limits: Character count for Articles is < 40,000, not including spaces. Count includes abstract, introduction, results, discussion, and acknowledgments. Count does not include title page, figure legends, materials and methods, references, tables, or supplemental legends.

2) Figures limits: Articles may have up to 10 main figures and 5 supplemental figures/tables.

** If possible, please endeavor to reduce supplemental figures to 5. We appreciate the diagram in supplemental figure 6 is helpful. If there is no reasonable way to reduce supplemental figures, we can consider granting an exception to this formatting guideline.

3) Figure formatting: Scale bars must be present on all microscopy images, including inset magnifications. Molecular weight or nucleic acid size markers must be included on all gel electrophoresis. Please avoid pairing red and green for images and graphs to ensure legibility for color-blind readers. If red and green are paired for images, please ensure that the particular red and green hues used in micrographs are distinctive with any of the colorblind types. If not, please modify colors accordingly or provide separate images of the individual channels.

** Please add scale bars to Supplemental Figure 3C.

4) Statistical analysis: Error bars on graphic representations of numerical data must be clearly described in the figure legend. The number of independent data points (n) represented in a graph must be indicated in the legend. Statistical methods should be explained in full in the materials and methods. For figures presenting pooled data the statistical measure should be defined in the figure legends. Please also be sure to indicate the statistical tests used in each of your experiments (either in the figure legend itself or in a separate methods section) as well as the parameters of the test (for example, if you ran a t-test, please indicate if it was one- or two-sided, etc.). Also, if you used parametric tests, please indicate if the data distribution was tested for normality (and if so, how). If not, you must state something to the effect that "Data distribution was assumed to be normal but this was not formally tested."

5) Abstract and title: The abstract should be no longer than 160 words and should communicate the significance of the paper for a general audience. The title should be less than 100 characters including spaces. Make the title concise but accessible to a general readership.

6) Materials and methods: Should be comprehensive and not simply reference a previous publication for details on how an experiment was performed. Please provide full descriptions in the text for readers who may not have access to referenced manuscripts. We also provide a report from SciScore and an associate score, which we encourage you to use as a means of evaluating and improving the methods section.

7) Please be sure to provide the sequences for all of your primers/oligos, plasmids, and RNAi constructs in the materials and methods. You must also indicate in the methods the source, species, and catalog numbers (where appropriate) for all of your antibodies. Please also indicate the acquisition and quantification methods for immunoblotting/western blots.

8) Microscope image acquisition: The following information must be provided about the acquisition and processing of images:

- a. Make and model of microscope
- b. Type, magnification, and numerical aperture of the objective lenses
- c. Temperature
- d. Imaging medium
- e. Fluorochromes
- f. Camera make and model

g. Acquisition software

h. Any software used for image processing subsequent to data acquisition. Please include details and types of operations involved (e.g., type of deconvolution, 3D reconstitutions, surface or volume rendering, gamma adjustments, etc.).

10) Supplemental materials: There are strict limits on the allowable amount of supplemental data. Articles may have up to 5 supplemental figures. Please also note that tables, like figures, should be provided as individual, editable files. A summary of all supplemental material should appear at the end of the Materials and methods section.

13) ORCID IDs: ORCID IDs are unique identifiers allowing researchers to create a record of their various scholarly contributions in a single place. At resubmission of your final files, please provide an ORCID ID for all authors.

15) A data availability statement is required for all research article submissions. The statement should address all data underlying the research presented in the manuscript. Please visit the JCB instructions for authors for guidelines and examples of statements at (<https://rupress.org/jcb/pages/editorial-policies#data-availability-statement>).

Please note that JCB requires authors to submit Source Data used to generate figures containing gels and Western blots with all revised manuscripts. This Source Data consists of fully uncropped and unprocessed images for each gel/blot displayed in the main and supplemental figures. Since your paper includes cropped gel and/or blot images, please be sure to provide one Source Data file for each figure that contains gels and/or blots along with your revised manuscript files. File names for Source Data figures should be alphanumeric without any spaces or special characters (i.e., SourceDataF#, where F# refers to the associated main figure number or SourceDataFS# for those associated with Supplementary figures). The lanes of the gels/blots should be labeled as they are in the associated figure, the place where cropping was applied should be marked (with a box), and molecular weight/size standards should be labeled wherever possible. Source Data files will be directly linked to specific figures in the published article.

WHEN APPROPRIATE: The source code for all custom computational methods published in JCB must be made freely available as supplemental material hosted at www.jcb.org. Please contact the JCB Editorial Office to find out how to submit your custom macros, code for custom algorithms, etc. Generally, these are provided as raw code in a .txt file or as other file types in a .zip file. Please also include a one-sentence summary of each file in the Online Supplemental Material paragraph of your manuscript.

** If you prefer, source code used in this manuscript may refer readers to the links indicated in the methods section.

Journal of Cell Biology now requires a data availability statement for all research article submissions. These statements will be published in the article directly above the Acknowledgments. The statement should address all data underlying the research presented in the manuscript. Please visit the JCB instructions for authors for guidelines and examples of statements at (<https://rupress.org/jcb/pages/editorial-policies#data-availability-statement>).

B. FINAL FILES:

Thank you for your attention to these final processing requirements. Please revise and format the manuscript and upload materials within 7 days. If you need an extension for whatever reason, please let us know and we can work with you to determine a suitable revision period.

Thank you for this interesting contribution, we look forward to publishing your paper in Journal of Cell Biology.

Sincerely,

Alexey Khodjakov
Monitoring Editor
Journal of Cell Biology

Tim Fessenden
Scientific Editor
Journal of Cell Biology

Reviewer #1 (Comments to the Authors (Required)):

This paper makes an important contribution to our understanding of the ways cells control the rate of microtubule elongation during centriole formation. The current version addresses all of the specific comments made by the reviewers and is now ready for publication. I have no further criticisms.

Reviewer #2 (Comments to the Authors (Required)):

The revision is great, and I recommend publication.

My only request is a few additions to the text to clarify the author's interpretation around CPAP promoting Cep97-CP110 tip tracking.

In their rebuttal letter for Major Concern 3, the authors state:

CPAP cannot displace the chimera from the blocked seed end, but when pre-mixed with the chimera, it can trigger slow growth of microtubule ends that were not blocked. This suggests that the complex of CP110 and CPAP promoting processive slow MT growth is likely formed in solution and not on the microtubule tips blocked by CP110.

The authors should include their "formed in solution" point in the text. Then, in the Discussion, they should comment on how this could be achieved at the centriole distal tips (perhaps high CP110 turnover rates).

Reviewer #3 (Comments to the Authors (Required)):

I am satisfied with the changes the author made in the revised manuscript. Overall, it is a very nice paper.